# Nasopharyngeal carcinoma cells promote regulatory T cell development and suppressive activity via CD70-CD27 interaction

Lanqi Gong [1,2,10], Jie Luo[1,10], Yu Zhang[3,4,5,10], Yuma Yang[1], Shanshan Li[2], Xiaona Fang[1], Baifeng Zhang [1], Jiao Huang[1], Larry Ka-Yue Chow [1], Dittman Chung[1], Jinlin Huang[1], Cuicui Huang[1,2], Qin Liu[1,2], Lu Bai[1,2], Yuen Chak Tiu[1], Pingan Wu[6], Yan Wang[7], George Sai-Wah Tsao[8], Dora Lai-wan Kwong[1,2], Anne Wing-Mui Lee [1,2,9], Wei Dai [1,2] & Xin-Yuan Guan [1,2,4,5,9] ✉

Despite the intense CD8+ T-cell infiltration in the tumor microenvironment of nasopharyngeal carcinoma, anti-PD-1 immunotherapy shows an unsatisfactory response rate in clinical trials, hindered by immunosuppressive signals. To understand how microenvironmental characteristics alter immune homeostasis and limit immunotherapy efficacy in nasopharyngeal carcinoma, here we establish a multi-center single-cell cohort based on public data, containing 357,206 cells from 50 patient samples. We reveal that nasopharyngeal carcinoma cells enhance development and suppressive activity of regulatory T cells via CD70-CD27 interaction. CD70 blocking reverts Treg-mediated suppression and thus reinvigorate CD8+ T-cell immunity. Anti-CD70+ anti-PD-1 therapy is evaluated in xenograft-derived organoids and humanized mice, exhibiting an improved tumor-killing efficacy. Mechanistically, CD70 knockout inhibits a collective lipid signaling network in CD4+ naïve and regulatory T cells involving mitochondrial integrity, cholesterol homeostasis, and fatty acid metabolism. Furthermore, ATAC-Seq delineates that CD70 is transcriptionally upregulated by NFKB2 via an Epstein-Barr virus-dependent epigenetic modification. Our findings identify CD70+ nasopharyngeal carcinoma cells as a metabolic switch that enforces the lipid-driven development, functional specialization and homeostasis of Tregs, leading to immune evasion. This study also demonstrates that CD70 blockade can act synergistically with anti-PD-1 treatment to reinvigorate T-cell immunity against nasopharyngeal carcinoma.

Nasopharyngeal carcinoma (NPC) is an epidemiologically prevalent malignancy in Asia and North Africa, intensively driven by chronic Epstein-Barr virus (EBV) infection. In the past decade, the age-standardized NPC mortality has been continuously reduced from 0.93 to 0.86[1], as a result of regional economic growth, improved healthcare awareness, large-scale population screening of EBV, and optimization of chemoradiotherapy, but the survival and life quality of patients with advanced and treatment-resistant

NPC have not progressively improved[2,3]. In our most recent randomized clinical trials at the Queen Mary Hospital and The University of Hong Kong (HKU)-Shenzhen Hospital, more than 30% of the most advanced locoregional NPC patients relapsed and progressed despite receiving the best chemoradiotherapy available[4]. Therefore, recurrence, metastasis, and chemoradiotherapy resistance remain significant challenges in the clinical management of NPC.

The NPC microenvironment is a highly heterogenous ecosystem infiltrated with various effector subpopulations, including cytotoxic T cells, helper T cells, memory B cells and plasma cells, collectively caused by chronic EBV infection and locoregionally enriched lymphoid structures[5]. Thus, for patients with advanced and chemoradiotherapy-resistant NPC, immunotherapy has been theoretically considered as a promising strategy that mobilizes in situ and peripheral effector cells to combat tumor progression. Nevertheless, recent clinical trials have demonstrated that anti-PD-1 monotherapy using pembrolizumab, nivolumab, and Camrelizumab, only generated a <3% complete response rate and a < 35% partial response rate in NPC patients[6–8]. Hence, there is an unmet clinical need that should be addressed by characterizing vital tumor microenvironment (TME)-related factors contributing to immunotherapy resistance, and evaluating how to enhance the immunotherapy response by targeting these factors.

Since 2020, the cellular constituents and functional dynamics of the NPC microenvironment have been initially identified and characterized by single-cell RNA sequencing (scRNA-seq) at multiple centers[9–12]. In the NPC microenvironment, many tumor-infiltrating effector T cells are highly specific to tumor and EBV antigens[13,14]. However, single-cell analysis has shown that these anti-tumor T cells are in disturbed immune homeostasis, exhibiting defective proliferation and cytotoxicity[5]. Such disrupted homeostasis might be modulated by various inhibitory signals, such as TGF-β, IL-10, IL-35, and cyclic adenosine monophosphate (cAMP), from abundant effector Tregs (eTregs) in the TME[5,15]. In noncancerous inflammatory nasopharyngeal (INP) tissues, only a minor fraction of Tregs is present to maintain immune homeostasis, whereas the TME contains a much higher proportion of Tregs, as tumor cells might recruit peripheral naïve Tregs (nTregs) and activate nTregs and resting Tregs (rTregs), which in turn facilitate the NPC cells to escape from immune surveillance[10,15,16]. The development and suppressive function of Tregs in the NPC microenvironment is a dynamic and context-specific process that has yet to be insufficiently investigated. The major obstacle for anti-PD-1 immunotherapy is that therapeutically expanded T cells can rapidly become dysfunctional due to in situ immunosuppressive signals unaffected by such treatment, leading to de novo and required resistance[17,18]. This phenomenon might explain the unsatisfied response rate of anti-PD-1 immunotherapy even in immune-inflamed NPC tumors. Therefore, targeting immunosuppressive subtypes, primarily FOXP3+ Tregs, might promote anti-tumor immunity and enhance the efficacy of PD-1 blockade[16,19,20].

In this work, to better understand how Tregs develop and exert suppressive functions in the NPC microenvironment from an integrated multi-omic perspective, we show that CD70+ NPC cells can enhance Treg development, homeostasis, and suppressive activity via lipid metabolism reprogramming. Active lipid metabolism was indispensable for the downstream immunosuppressive effect of CD70-CD27 signaling in Tregs. Therapeutic inhibition of CD70 overcomes Treg immunosuppression, enhances anti-tumor immunity, and synergizes with PD-1 blockade in pre-clinical animal models. Furthermore, the EBV-NFKB2-CD70 axis is identified as of high importance in regulating immune evasion during NPC progression. Taken together, these results provide a molecular and pre-clinical basis for optimizing first-line immunotherapy regimens and reducing mortality in NPC patients.

## Results

### Increased intratumoral Treg abundance and activation are regulated by NPC-mediated interaction

To comprehensively investigate the developmental and functional dynamics of intratumoral Tregs affected by NPC cells at single-cell resolution, we established a large-scale cohort containing 189,750 T cells clustered into 41 subtypes from 36 NPC tissues, 10 paired NPC peripheral blood samples, and 4 INP tissues, originated from 3 NPC studies[9–11] (Fig. 1A, S1A, and S1B). Within the T-cell subpopulations, we identified three FOXP3+ Treg subtypes, namely nTregs, rTregs, and eTregs, with distinct transcriptome profiles[15] (Supplementary Fig. 1c and Supplementary Data 1). The single-cell trajectory analysis inferred two developmental lineages of CD4+ T cells, where naïve T cells either differentiated into helper T cells or eTregs, transcriptionally driven by *BATF* and *FOXP3* (Figs. 1b, 1c and Supplementary Fig. 1d).

To understand how the lineage commitment of CD4+ T-cell subtypes was shifted in the TME, we first quantified the normalized abundance of each subtype across patients. Inflammatory tissues and NPC peripheral blood samples harbored enriched fractions of CD4+ naïve T cells and *SELL* + nTregs, whereas in the TME, these fractions were limited (Fig. 1d). Among the differentiated CD4+ T-cell subtypes, rTregs and eTregs exhibited a TME preference, while follicular helper T (T_FH) cells and central memory T (T_CM) cells were more enriched in INP tissues (Fig. 1d and Supplementary Fig. 1e). Then, to confirm that the homeostasis between Tregs and naïve CD4+ T cells was impaired in the TME, we established two multivariate linear regression models using eTreg and naïve T cell-specific signatures to compute the Treg immunosuppressive activity and T-cell naïveness based on NPC bulk RNA-seq cohorts (Supplementary Figs. 1f–i). Based on the functional modules, we illustrated that a high Treg and low naïve T cell infiltration was a malignant hallmark in NPC patients and associated with inferior prognosis (Figs. 1e, f). CIBERSORTx deconvolution based on scRNA-seq signatures further validated that the higher intratumoral infiltration and suppressive activity of Tregs collectively contributed to worse prognosis, but were not associated to other clinical parameters in NPC patients (Supplementary Figs. 1j–l and Supplementary Table 1). As the most developed subtype in the TME, eTregs had the highest expression of immunosuppressive signatures, such as *CTLA4*, *LAYN* and *TNFRSF4*, as well as the most robust cytokine communication, metabolism and T-cell suppression activities, collectively induced by STAT, TGF-β and IFN signaling[21] (Figs. 1g, h). In contrast, nTregs and rTregs, modulated by canonical WNT signaling[22], exhibited minimal regulatory effects on the activation, proliferation and apoptosis of T cells (Fig. 1h). Considering an eTreg-polarized CD4+ T-cell landscape in NPC, we hypothesized that increased intratumoral Treg abundance and suppressive activity might be collectively caused by tumor-mediated CD4+ naïve T cell-to-Treg development and Treg activation.

Therefore, to delineate the tumor-dependent mechanism by which Tregs developed and activated in the TME, we established in vitro co-culture systems where CD4+naïve T cells were either directly or indirectly co-cultured with the NPC cell line C666 or normal nasopharyngeal epithelial (NPE) cell lines, NP460 and NP69 (Fig. 1l). Compared to induced Treg development from CD4+ naïve T cells in the absence of tumor cells and the transwell-based co-culture system (Supplementary Figs. 1m,n), the direct co-culture system demonstrated that C666 cells enhanced polarization from CD4+ naïve T cells to FOXP3+ Tregs and upregulated the vital eTreg marker, CTLA4[23] (Figs. 1j–l). The Treg-polarizing and activating effect of C666 cells was further validated by qRT-PCR of a selected gene panel related to lineage determination and functionality of Tregs (Fig. 1m), and by ELISAs of secreted immunosuppressive factors in the co-culture system, including IL-10, TGF-β, and adenosine (Fig. 1n). Altogether, transcriptome analysis and functional assays preliminarily suggested that the development and activation of Tregs in NPC were regulated by cell-cell contact instead of cytokine communication.

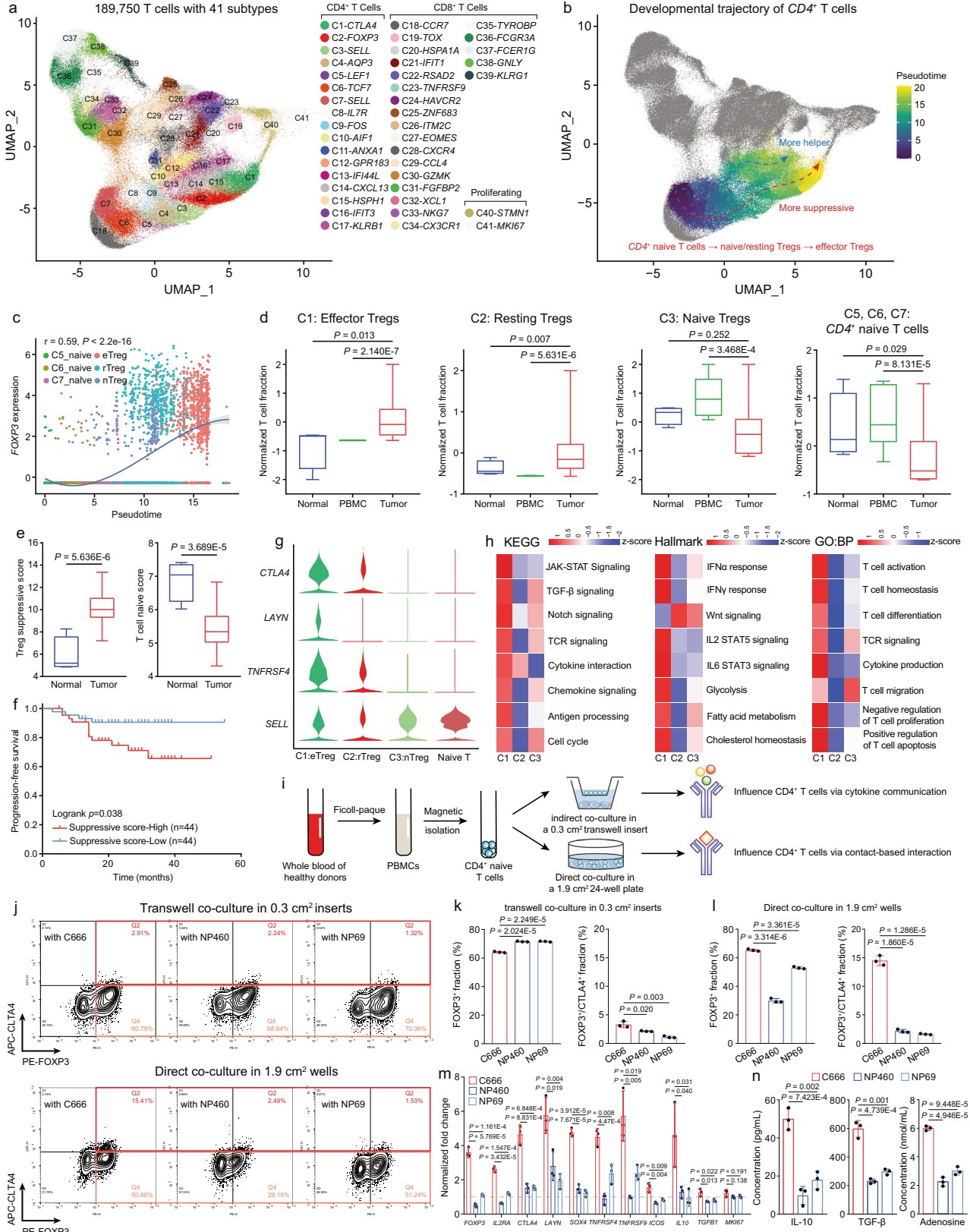

## Tumor-restricted CD70 is correlated to Treg abundance and activation via interacting with CD27

To further identify the exact mode of action occurring among CD4+ naïve T cells, Tregs, and NPC cells, we used two well-established programs, CellPhoneDB[24] and CellChat[25], that evaluated the strength and direction of cell-cell interactions based on scRNA data. CD70-CD27 signaling was one of the most predominant interactions between NPC cells, naïve CD4+ T cells and Treg subtypes and occurred in a CD70/CD27 expression-dependent manner (Figs. 2A, B). Although other interactions that have been reported associated with Treg development and functions, including *TIGIT*[26], *CD226*[27], *4-1BB*[28] and *LGALS9*[29], were identified, they did not appear to be upregulated in both CD4+

**Fig. 1 | Polarized CD4+ T naïve-to-Treg development and activation is a tumor-specific characteristic in NPC. a** The UMAP plot of 189,750 T cells with 41 subtypes identified. **b** The developmental trajectories of CD4+ T cells. **c** The change of *FOXP3* expression through the pseudotime developmental process from CD4+ naïve T cells to three Treg subtypes, modeled by two-sided polynomial regression analysis with the 95% confidence band. **d** The normalized fractions of CD4+ naïve T cells and three Treg subtypes in NPC tissues (n = 36) versus in NPC peripheral blood (n = 10) or INP tissues (n = 4) (two-sided unpaired t-test). **e** The Treg suppressive score and T cell naïve score computed from GSE68799 (tumor n = 42, normal n = 3, two-sided unpaired t-test). **f** The progression-free survival for NPC patients from GSE102349, stratified by the Treg suppressive score (high n = 44, low n = 44, two-sided log-rank test). **g** Expression of Treg-specific and naïve signatures in CD4+ naïve T cells and Treg subtypes. **h** Major biological signaling and activities in Treg subtypes computed by GSVA. **i** Illustration of the in vitro co-culture systems of CD4+ naïve T cells, NPC cells, and NPE cells. **j–l** The fraction of total FOXP3+ Tregs, and FOXP3+/CTLA4 + activated Treg in the co-culture systems (**k**, transwell co-culture; **l**, direct co-culture) with C666 versus with NP460 or NP69 cells (n = 3, two-sided unpaired t test). **m** The normalized mRNA fold change of Treg lineage- and activation-specific markers in CD4+ naïve T cells co-cultured with C666 versus with NP460 or NP69 cells (n = 3, two-sided unpaired t test). **n** The change of immunosuppressive factors in the co-culture systems with C666 versus with NP460 or NP69 cells (n = 3, two-sided unpaired t test). The n number represents n biologically independent samples/experiments in each group. The data are presented as the mean ± SD (bar plots), median ± IQR (whiskers = 1.5 × IQR, box & whiskers plots), and KDE (violin plots).

naïve T cells and Treg subtypes, nor significantly strengthed in the eTregs in the NPC microenvironment (Supplementary Fig. 2a–d). Thus, CD70-CD27 signaling was identified as the only significant contact interaction occurring among CD4+ naïve T cells, Tregs, and NPC cells, thus with the potential to regulate de novo Treg development and suppressive activity in the TME.

CD70-CD27 signaling is a co-stimulatory pathway that has been shown to promote Treg proliferation[30,31], but the exact role and molecular mechanism of CD70-CD27 signaling in the development and activation of tumor-infiltrating Tregs remain inexplicit. Thus, we initially analyzed the spatial proximity among NPC cells, Tregs and CD70 expression based on Visium data, where the integrated NPC scRNA-seq data was used as a reference to infer tumor and Treg fractions via anchor-based deconvolution[32] and cell2location[33]. Spatially, Tregs were not only highly co-localized with NPC cells, but also were significantly correlated to CD70 expression (Figs. 2c, 2d, Supplementary Figs. 2e,f), indicating an active role of CD70+ NPC cells in modulating proximal Tregs. We further examined CD70 expression on NPC cells isolated from fresh endoscopic biopsies via flow cytometry, showing that approximately 60% of EPCAM+ tumor cells were CD70+, whereas only 10% CD70+ cells in EPCAM- infiltrating immune/stromal cells (Fig. 2e). Moreover, IHC and IF staining exhibited significantly higher CD70 expression and FOXP3+/CTLA4+ eTreg infiltration in primary NPC tissues than in non-malignant nasopharyngeal lymphoid tissues (Figs. 2f, g and Supplementary Fig. 2g).

Although minor fractions of effector T cells, B cells and dendritic cells transiently expressed CD70 upon activation[34], single-cell analysis confirmed that CD70 expression in the TME was highly restricted to NPC cells (Fig. 2h). We stratified patients in the integrated scRNA-seq cohort into CD70-high and CD70-low groups based on median CD70 expression, and showed that CD70-high patients harbored enriched fractions of rTregs and eTregs, but decreased fractions of nTregs and CD4+ naïve T cells (Fig. 2i). Such phenomenon was corroborated by qRT-PCR on primary NPC biopsies, showing higher expression of Treg lineage and activation markers, *FOXP3*, *IL2RA* and *CTLA4* in CD70-high patients (Supplementary Fig. 2h), who was stratified by the above flow cytometry result (Fig. 2e). Meanwhile, we found that the negative correlation between T cell cytotoxicity/proliferation and the Treg fraction was evident only in CD70-high patients (Fig. 2j), implying a feedback loop among CD70, Tregs and T-cell immunity.

In bulk RNA-seq cohorts, *CD70* expression was consistently upregulated in NPC patients and associated with inferior survival (Figs. 2k, l and Supplementary Fig. 2i). We further revealed that the CD70+ fraction in the C666 cells was 3-fold higher than that of NP460 and NP69 cells (Fig. 2m), thus inducing higher soluble CD27 (sCD27) cleaved from the cell surface of co-cultured CD4+ naïve T cells upon CD70-CD27 binding[35] (Fig. 2N). In the time-resolved transcriptome analysis of induced CD4 + naïve T cell-to-Treg differentiation, the computed Treg suppressive score increased proportionally to *FOXP3*, *CTLA4*, and *CD27* expression, whereas the naïve score exhibited an opposite trend during this process (Fig. 2o). In the bulk RNA-seq

cohort, we determined that the *CD70* expression was positively correlated with the suppressive score and negatively correlated with the naïve score (Fig. 2p). These results implied a Treg-polarizing and activating effect of CD70+ NPC cells, but it might be dually dependent on the CD70+ fraction in NPC cells and the CD27+ fraction in CD4+ naïve T cells and Tregs.

Since NPC is a unique EBV+ head and neck squamous cell carcinoma (HNSCC), we later explored whether the above-predicted feedback loop between CD70+ tumor cells and Tregs existed in other human papillomavirus+ (HPV+) HNCs. By analyzing two independent HNSCC scRNA-seq cohorts[36,37] (Supplementary Figs. 2m, n), we found that rTregs and eTregs were not significantly infiltrated in the HPV+ HNSCC microenvironment, compared to the normal tonsil microenvironment (Supplementary Fig. 2o). In addition, *CD70* expression in HNSCC cells was minimal and was not correlated to worse prognosis in HNC patients either (Supplementary Figs. 2p, q). These results supported that CD70-mediated Treg immunosuppression was an NPC-specific characteristic, and might not contribute to tumor progression or immunotherapy failure in HPV+ HNSCC patients.

## CD70 knockout in NPC cells alleviates immunosuppression by inhibiting Treg development and functionality

To elucidate the direct effect of CD70 on Treg development and activation, we treated CD4+ naïve T cells with active recombinant CD70 protein. The agonist treatment upregulated FOXP3 and CTLA4 expression in CD4+ naïve T cells and increased secretion of immunosuppressive factors through CD70-CD27 interaction (Figs. 3a–d and Supplementary Fig. 3a). Subsequently, we performed CRISPR-mediated knockout of CD70 (CD70-KO) in C666 cells, which did not affect PD-L1 expression or cell proliferation in C666 cells (Supplementary Figs. 2k,l). However, CD70-KO significantly reduced Treg development and activation from CD4+ naïve T cells by blocking CD27 interaction (Figs. 3e–g), and inhibited the Treg secretome profile of sCD27, IL-10, TGF-β, and adenosine (Figs. 3g, h and Supplementary Fig. 3a). The impaired Treg suppressive activity was further confirmed by flow cytometry and qRT-PCR analysis on additional Treg activation markers, including ICOS, *TNFRSF4*, 4-1BB (encoded by *TNFRSF9*), GITR (encoded by *TNFRSF18*), and *TIGIT* (Fig. 3i, Supplementary Figs. 3b,c). *TIGIT* was the only marker in the panel unaffected by CD70-KO, but this result was consistent with previous studies showing expression of *TIGIT* on Treg was independent of CD70-CD27 co-stimulation[38,39]. The Treg suppression assay further corroborated that CD70-KO functionally inhibited Treg suppressive activity on the proliferation of paired CD8+ T cells, echoing the above-mentioned results (Supplementary Fig. 3d).

Considering the impaired suppressive activity of Tregs might reinvigorate anti-tumor immunity mediated by CD8+ T cells, as suggested by the GSEA analysis on bulk RNA-seq data (Supplementary Fig. 3e), we established a tumor-peripheral blood mononuclear cell (PBMC) co-culture system which consisted of principally CD4+ and

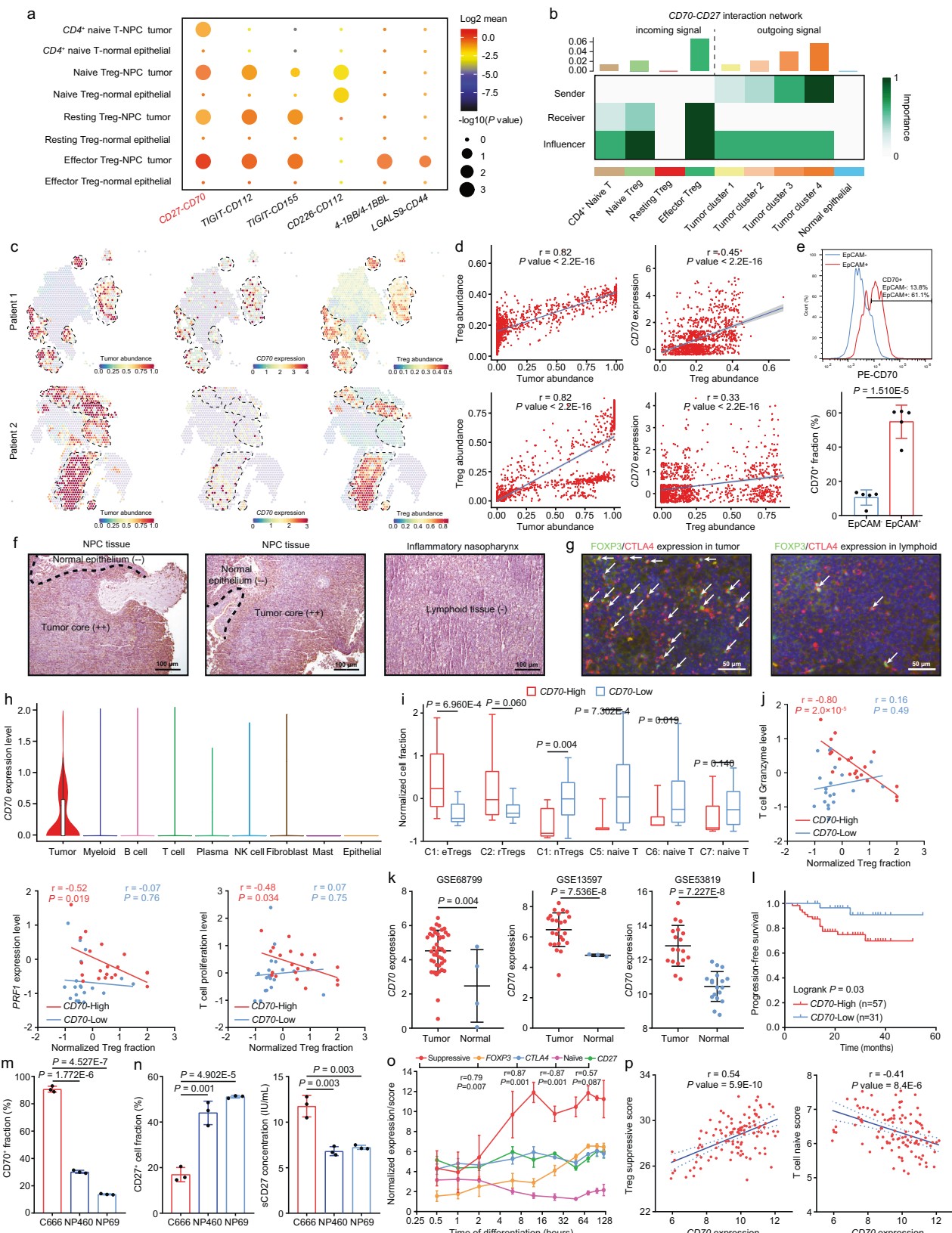

CD8+ T cells (Supplementary Figs. 3f, g). We found that CD70-KO in C666 cells greatly enhanced cytotoxicity-dependent tumor cell death (Figs. 3j–l), preferentially in a co-culture system similar to the physiological T-cell landscape in NPC patients, with a higher expression of CD27 on CD4+ T cells (Supplementary Figs. 3h–j). Higher tumor cell death might be directly due to loss of CD70 co-stimulation in co-

cultured CD4+ naïve T cells and Tregs induced by CD70-NC C666 cells rather than CD4+ and CD8+ T cells in the co-culture system (Supplementary Figs. 3k–m). In the CD70-KO system, CD8+/granzyme A+ and CD8+/perforin+ T cell fractions, secretion of TNF-α and IFN-γ, as well as CD8+ T cell proliferation were elevated, and expression of PD-1 and TIM-3 was diminished (Figs. 3m–q and Supplementary Fig. 3n),

**Fig. 2 | CD70 + NPC cells contribute to abundant and suppressive Treg infiltration by interacting with CD4+/CD27+ T cells. a** CellPhoneDB revealed enriched ligand-receptor pairs among *CD4*+ naïve T cells, Tregs, NPC, and NPE cells (two-sided permutation test). **b** CellChat revealed NPC cells as the sender of *CD70-CD27* signaling to *CD4*+ naïve T cells and Tregs. **c** Spatial co-localization of NPC cells, Tregs, and *CD70* (*n* = 1089 and 1331). **d** The Pearson correlation (two-sided) between NPC cells/*CD70* expression and Tregs (*n* = 1089 and 1331). **e** The CD70+ fractions in EPCAM + NPC cells and EPCAM- cells from NPC biopsies (*n* = 5, two-sided unpaired t test). IHC (**f**) and IF staining (**g**) showed CD70, FOXP3, and CTLA4 expression in NPC, normal epithelium and lymphoid tissues. **h** Expression of *CD70* across cell lineages. **i** Normalized fractions of Treg and CD4+ naïve subtypes in *CD70*-high (*n* = 20) and *CD70*-low patients (tumor *n* = 16, normal *n* = 4) from scRNA-seq data (two-sided unpaired t-test). **j** The Pearson correlation (two-sided) between the normalized Treg fraction and average expression of granzymes, *PRF1*, and T-cell proliferation markers in *CD70*-high (*n* = 20) and *CD70*-low patients (tumor *n* = 16,

normal *n* = 4) from scRNA-seq data. **k** Normalized *CD70* expression in three NPC RNA-seq cohorts. Left: GSE68799 (tumor *n* = 42, normal *n* = 3); Middle: GSE13597 (tumor *n* = 25, normal *n* = 3); Right: GSE53819 (tumor *n* = 18, normal *n* = 18) (two-sided unpaired t test). **l** The progression-free survival for NPC patients from GSE102349 (high *n* = 57, low *n* = 31, two-sided log-rank test). **m** The fraction of CD70+ cells in C666 cells versus in NP460 or NP69 cells (*n* = 3, two-sided unpaired t test). **n** Changes of surface and sCD27 in co-culture systems with C666 versus with NP460 or NP69 cells (*n* = 3, two-sided unpaired t test). **o** The change of Treg suppressive and T naïve scores and signatures during induced Treg differentiation (*n* = 3, two-sided Pearson correlation analysis). **p** The Pearson correlation (two-sided) between the Treg/T naïve scores and *CD70* expression in GSE102349 (*n* = 112). The n number represents *n* biologically independent spots/samples/experiments in each group. The data are presented as the mean ± SD (bar plots), median ± IQR (whiskers = 1.5 × IQR, box & whiskers plots), and KDE (violin plots).

---

suggesting that impaired Treg suppressive activity could re-activate anti-tumor CD8+ T cells.

To more comprehensively demonstrate the immunosuppressive effect of CD70-CD27 interaction in NPC-infiltrating T cells, we designed an autologous co-culture system between CD19+CD70-NC/CD19+ CD70-KO C666 cells with anti-CD19 CAR-T cells (Figures S3O,P). The autologous co-culture system enabled antigen-specific T-cell activation in both CD4+ and CD8+ T cells and antigen-specific killing to the CD19+ CD70-NC and CD19+ CD70-KO C666 cells. Compared to the results from the allogenic co-culture system, CD70-KO in C666 cells in the autologous co-culture system consistently inhibited Treg polarization, activation, and suppressive activity, leading to enhanced antigen-specific T-cell killing and cytotoxicity (Supplementary Figs. 3q–s). Furthermore, we established immuno-resistant C666 cells by sequential PBMC challenge to the parental C666 cells to mimic progressive immune escape during NPC progression. We found a consistent upregulation of *CD70* expression in the immuno-resistant cells (Supplementary Fig. 3t), validating the active role of CD70+ NPC cells in facilitating immune evasion.

Additionally, the lack of murine NPC cell lines, spontaneous mouse models, and the inability of EBV infection in mice constantly hindered the translational research on NPC immunology and immunotherapy. Thus, we established a PBMC-engrafted humanized NSG mouse model to study the in vivo effect of CD70-KO C666 cells on the development and suppressive activity of infiltrating Tregs. Consistent with the in vitro results, CD70-KO did not alter the in vivo tumorigenicity of NPC in immunodeficient NSG mice (Fig. 3r), but led to significant tumor shrink in humanized NSG mice (Fig. 3s), further confirming an inhibitory role of CD70 on anti-tumor immunity. While H&E staining exhibited no significance difference between total immune infiltration in CD70-NC and CD70-KO tumors (Supplementary Fig. 4a), CD70-KO tumors harbored a decreased fraction of FOXP3+ Tregs with impaired activation, thus lowering IL-10 and TGF-β concentrations in the TME (Figs. 3t–v, Supplementary Figs. 4b,c). Thus, CD70-KO tumors revigorated the effector profile of CD8+ T cells by promoting cytotoxic cytokine release and preventing T-cell exhaustion (Figs. 3t, u, Supplementary Figs. 4b–e). These results implied that therapeutic inhibition of CD70 in NPC patients might be an effective strategy to overcome Treg immunosuppression which in turn activates T-cell immunity.

### Cusatuzumab enhances antitumor immunity and acts synergistically with anti-PD-1 treatment in patient-derived models

Considering the high CD27 expression on T cells, NK cells, and B cells that might lead to off-target effects when using CD27 blockade[40], CD70 has emerged as a more effective and safe target since it is not expressed on normal tissues nor hematopoietic lineages during homeostasis[41]. Cusatuzumab is a human αCD70 monoclonal antibody that can effectively block CD70-CD27 interaction and its downstream

signaling. So far, cusatuzumab is being evaluated in patients with acute myeloid leukemia (AML) in multiple clinical trials (NCT04023526 and NCT04150887), and has shown a potent killing efficacy to CD70 + leukemia stem cells with manageable adverse events[42]. However, the efficacy of cusatuzumab has not been comprehensively described in solid tumors in pre-clinical settings. By treating C666 cells with 5 μg/mL cusatuzumab, we found Treg development, activation, secretion of immunosuppressive factors, and the ability to suppress CD8+ T cell proliferation in the CD4+ naïve T cell co-culture system were inhibited to a comparable extent as CD70-KO (Figs. 4a–d and Supplementary Fig. 4f). Thus, cusatuzumab treatment induced higher tumor cell death by promoting proliferation and cytotoxicity of CD8+ T cells in the PBMC co-culture system (Figs. 4e–i, Supplementary Figs. 4g,h). We also evaluated the inhibitory role of a CD70-blocking-only antibody on immunosuppression in the PBMC co-culture system. CD70 blocking in C666 cells showed comparable tumor-killing and cytotoxicity-enhancing effects with cusatuzumab treatment (Supplementary Figs. 4i–l), further confirming CD70 inhibition is effective in inducing stronger anti-tumor immunity. Next, the immune-activating effect of cusatuzumab was further validated in the co-culture system between patient-derived primary NPC cells and autologous PBMCs. Compared to IgG treatment, cusatuzumab treatment inhibited the suppressive activity of Tregs, which in turn activated the effector function of CD8+ T cells, eventually resulting in an enhanced tumor-killing efficacy (Supplementary Figs. 4m–p).

In this study, cusatuzumab was shown to enhance T-cell immunity by alleviating Treg development and suppressive activity in NPC. Alternatively, the anti-PD-1 treatment directly unleashes proliferation and activation of CD8+/PD-1+ T cells by preventing T-cell exhaustion. Therefore, a synergistic effect that could enhance antitumor immunity might be achieved by combining anti-CD70 and anti-PD-1 treatments together. To investigate this potential synergistic effect, we first used the TIDE method to predict ICB responsiveness and T-cell signatures in *CD70*-high and *CD70*-low NPC patients[43]. *CD70*-high NPC patients were predicted to be more resistant to ICB blockade, showing lower T-cell cytotoxicity and inflammation but higher T-cell exclusion (Supplementary Fig. 5a). Despite NPC, CD70 is also preferentially expressed in other solid tumors, such as melanoma[44]. Some studies in melanoma have generated TME transcriptome profiles from responders and non-responders receiving ICB[45,46], and computational analysis of these profiles illustrated that *CD70* expression was associated with T-cell dysfunction and inferior ICB outcomes (Supplementary Figs. 5b and 5c). To verify the pro-tumor effect of CD70 in melanoma, we performed CD70-KO in a murine melanoma cell line: B16-F10 (Supplementary Fig. 5d), and orthotopically injected CD70-NC and CD70-KO B16-F10 cells in C57BL/6J mice. CD70-KO led to significant melanoma shrink, but we also observed a tumor-specific response to CD70-KO, possibly due to the mouse-specific TME (Supplementary Figs. 5e and 5f).

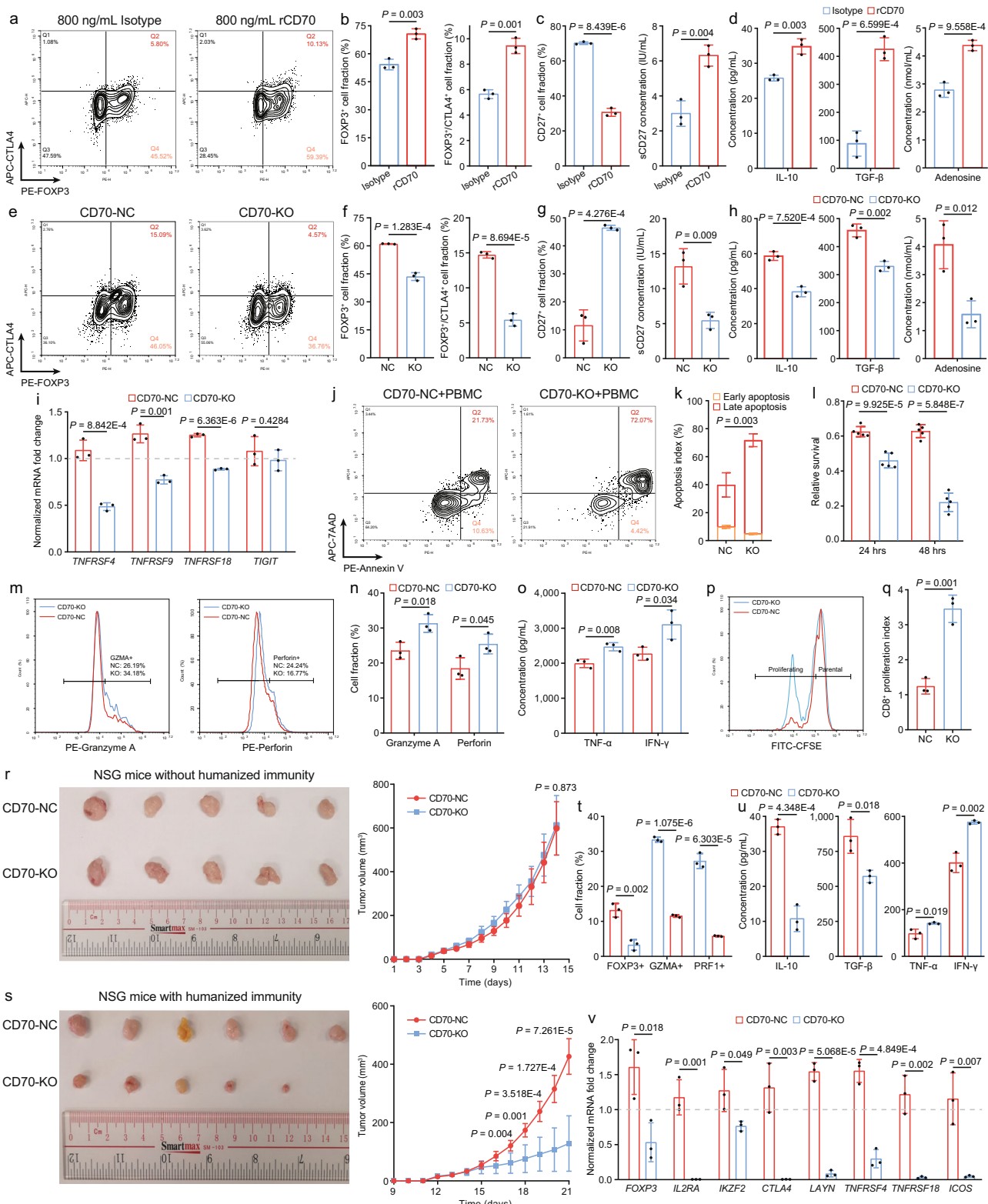

Experimentally, we evaluated the efficacy of CD70+ PD-1 blockade in NPC organoids, which mimicked the physiological condition of primary NPC tumors. We established NPC organoids from an EBV+ NPC patient-derived xenograft (Xeno76) established in 2018[47], and evaluated CD70 and PD-L1 expression levels in Xeno76 accordingly (Fig. 4j). In the organoid-PBMC co-culture system, both anti-CD70 monotherapy and anti-PD-1 monotherapy significantly increased tumor cell death, but combination therapy exhibited greater tumor-

killing efficacy in terms of the organoid number, size, and viability (Figs. 4k–n). We further compared the in vivo efficacy of the combi-nation therapy to monotherapy in Xeno76-transplanted PBMC-engrafted humanized NSG mice. Consistently, the anti-CD70 + anti-PD-1 combination therapy showed the highest efficacy in inhibiting patient-derived xenograft (PDX) growth (Figs. 4o, p). Meanwhile, the fractions of tumor-infiltrating Tregs and CD8+ cytotoxic T cells were significantly decreased and increased upon receiving the combination

**Fig. 3 | Genetic ablation of CD70 in NPC cells reverts Treg suppressive activity and enhances CD8+ T cell function in the TME. a, b** Immunophenotyping of CD4+ naïve T cells treated with IgG antibody and recombinant CD70 antibody (rCD70, $n = 3$, two-sided unpaired t test). **c** Changes of surface and sCD27 in the co-cultured systems treated with IgG antibody and rCD70 ($n = 3$, two-sided unpaired t test). **d** The change of immunosuppressive factors in the IgG-treated and rCD70-treated co-culture systems ($n = 3$, two-sided unpaired t test). **e, f** Immunophenotyping of CD4+ naïve T cells co-cultured with CD70-NC and CD70-KO C666 cells ($n = 3$, two-sided unpaired t test). **g** Changes of surface and sCD27 in the co-cultured systems with CD70-NC and CD70-KO C666 cells ($n = 3$, two-sided unpaired t test). **h** The change of immunosuppressive factors in the CD70-NC and CD70-KO co-culture systems ($n = 3$, two-sided unpaired t test). **i** The normalized mRNA fold change of Treg activation-specific markers in CD4+ naïve T cells co-cultured with CD70-NC

and CD70-KO C666 cells ($n = 3$, two-sided unpaired t test). T-cell cytotoxicity measured in the CD70-NC and CD70-KO (**j** and **k**, flow cytometry, $n = 3$; **l**, XTT assay, $n = 5$) PBMC co-culture systems (two-sided unpaired t test). The change of CD8+ T-cell cytotoxicity markers (**m** and **n**, flow cytometry), cytokines (**o**, ELISA) and proliferation (**p** and **q**, CFSE) in the CD70-NC and CD70-KO PBMC co-culture systems ($n = 3$, two-sided unpaired t test). (**r, s**) The CD70-NC and CD70-KO tumor growth (**r**, $n = 5$ for NSG mice; **s**, $n = 6$ for humanized mice) in immunodeficient NSG and PBMC-engrafted humanized mice (two-sided unpaired t test). The change of immunosuppressive and cytotoxic T cell subtypes (**t**, flow cytometry), cytokines (**u**, ELISA), and markers (**v**, qRT-PCR) in CD70-NC and CD70-KO tumors ($n = 3$, two-sided unpaired t-test). The n number represents $n$ biologically independent samples/experiments in each group. The data are presented as the mean ± SD (bar plots).

therapy, respectively (Fig. 4q and Supplementary Fig. 5g), suggesting that combining anti-CD70 and anti-PD-1 treatments achieved a synergistic effect that stimulated T-cell immunity by overcoming Treg-mediated immunosuppression.

## CD70-CD27 signaling contributes to lipid metabolic reprogramming in Tregs

Tregs are a resilient T-cell subtype with strong tissue adaptiveness and molecular flexibility that facilitates their survival, homeostasis, and metabolism in different ecosystems, including periphery, lymph nodes, thymus and tumors[48-50]. As we demonstrated that CD70-KO and inhibition in NPC cells enhanced anti-tumor immunity by limiting Treg activities, we further investigated how CD70-CD27 signaling influenced downstream homeostasis of Tregs, particularly in the TME of NPC. Hence, we applied additional 3′ scRNA-seq to PBMCs co-cultured with CD70-NC and CD70-KO C666 cells (Fig. 5a). Consistent with our in vitro and in vivo results, CD70-KO reduced the abundances of nTregs, rTregs and eTregs, and expanded pro-inflammatory subtypes, including CD8+cytotoxic T cells, CD4+T$_{FH}$ cells and CD4+ naïve T cells (Fig. 5b).

Moreover, Tregs in the PBMCs co-cultured with CD70-KO C666 cells (KO-Tregs) exhibited lower expression of lineage-specific (*FOXP3*, *IL2RA* and *SOX4*), activation (*CTLA4* and *LAYN*) and co-stimulatory (*TNFRSF4* and *TNFRSF9*) markers, but higher expression of naïve (*SELL*) signature markers (Fig. 5c). Particularly, some co-cultured eTreg transitioned into a fragile state which has been previously characterized by retention of FOXP3 expression but aberrant IFN-γ production[51] (Supplementary Figs. 6a,b). However, upregulation of PD-1 in CD70-co-cultured Tregs protected these cells from IFN-induced fragility while maintaining functional homeostasis (Fig. 5c and Supplementary Fig. 6c). Thus, the synergy observed upon inclusion of PD-1 blockade in the anti-CD70 treatment might also partially reduce immunosuppression by inducing Treg fragility[52]. As for CD8+ cytotoxic T cells in the CD70-KO group, expression of proliferation (*MKI67*), cytotoxicity (*GZMA* and *LTB*) and activation (*TNFRSF1B* and *CD28*) markers was elevated, whereas naïve (*TCF7* and *CCR7*) and exhaustion (*LAYN*, *LAG3*, and *HAVCR2*) markers were decreased, further corroborating that disrupted suppressive activity of Tregs promoted proliferation, activation, and alleviated exhaustion in CD8+ T cells (Fig. 5d). CD70-mediated Treg immunosuppression also hampered the inflammatory response of CD8+ cytotoxic T cells, thus lowering their anti-tumor effect (Supplementary Fig. 6d).

To further delineate the downstream mechanisms involved with CD70-CD27 signaling in Tregs, we performed GSEA between CD70-induced Tregs and KO-Tregs. CD70+ C666 cells potently reprogramed metabolic profiles of co-cultured Tregs via enhancing oxidative phosphorylation (OXPHOS) driven by a complex lipid signaling network (Fig. 5e and Supplementary Fig. 6e). Tregs frequently require robust metabolism to sustain their development and suppressive activity in the TME. Particularly, Tregs possess high energy-driven plasticity, and thus can utilize different metabolic resources, including

lactic acids, glucose, and fatty acids (FAs), to sustain functional homeostasis in the metabolite-deficient milieu[20,53-55]. Pathway analysis revealed that fatty acid metabolism, cholesterol homeostasis, and mitochondrial functions were top enriched signatures in CD70-induced Tregs, which have been recently reported to facilitate Treg lineage determination, functional fitness, and immunosuppression (Fig. 5e). For example, increased OXPHOS driven by fatty acid oxidation was a hallmark of Treg development and suppressive activity in the TME[53]. Meanwhile, upregulated detoxification and redox homeostasis protected CD70-induced Tregs from a high level of reactive oxygen species produced by OXPHOS and ATP production while maintaining mitochondrial integrity (Fig. 5f). In contrast, KO-Tregs exhibited impaired functional homeostasis due to enhanced lipid efflux and clearance as well as loss of mitochondrial integrity (Fig. 5f).

## Mitochondrial integrity, cholesterol homeostasis, and fatty acid oxidation collectively contribute to the functional specialization of Tregs

To gain mechanistic insights into how the CD70-induced complex lipid network strengthened the functional specialization and homeostasis of Tregs in the NPC microenvironment, we evaluated the intracellular changes of total lipid and key lipid constituents, such as cholesterol and fatty acids, in co-cultured CD4+ T cells with CD70-NC and CD70-KO C666 cells. First, genetic ablation and therapeutic inhibition of CD70 in NPC cells significantly reduced the total intracellular lipid and cholesterol in co-cultured Tregs (Figs. 6a, b). Second, there was an increase of cell-free cholesterol from FOXP3- non-Tregs, FOXP3+/CTLA4- rTregs to FOXP3+/CTLA4+ eTregs, which conveyed that intracellular cholesterol accumulation was associated with Treg development and suppressive activity (Fig. 6c). In the integrated NPC scRNA-seq cohort, we validated eTregs had more robust cholesterol metabolism and biosynthesis, but lower cholesterol export (Supplementary Fig. 7a).

Contemplating that lipid signaling was a highly complex network involving subtle changes in various metabolites, we performed mass spectrometry-based metabolomics to quantify intracellular metabolic alterations of polar molecules and fatty acids in co-cultured CD4+ T cells (Fig. 6c). Mass spectrometry analysis revealed a consistent decrease of intracellular cholesterol and adenosine in the CD70-KO group with lower CD39 expression (Fig. 6d, Supplementary Figs. 7b,c). Meanwhile, increased glycerol could protect Tregs from lipid-accumulated toxicity, which damaged FOXP3 stability[56] (Fig. 6d). The OXPHOS-derived metabolites, including succinate, 2-hydroxyglutarate (2-HG) and malate, which have previously been reported associated with impaired Treg development and suppressive activity through epigenetic repression of PD-1[57], were actively metabolized (Fig. 6d). Whereas citrate, aspartate and glutamine remained unchanged in CD4+ T cells between CD70-NC and CD70-KO groups. For the fatty acid panel, the total fatty acid content in the CD70-KO group was markedly decreased (Fig. 6d). The concentrations of almost all intracellular fatty acid derivatives from C4 to C26 were reduced

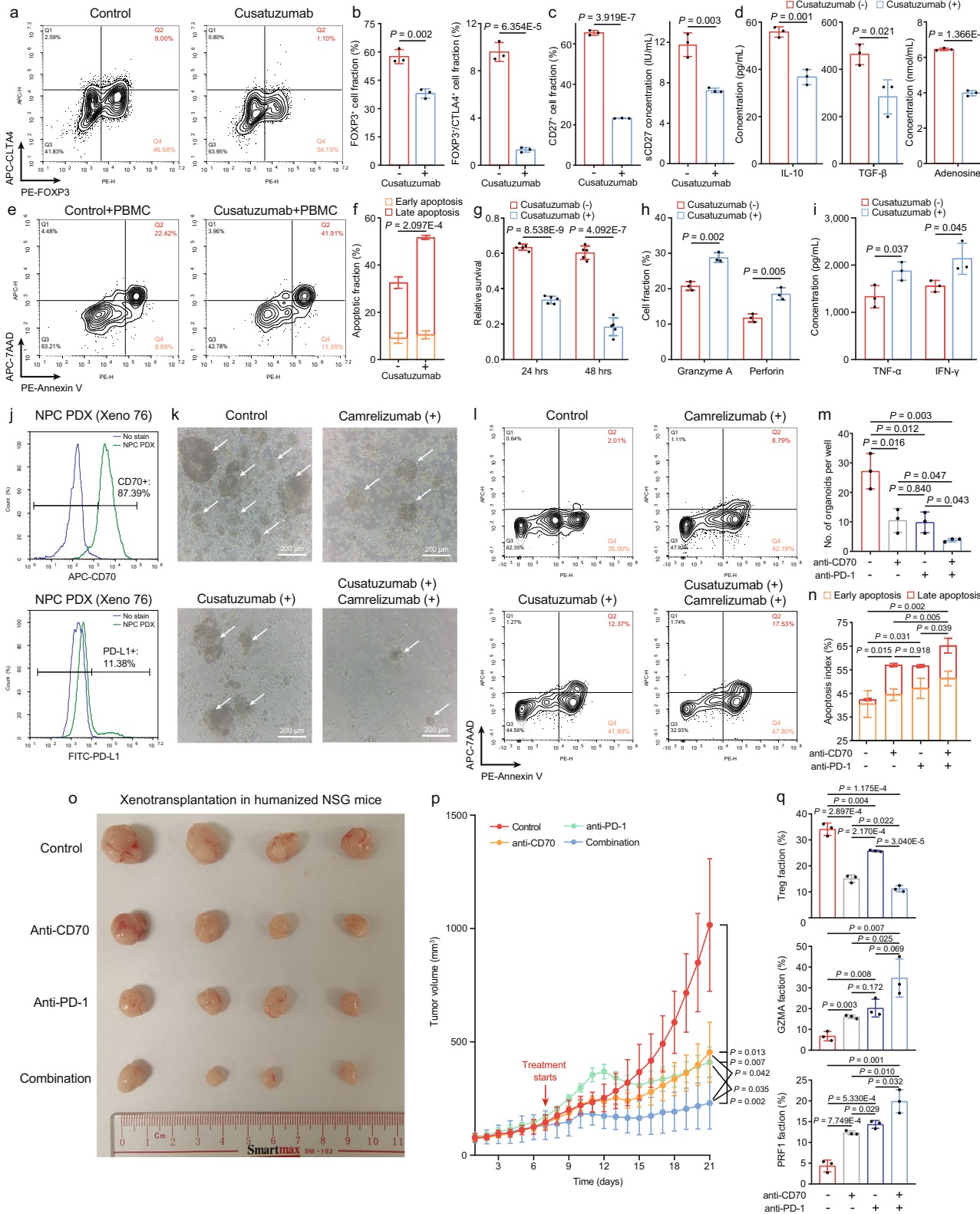

(Supplementary Table 2), indicating that block of CD70-CD27 signaling in Tregs might undermine de novo fatty acid uptake, synthesis and oxidation.

Based on the observed metabolic adaptation of Tregs in CD70/CD27-mediated lipid signaling, we proposed that mitochondrial integrity, cholesterol homeostasis, and fatty acid metabolism collectively contributed to metabolic advantages and functional specialization of Tregs via CD70-CD27 signaling in the NPC microenvironment.

We speculated that CD70 served as a master metabolic switch that reprogramed lipid metabolism in CD4+ naïve T cells and Tregs to meet the metabolic demands committed these cells to Treg development, homeostasis and activation in a hostile milieu. To validate our hypothesis, we first showed that KO-Tregs had fewer and unhealthier mitochondria examined by electron microscopy (Fig. 6e), resulting in decreased ATP generation, OXPHOS and mitochondrial potential shown by the Seahorse assay, and JC-1 staining (Figs. 6f–h).

**Fig. 4 | Therapeutic inhibition of CD70 promotes anti-tumor immunity and anti-PD-1 efficacy. a**, **b** Immunophenotyping of CD4+ naïve T cells treated with IgG antibody and cusatuzumab (*n* = 3, two-sided unpaired t test). **c** The change of surface and sCD27 in the co-cultured systems treated with IgG antibody and cusatuzumab (*n* = 3, two-sided unpaired t test). **d** The change of immunosuppressive factors in the co-cultured systems treated with IgG antibody and cusatuzumab (*n* = 3, two-sided unpaired t test). T-cell cytotoxicity measured in the IgG-treated and cusatuzumab-treated (**e** and **f**, flow cytometry, *n* = 3; **g**, XTT assay, *n* = 5) C666/ PBMC co-culture systems (two-sided unpaired t test). The change of CD8 + T-cell cytotoxicity markers (**h**, flow cytometry) and cytokines (**i**, ELISA) in IgG-treated and cusatuzumab-treated C666/PBMC co-culture systems (*n* = 3, two-sided unpaired t test). **j** Expression of CD70 and PD-L1 in NPC PDX Xeno76. T-cell cytotoxicity measured in the IgG-treated and cusatuzumab-treated (**k** and **m**, microscopy; **l** and **n**, flow cytometry) PDX/PBMC co-culture systems (*n* = 3, two-sided unpaired t test). **o**, **p** Tumor growth in PDX-bearing humanized mice treated with different regimens (*n* = 4 for each treatment group), measured from day 1 post intravenous PBMC injection (two-sided unpaired t test). **q** The change of Tregs and cytotoxic T cells in tumors compared between every two treatment groups (*n* = 3 per treatment group, two-sided unpaired t test). The *n* number represents n biologically independent samples/experiments in each group. The data are presented as the mean ± SD (bar plots).

Furthermore, we evaluated the changes of 46 representative genes involved in lipid synthesis, transport, and oxidation in Tregs, in which more than 70% of the selected signatures were significantly influenced by CD70-CD27 signaling (Fig. 6i).

In light of elevated intracellular cholesterol in CD70-induced Tregs, we first inspected the change in *LDLR*, which was involved in regulating cholesterol uptake[58]. Interestingly, expression of *LDLR* did not increase in CD70-induced Tregs, whereas cholesterol efflux-associated genes, such as *SCARB1* and *SDC1*, were found downregulated, suggesting the cholesterol accumulation was caused by inhibited cholesterol export. Moreover, the key regulators and enzymes involved with cholesterol metabolism and Treg homeostasis via the mevalonate pathway, including *SREBF1*, *SREBF2*, *HMGCS1*, *SOAT1*, *SOAT2*, *FNTB*, *PGGT1B*, *GGPS1*, *RAC2* and *PD-1*, which fostered Treg development and maintenance[59,60], were also found augmented by CD70-CD27 signaling (Fig. 6i).

Intracellular fatty acid accumulation in Tregs caused by fatty acid uptake and de novo synthesis was actively involved in IL-2-dependent proliferation and TCR-dependent activation[60,61]. CD70-KO had impaired fatty acid uptake and oxidation in co-cultured CD4+ T cells, as determined by BODIPY $C_{12}$ tracing and levels of three enzymes (ACADVL, ACADM, and HADHA) in the FAO pathway (Supplementary Figs. 7d and 7e). Increased *ACACA*, instead of *FASN*, in CD70-co-cultured CD4+ naïve T cells, might also contribute to intracellular fatty acid accumulation via de novo FA synthesis, which upregulated TCR-dependent Treg activation and maturation markers *TNFRSF18* and *CD44* (Fig. 6i). As the above metabolic assays demonstrated, the accumulated fatty acid pool in CD70-induced Tregs provided sufficient resources for FAO via mitochondrial OXPHOS, which sustainably generated ATP sufficient to meet the high energy demands of Treg functionalization and survival in the TME[57].

Mitochondrial complex III has been previously demonstrated to be essential for Treg suppressive activity and enhanced FAO-driven OXPHOS[62]. *UQCRFS1*, *UQCRQ*, and *CYC1*, associated with the integrity and function of mitochondrial complex III, were upregulated in Tregs activated by CD70-CD27 signaling, orchestrating with the electron microscope images (Figs. 6e, i). We also found upregulation of key regulators and enzymes in the TCA cycle and electron transport chain (ETC), including *NR4A1*, *NR4A2*, *MDH1*, *MDH2*, *ECHS1* and *ECH1* (Fig. 6i)[63,64]. Other transcription factors, including *STAT5A*, and cAMP-inducible factors, such as *CREB1* and *CEBPB*, which were related to metabolic fitness of Tregs[65,66], might ensure functional specialization of Tregs in the NPC microenvironment (Fig. 6i). However, several other vital lipid metabolism-associated genes, including *SQLE*, *SCAP*, *HMGCR*, *ATF4*, *CCDC5B*, *LAGLS3*, *CTNB1*, and *EGR1*, were not differentially expressed (Fig. 6i), indicating an alternative and CD70-independent lipid signaling in Tregs within a tumor-specific niche.

We further constructed a lipid signaling module comprising 15 representative signature genes selected from the qPCR panel based on multivariate linear regression (Supplementary Fig. 7f), to quantitatively evaluate the lipid signaling activity in Tregs and other CD4+ T cells in the single-cell and bulk transcriptome datasets. First, genetic ablation of CD70 in NPC cells significantly impaired lipid signaling in the co-cultured Tregs (Fig. 6j). Meanwhile, in the integrated NPC single-cell cohort, eTregs possessed the highest lipid signaling activity, exceeding that of rTregs, nTregs and other CD4+ subtypes, including $T_{FH}$ cells, $T_{CM}$ cells and naïve cells (Fig. 6k). In the time-resolved Treg differentiation model, the lipid signaling activity was positively correlated with the Treg suppressive score and CD25 expression during naïve T-cell-to-Treg differentiation and negatively correlated with the naïve score and IL7R expression (Fig. 6l). In the bulk RNA-seq cohort, the lipid signaling activity was highly correlated to the suppressive activity of Tregs and inferior prognosis in NPC patients (Figs. 6m, n). Furthermore, the correlations between the 15 representative signature genes and *CD70*, *CD27*, *FOXP3*, *CTLA4*, and *CD25* were independently validated in transcriptome data for 337 whole blood samples from GTEx (Fig. 6o), suggesting that the lipid module was consistently related to Treg activities across individuals. In summary, these findings demonstrated intimate cooperation between lipid metabolism reprogramming and CD70-CD27 signaling in the lineage determination, homeostasis, and the suppressive activity of NPC-infiltrating Tregs.

To validate the importance of lipid metabolism in CD70-induced Treg development and activation in the NPC microenvironment, we assessed functional and metabolic changes in CD70-NC and CD70-KO co-culture with a lipid-depleted microenvironment. In the lipid-depleted co-culture system, CD70+ NPC cells could no longer induce a higher fraction of total and activated Tregs differentiated from CD4+ naïve T cells, nor induce stronger suppression on CD8+ T cell proliferation (Figs. 6p, q and Supplementary Fig. 7g). The immunosuppressive secretome profiles between CD4+ T cells co-cultured with CD70-NC and CD70-KO C666 cells in a lipid-depleted system were not distinctive either (Fig. 6r). Therefore, lipid depletion also resulted in the incapability of CD70-CD27 interaction in regulating Treg immunosuppression in the NPC-PBMC co-culture system, causing elevated anti-tumor immunity in both CD70-NC and CD70-KO groups (Fig. 6s). Moreover, the total intracellular lipid content in NC- and KO-Tregs remained comparable, due to the lack of free lipids that could be taken up and synthesized (Fig. 6t). Metabolic assays further exhibited that ATP generation, OXPHOS, mitochondrial integrity, and fatty acid oxidation in Tregs were significantly impaired upon lipid depletion, even with active CD70-CD27 signaling (Figs. 6u–x). These results successfully demonstrated that lipid metabolism was indispensable for CD70-CD27 interaction to enhance development, homeostasis, and the suppressive activity of NPC-infiltrating Tregs.

## CD70 is transcriptionally upregulated in NPC cells by NFKB2 via the EBV-dependent epigenetic modification

Since CD70 overexpression in NPC tissues was frequently observed in clinical practice, we wanted to explore the upstream mechanism contributing to CD70-mediated immune evasion. The single-cell analysis found the transcriptome level of *CD70* significantly overexpressed in EBV+ NPC cells (Fig. 7a). We corroborated that the CD70+ fraction was higher in EBV+ NPC43 cells, compared to the EBV- counterparts (Fig. 7a). EBV infection in NPC causes genomic instability and thus epigenetically stimulates the transcription of oncogenes to facilitate tumor progression and immune escape[67,68]. The chromosomal

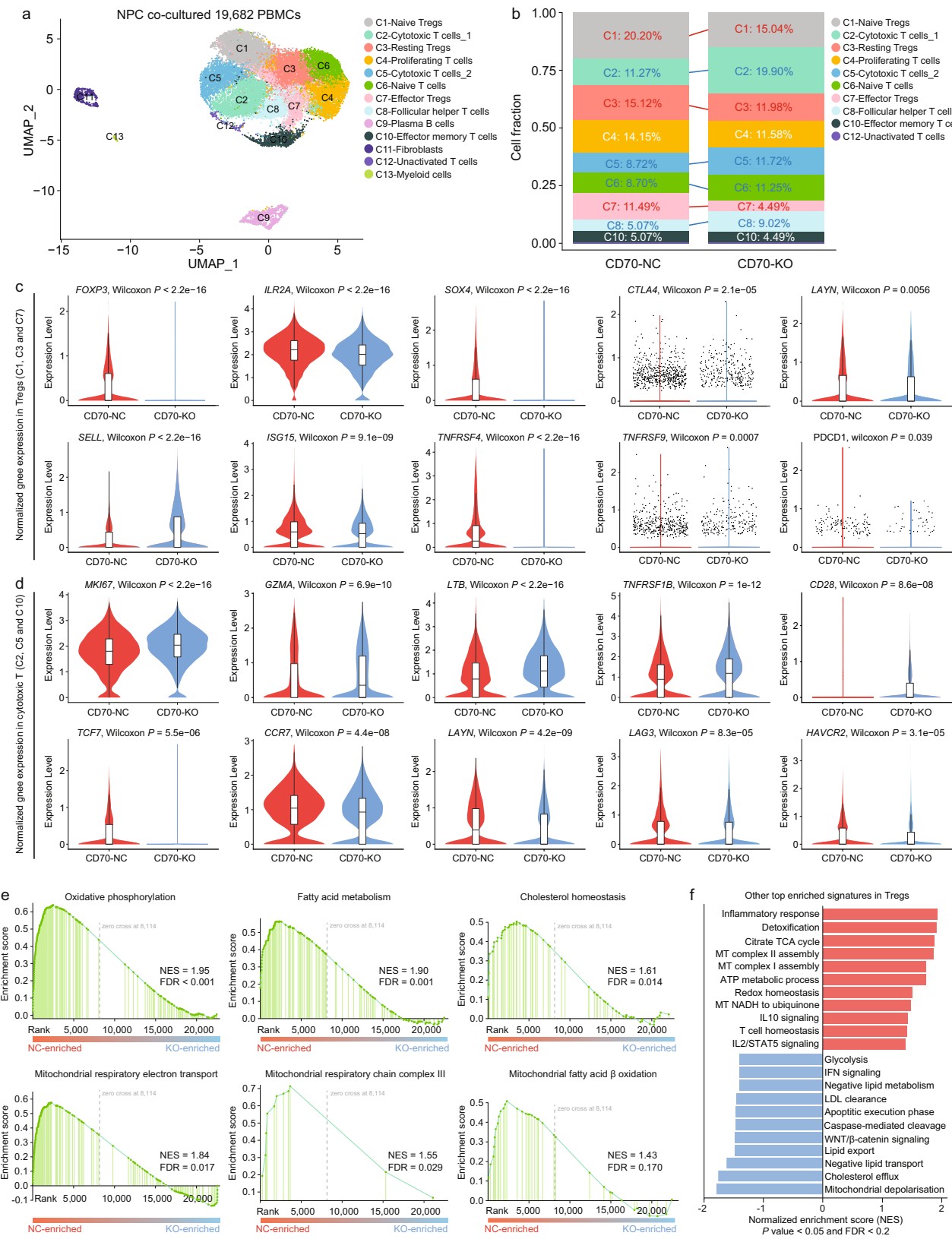

locus of the *CD70* gene on chromosome 19 has been reported to be frequently amplified in NPC[69,70]. Thus, to delineate whether CD70 overexpression in NPC was epigenetically induced by EBV-mediated chromatin accessibility alterations, we performed ATAC-seq on EBV+ NPC cells, EBV- NPC cells, and NPE cells. The chromatin accessibility at the *CD70* promoter region was enhanced in EBV+ NPC cells compared

to the EBV- and normal counterparts, enabling a more effective transcription factor binding to promote *CD70* mRNA transcription (Fig. 7b). Based on the ATAC analysis, we identified the top 15 transcription factors with the highest transcriptional activity in EBV+ NPC cells, that could potentially bind to the *CD70* promoter region with higher accessibility (Fig. 7c). Among the 15 transcription factors, we

**Fig. 5 | Single-cell analysis reveals that CD70-CD27 signaling enhances lipid-driven OXPHOS in Tregs. a** The UMAP plot of 19,682 PBMCs with 13 subtypes identified. **b** The relative abundance of each T cell subtype in the PBMC co-culture system with CD70-NC and CD70-KO C666 cells. **c** Expression of Treg-specific and naïve signatures in Tregs (C1: nTregs + C3: rTregs + C7: eTregs) co-cultured with CD70-NC and CD70-KO C666 cells (*n* = 3731 and 3264, two-sided Wilcoxon signed-rank test). **d** Expression of cytotoxic, activation, naïve, and exhausted signatures in CD8+ cytotoxic T cells (C2, C5, and C10) co-cultured with CD70-NC and CD70-KO C666 cells (*n* = 1997 and 3741, two-sided Wilcoxon signed-rank test). **e** GSEA revealed enriched OXPHOS and lipid metabolism-associated signaling in NC-Tregs, compared to KO-Tregs (GSEA-computed false discovery rate). **f** Other top enriched hallmark, KEGG, REACTOME, and GP:BP activities (NES > 1.4/NES < −1.4, *P* value <0.05 and FDR < 0.2) in NC-Tregs and KO-Tregs (two-sided permutation test, the false discovery rate was computed by GSEA and used to avoid false positive pathways). The n number represents n biologically independent cells in each group. The data are presented as the median ± IQR (whiskers = 1.5 × IQR, box & whiskers plots) and KDE (violin plots).

characterized that *NFKB2* was the one most correlated to *CD70* expression validated in two NPC RNA-seq cohorts and Visium spatial data, and had the highest binding score to the *CD70* promoter region predicted by JASPER (Figs. 7d, e and Supplementary Data 2). In EBV-infected lymphocytes, *NFKB2* was also found highly correlated to *CD70*, indicating a consistent EBV-modulated epigenetics in both immune and epithelial cells (Fig. 7f).

NFKB2 is a master regulator of the NF-κB pathway, which is widely considered as the homogeneous driver in NPC initiation and progression. *NFKB2* was frequently overexpressed in EBV+ NPC cells, instead of EBV- tumor cells, NPE cells, stromal cells, and immune cells in the TME (Fig. 7g), and has been previously shown to promote tumor growth and survival[71]. Nevertheless, its regulatory role on immune evasion of NPC and the molecular linkage to CD70 upregulation has not been comprehensively elucidated. Thus, to delineate the regulatory role of NFKB2 on *CD70* transcription, we identified a unique binding motif of NFKB2 on the *CD70* promoter region with enhanced accessibility caused by EBV infection (Fig. 7b). Thus, we designed a point-mutated *CD70* binding motif with luciferase and demonstrated that NFKB2 could specifically and effectively bind to this site to enhance CD70 transcription (Fig. 7h). To further validate NFKB2 as a transcriptional enhancer for CD70, we performed *NFKB2* knockdown and inhibition in C666 cells, showing consistent decreases in both *CD70* mRNA and CD70 protein (Figs. 7i to 7k). Loss of NFKB2 in C666 cells thus sensitized them to immune attack in the PBMC co-culture system (Figs. 7l, m). These data unveiled the EBV-NFKB2-CD70 axis as a vital mechanism for NPC cells to escape immune surveillance by upregulating Treg immunosuppression that lowered T-cell immunity.

## Discussion

Given the limited clinical impacts of ICB therapy in NPC patients, developing targets that best represent the TME characteristics remains essential for improving the clinical efficacy of precision immunotherapy. To address this need, we established a large-scale and multicentral NPC clinical cohort containing public single-cell information from 50 primary tissues and peripheral blood samples. Our analysis demonstrated that abundant and activated Tregs were the dominant source of immunosuppression in the NPC microenvironment. We proposed that most tumor-infiltrating Tregs were differentiated from in situ and peripheral CD4+ naïve T cells and activated via CD70-CD27 signaling. We identified and characterized that CD70+ NPC cells were the primary contributor to such Treg-mediated immunosuppression. Genetic ablation and therapeutic inhibition of CD70 in NPC cells undermined Treg abundance and suppressive activity, thus enhancing anti-tumor immunity mediated by CD8+ T cells and the efficacy of anti-PD-1 treatment (Fig. 7n), as comprehensively validated in cell lines, primary culture, organoids, PDX, and humanized mouse models.

Therapeutically, CD70-targeted immunotherapy has been primarily evaluated in patients with hematologic malignancies, such as acute myeloid leukemia (AML) and B cell lymphoma[72,73]. However, the efficacy of anti-CD70 treatment in solid tumors has not been comprehensively described. Here, we exhibited that anti-CD70 monotherapy and anti-CD70+ anti-PD-1 combination therapy overcame Treg-mediated immunosuppression and elevated the effector function of CD8+ T cells in NPC, providing sufficient experimental evidence of how anti-CD70 treatment exerted its tumor-killing effect in a Treg-enriched TME. We also unveiled that CD70-targeted cusatuzumab treatment destabilized Tregs in a manner dependent on the CD70 expression level on tumor cells, the CD27 expression level on CD4+ T cells, and the CD4+/CD8+ ratio in the TME. Clinical practice revealed that most NPC patients had CD70-high tumor histology, CD27-high lymphocyte infiltration, and a high CD4+/CD8+ ratio[74,75], which could theoretically benefit from anti-CD70 monotherapy and anti-CD70/ anti-PD-1 combination therapy, and yet this hypothesis requires translational validation in the future.

Mechanistically, at present, little is known about the downstream molecular mechanism of CD70-CD27 co-stimulatory signaling in tumor-infiltrating Tregs. We identified CD70+ NPC cells as a metabolic switch that could turn on lipid metabolism reprogramming on CD4+/CD27+ T cells, particularly CD4+ naïve T cells and Tregs, via enhanced transcriptional and enzymatic activities in cholesterol homeostasis via the mevalonate pathway, de novo fatty acid uptake, synthesis, oxidation, and mitochondrial integrity. Enhanced intracellular cholesterol accumulation and synthesis profoundly induced Treg suppressive activity and maintained Treg homeostasis in the TME via PD-1 induction to prevent IFNG-induced fragility, yet potentially being targeted by the synergistic PD-1 blockade. In CD70-induced Tregs, mitochondrial fatty acid oxidation-driven OXPHOS and ATP generation were significantly enhanced to maintain Treg metabolic fitness in the TME by activating 2-HG and succinate metabolism and promoting epigenetic activation of immunosuppressive genes. Instead, we also identified some key regulators and enzymes associated with lipid signaling, such as *SCAP*, *SQLE*, *LDLR*, and *FASN*, were unaffected by CD70-CD27 interaction, implying that the lipid signaling network in Tregs is highly complicated than is currently deciphered. With recent advances in elucidating the lipid signaling involved in the functional specialization of Tregs in glycolysis-low and lipid-enriched TMEs[20,60,76], we are approaching ever closer to resolving the mystery of metabolic reprogramming and the development of more effective therapies to disrupt lipid signaling in Tregs, which might alleviate tumor progression and synergistically work with other ICBs. However, the metabolic profile of tumor-infiltrating Tregs varies among cancer types due to the high tissue adaptivity and molecular flexibility of these cells[77]. Nevertheless, how CD70-CD27 signaling coordinates lipid signaling in other malignancies and identification of the alternative lipid metabolic mechanisms in NPC-infiltrating Tregs are essential subjects for future investigations.

Here, we particularly described the molecular mechanism associated with Treg development, homeostasis, and suppressive activity in one of the EBV-associated and CD70-high malignancies. But how EBV infection facilitates immune evasion of other malignancies, and whether other EBV+ cancers, including B cell lymphoma, T cell lymphoma, gastric carcinoma, and leiomyosarcoma, exhibited a similar Treg landscape and immunosuppression mechanism as NPC remain insufficiently recognized. To facilitate the translation of CD70 blockade in other CD70-high solid tumors, we found CD70 also significantly

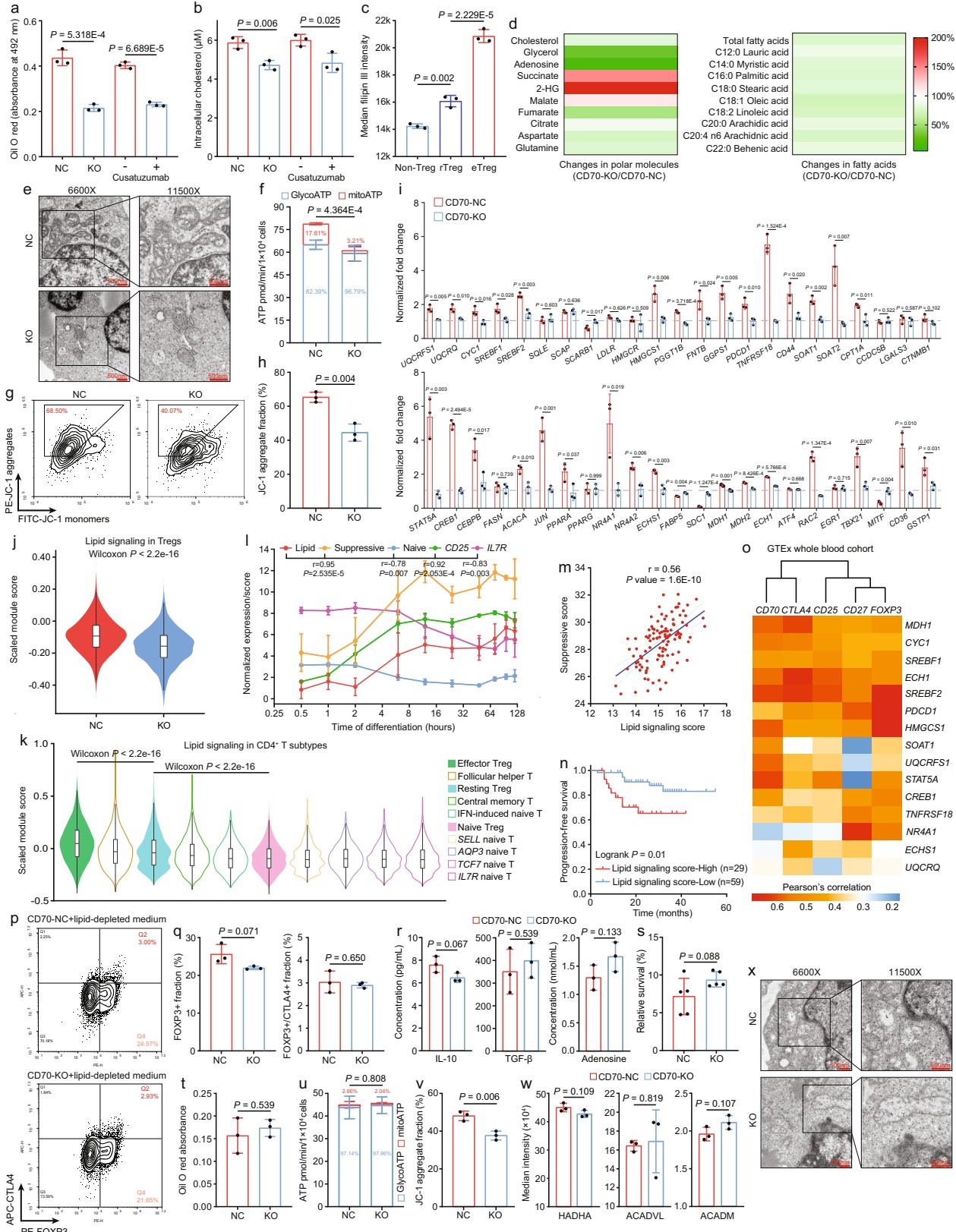

influenced ICB outcomes and T cell dysfunction in melanoma patients, and CD70-KO inhibited orthotopic melanoma growth without interfering with tumor cell proliferation. Likewise, future studies should focus on delineating how CD70-CD27 signaling enhances the suppressive activity and reprograms lipid metabolism in Tregs in other CD70-high solid tumors, such as melanoma.

CD70-CD27 signaling plays a bifacial role in regulating T cell responses in the TME. Previous literature has reported that CD70 can reduce immune surveillance by inducing T cell exhaustion, Treg activation, NK cell depletion, and macrophage recruitment in multiple malignancies, whereas some other studies have suggested that the CD70-CD27 axis also contributes to the proliferation and survival of

**Fig. 6 | Lipid signaling promotes Treg development, homeostasis, and suppressive activity.** Intracellular lipid (**a**) and cholesterol (**b**) in co-cultured CD4+ naïve T cells ($n = 3$, two-sided unpaired t test). **c** Median Filipin III intensity in each cell subtypes ($n = 3$, two-sided unpaired t test). **d** The change of intracellular metabolites in co-cultured CD4+ naïve T cells ($n = 1$). **e** Electron microscope images of co-cultured CD4+ naïve T cells. **f** OXPHOS and glycolysis ATP from co-cultured CD4+ naïve T cells ($n = 4$, two-sided unpaired t test). **g, h** JC-1 aggregates/monomers in co-cultured CD4$^+$ naïve T cells ($n = 3$, two-sided unpaired t test). **i** mRNA changes of lipid signaling-associated genes in co-cultured CD4+ naïve T cells ($n = 3$, two-sided unpaired t test). **j** Lipid signaling scores in Tregs ($n = 3731$ and $3264$, two-sided Wilcoxon signed-rank test). **k** Lipid signaling scores in CD4+ T cell subtypes ($n = 50$, two-sided Wilcoxon signed-rank test). **l** Changes of T-cell module scores and genes during Treg differentiation ($n = 3$, two-sided Pearson correlation analysis). **m** Pearson correlation (two-sided) between lipid and suppressive scores ($n = 112$). **n** Progression-free survival for NPC patients (high $n = 29$, low $n = 59$, two-sided log-rank test). **o** Pearson correlations (two-sided) between lipid/Treg-related genes in whole blood samples ($n = 337$). **p, q** Immunophenotyping of co-cultured CD4+ naïve T cells without lipid supplementation ($n = 3$, two-sided unpaired t test). **r** The change of immunosuppressive factors in lipid-depleted co-culture ($n = 3$, two-sided unpaired t test). **s** T-cell cytotoxicity in lipid-depleted co-culture ($n = 5$, two-sided unpaired t test). **t** Intracellular lipid in co-cultured CD4+ naïve T cells without lipid ($n = 3$, two-sided unpaired t test). **u** OXPHOS and glycolysis ATP from co-cultured CD4+ naïve T cells without lipid ($n = 4$, two-sided unpaired t test). **v** JC-1 aggregates/monomers in co-cultured CD4+ naïve T cells without lipid ($n = 3$, two-sided unpaired t test). **w** Median intensity of FA metabolism enzymes in co-cultured CD4+ naïve T cells without lipid ($n = 3$, two-sided unpaired t test). **x** Electron microscope images of co-cultured CD4+ naïve T cells without lipid. The n number represents $n$ biologically independent samples/cells/experiments. The data are presented as the mean ± SD (bar plots), median ± IQR (whiskers = 1.5×IQR, box & whiskers plots), and KDE (violin plots).

CD8+ T cells. In the present study, we have found the effector function of CD8$^+$ T cells is elevated upon CD70-KO and inhibition, due to impaired Treg-mediated immunosuppression, whereas the direct influence of CD70-CD27 signaling on CD8$^+$ T cells is negligible[41]. We have speculated that NPC has a CD4/CD27-enriched TME where the effect of CD70-CD27 interaction overwhelmingly takes place in CD4$^+$ naïve T cells and Tregs. In addition, considering the high CD27 expression in B cell lineages and high B-cell infiltration in the NPC microenvironment, the study is limited to only investigating the interaction between NPC cells and T cells with active or blocked CD70-CD27 signaling, as PBMC-engrafted humanized mice only generate a CD3+ T cell-enriched immune system. In the future, a more comprehensive study of CD70-CD27 signaling in NPC should utilize humanized mice generated by transplantation of human CD34$^+$ hematopoietic stem cells (HSCs), which can induce a more systematic humanized immunity consisting of T cells, B cells, NK cells, and myeloid cells.

In conclusion, our multiomics-driven functional investigations provide pre-clinical insights into how Tregs are developed, activated, and sustained in the NPC microenvironment, and show that CD70 inhibition is a therapeutically feasible approach to overcome the immunosuppressive TME, synergistically enhancing the efficacy of anti-PD-1 treatment. The anti-CD70+ anti-PD-1 combination therapy might produce particular benefits in patients with advanced or treatment-resistant NPC, which optimizes the conventional management of these patients. We also consider that the multiomics NPC T-cell cohort, along with the established functional modules, such as Treg suppression, T cell naiveness, and Treg lipid signaling modules, can be applied in future studies of immunosuppression and metabolic reprogramming in tumor-infiltrating T cells in other malignancies or in routine clinical applications for the quantitative evaluation of immunotherapy response. Finally, this lipid-driven immunosuppressive mechanism induced by CD70-CD27 signaling opens CD70-targeted precision therapy as additional avenue for NPC patients, and is potentially feasible and effective in patients with melanoma, depending on their specific TME landscapes.

## Methods

This research complies with all relevant ethical regulations, approved by the ethical committees at the University of Hong Kong, the University of Hong Kong-Shenzhen Hospital, the Hong Kong Red Cross, and the Committee of the Use of Live Animals in Teaching and Research at the University of Hong Kong.

### Patient enrollment

The study was approved by the ethics committee at the University of Hong Kong, the University of Hong Kong-Shenzhen Hospital, and the Hong Kong Red Cross. We complied with all related ethical regulations. Written informed consent was obtained from healthy blood donors and all patients with primary NPC and non-malignant nasopharyngeal inflammation for their tissues to be used in the spatial transcriptome sequencing, IHC/IF staining, flow cytometry analysis, and primary culture in this study.

### Sample aggregation and batch effect correction in single-cell sequencing

The raw fastq data of 50 samples from Gong et al., Liu et al., and Chen et al. NPC single-cell cohorts were processed and aggregated by Cell Ranger (v 6.1.2, 10x Genomics). Batch effects among single cells originated from different studies were corrected by Harmony (v 1.0, Broad Institute). After batch effect correction in the aggregated data, no significant batch effect was observed across the original studies (Supplementary Fig. 1b).

### Quality control, normalization, and clustering of the aggregated single-cell data

The gene expression matrix metadata was converted to a Seurat object using Seurat R package (v 4.0). Single cells with unique feature counts >4000 or <200, or >15% mitochondrial counts, were filtered out from the downstream analysis. Doublets were removed by DoubletFinder (v 2.0). From the remaining 357,206 cells, the gene expression dataset was normalized, scaled, and subsequently dimensionally reduced based on 5000 variant genes and principal components ($n = 30$), determined by the Elbow plot. Major cell lineages, including tumor cells, T cells, B cells, plasma cells, NK cells, myeloid cells, mast cells, fibroblasts, and normal epithelial cells, were projected on the two-dimensional UMAP representation and annotated using well-recognized signatures. To further identify finer subtypes within the T cell lineage, we performed the second-round UMAP reduction on 201,746 T cells with principal components ($n = 30$) and a resolution of 2. The doublets were further identified and removed by both graph-based clustering and DoubletFinder. The second-round UMAP reduction revealed 41 distinct subtypes in 189,750 T cells.

### Pseudotime developmental trajectory in CD4+ T cell subtypes

Monocle 3 was applied to determine the developmental processes within the CD4+ T subpopulations[78]. The Seurat object of T cells was converted into a cell data set (cds) object and the original UMAP clusters were retained based on implemented functions available in Seurat wrappers. Subsequently, Monocle 3 learned the trajectory graph and ordered cells in pseudotime with a specified root node (C5-SELL-CD4+ naïve T cells) in the converted cds object. The pseudotime trajectory and development lineages were then projected on the original UMAP plot. Cells with higher pseudotime indicated that they were in a later developmental stage. Two-sided Pearson correlation analysis and polynomial regression analysis were performed to evaluate the change of Treg and T$_{FH}$-specific transcription factors *FOXP3* and *BATF* expression during Treg and T$_{FH}$ developmental processes.

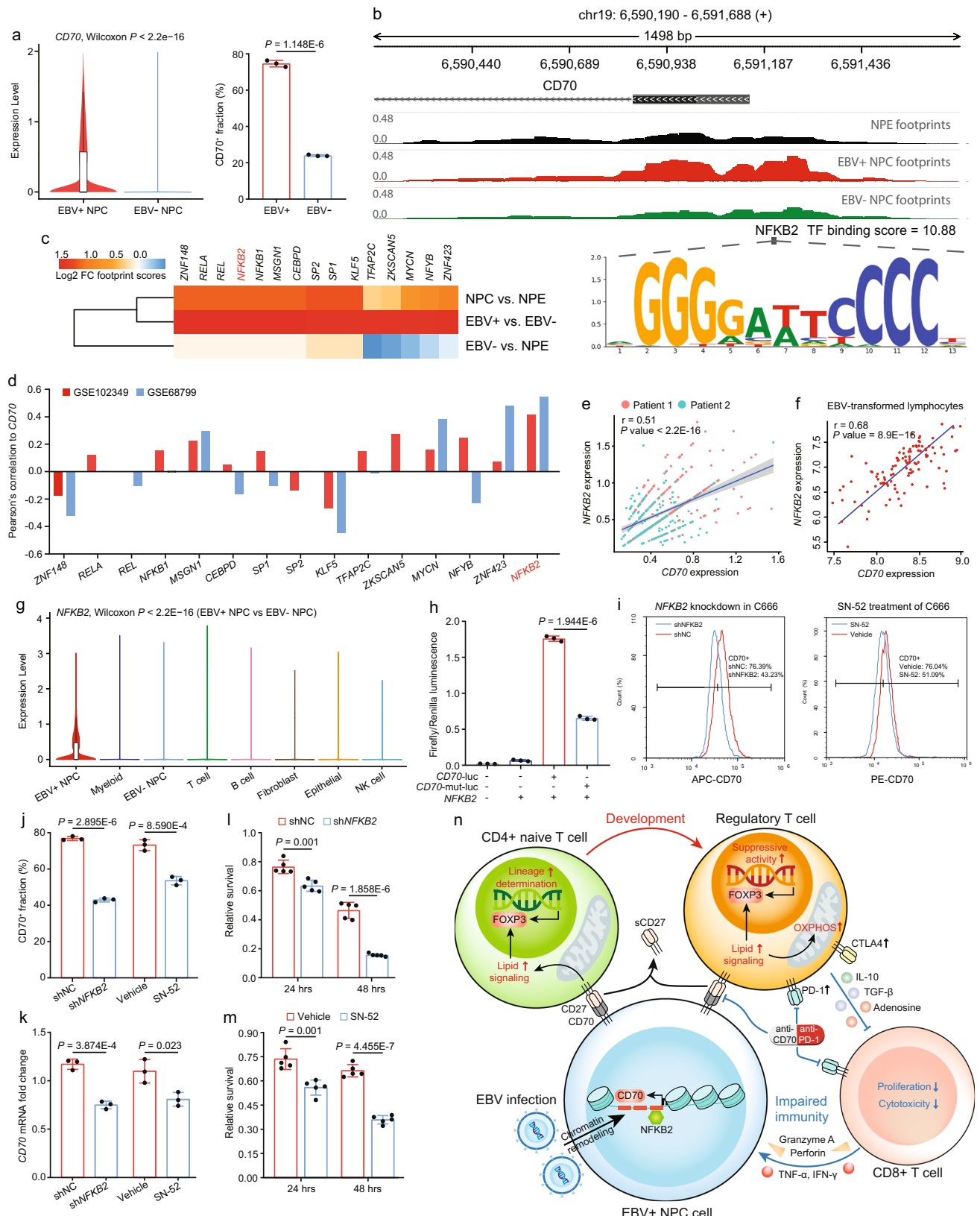

## Establishment of functional modules based on single-cell data

Based on MAST analysis and functional results, we identified and selected genes that were most representative to Treg suppressive activity, T-cell naiveness, and lipid signaling, listed in Supplementary Fig. 1f, h, and f. We subsequently used these signatures to establish generalized binomial linear regression models in the eTreg cluster for constructing the Treg suppressive and lipid signaling modules, and in naïve T cell clusters for constructing the T cell naïve module, via the glm function in R (v 4.0). The glm function assigned each single cell with a categorical variable (1 and 0), stating whether it belongs to the designated cluster or not. The binomial linear regression model calculated a distinct coefficient and $P$ value for each representative

**Fig. 7 | EBV infection increases chromatin accessibility at the *CD70* promoter and promotes *CD70* transcription via NFKB2. a** Left, the single-cell expression of *CD70* in EBV+ and EBV- NPC cells (tumor sample *n* = 36, two-sided Wilcoxon signed-rank test); Right, the quantified CD70+ cell fractions in EBV+ and EBV- NPC43 cells (*n* = 3, two-sided unpaired t test). **b** The chromatin accessibility measured by ATAC-seq at the *CD70* promoter in EBV+, EBV- NPC cells, and NPE cells. **c** Top 15 enriched TFs with the highest transcriptional activity in EBV+ NPC cells. **d** The Pearson correlation (two-sided) between each TF and *CD70* in GSE102349 (*n* = 112) and GSE68799 (*n* = 42) cohorts. **e** The spatial co-localization of *NFKB2* and *CD70* expression in down-sampled spatial spots (*n* = 100, two-sided Pearson correlation analysis with the 95% confidence band). **f** The Pearson correlation (two-sided) between *NFKB2* and *CD70* expression in EBV-transformed lymphocytes (*n* = 107, with the 95% confidence band). **g** Expression of *NFKB2* across major cell lineages in the NPC scRNA-seq cohort (*n* = 50, two-sided Wilcoxon signed-rank test). **h** Relative

firefly luminescence normalized by renilla luminescence in each experimental group (*n* = 3, two-sided unpaired t test). **i, j** The change of CD70 + fraction in shNC, shNFKB2, vehicle-treated, and SN-52-treated C666 cells (*n* = 3, two-sided unpaired t test). **k** The 18s-normalized mRNA fold change of *CD70* in shNC, shNFKB2, vehicle-treated, and SN-52-treated C666 cells (*n* = 3, two-sided unpaired t test). **l** T-cell cytotoxicity measured in the shNC and shNFKB2 PBMC co-culture systems (*n* = 5, two-sided unpaired t test). **m** T-cell cytotoxicity measured in PBMCs co-cultured with vehicle and SN-52-treated C666 cells (*n* = 5, two-sided unpaired t test). **n** The schematic illustration of the molecular mechanism of the EBV-NFKB2-CD70-CD27 signaling axis and feedback loop between NPC cells, CD4+ naïve T cells/Tregs, and CD8+ T cells in the NPC microenvironment. The *n* number represents *n* biologically independent spots/samples/experiments in each group. The data are presented as the mean ± SD (bar plots), median ± IQR (whiskers = 1.5 × IQR, box & whiskers plots), and KDE (violin plots).

signature based on its expression weight across all clusters. Only genes with *p* value lower than 0.05 were used in constructing the functional modules. The functional scores were then computed based on normalized gene matrix in bulk RNA-seq cohorts and single-cell cohorts, representing the quantified extent of Treg suppressive, T cell naïve, and lipid signaling activities.

### Processing of RNA-seq data
The NPC RNA-seq datasets used in this study included: GSE102349 (tumor *n* = 112, one sample was removed by failing the quality control, and 88 samples with complete follow-up data were included in the subsequent survival analysis), GSE68799 (tumor *n* = 42, normal *n* = 4), GSE13597 (tumor *n* = 25, normal *n* = 3), GSE53819 (tumor *n* = 18, normal *n* = 18) and GSE118719 (tumor *n* = 7, normal *n* = 4). The read counts in each RNA-seq cohort were generated by HTSeq (version 0.9.1) and normalized by DESeq2 (version 1.22.2). The quality of RNA-seq data was evaluated by Picard metrics (version 2.17.4) and RSeQC (version 2.6.4). Only the samples with at least 60% of reads mapped to coding regions were included in the downstream analysis.

### Single-cell gene set variation analysis (GSVA)
The gene count data matrix with cluster annotation information was constructed in eTregs, rTregs, and nTregs. The gene list files were parsed from http://www.gsea-msigdb.org/gsea/downloads.jsp. GSVA on the single-cell data from the three Treg subtypes was performed with default settings and Poisson kernel argument, as implemented in the GSVA package (v 1.40.1). The pathway enrichment scores were first assigned to individual cells, and subsequently averaged within each Treg cluster and z-normalized across all Treg clusters.

### Isolation of peripheral blood mononuclear cells from healthy donors
The collection of peripheral blood from healthy donors was approved and performed by the Hong Kong Red Cross. All donors formally consent for their blood to be used in experiments and analysis mentioned in this study. The collected blood was transferred from the Hong Kong Red Cross to the laboratory at room temperature. The whole blood was mixed with citrate phosphate dextrose (CPD) and 0.5% EDTA in phosphate-buffered saline (PBS) with a 1:1 ratio. Subsequently, 38 mL blood mixture was transferred to each 50 mL conical tube with 12 mL Ficoll-Paque PLUS (Cytiva). The blood mixture was centrifuged at 400× *g* with the lowest acceleration and deceleration speed for 25 mins at room temperature. The PBMCs were carefully transferred into new 50 mL conical tubes and centrifuged at 300×g for 10 mins. The supernatant was aspirated, and PBMCs were washed with 50 mL PBS containing 1% FBS (Gibco) and 0.5% EDTA, and were centrifuged at 300×g for 10 mins. 2 mL red blood cell lysis buffer (Sigma) was added per tube, and the PBMC mixture was incubated for 5 mins at room temperature. 10 mL RPMI-1640 supplemented with 10% FBS and 1% PS (Gibco) was added to

neutralize the lysis buffer in each tube. The PBMCs were centrifuged at 300× *g* for 5 mins and were ready for isolation and activation. The remaining PBMCs were re-suspended in the freezing medium containing 90% FBS and 10% DMSO (Sigma) and frozen at −80 °C using the vertical gradient freeze method. The frozen PBMCs were transferred into liquid nitrogen for long-term storage. Freshly isolated PBMCs and PBMCs that had been frozen for less than one month were used in the study.

### Magnetic isolation and TCR activation of CD4+ naïve T cells from PBMCs
CD4+ naïve T cells were magnetically isolated from fresh PBMCs using the naïve CD4+ T cell isolation kit II, human (Miltenyi), as per manufacturer's instructions. The purity and CD27 expression of isolated CD4+ naïve T cells was determined via flow cytometry by staining the isolated T cells with FITC anti-human CD45, APC/Cy7 anti-human CD45RA, and APC anti-human CD27 antibodies (BioLegend). The CD4+ naïve T cells were activated in a 96-well plate with a density of 3×10^5 cells/well in 150 μL TexMACS™ GMP medium (Miltenyi) supplemented with 10% heat-inactivated FBS (HI-FBS), 1% PS, and 50 μM 2-mercaptoethanol (Gibco), in the presence of 10 μg/mL plate-coated anti-human CD3 (TONBO) or 1:100 T Cell TransAct™, human (Miltenyi) for 2-3 days, as per manufacturer's instructions.

### Cell lines and culture
Human NPC cell lines C666 (EBV+) and NPC43 (EBV+ and EBV−), as well as human NPE cell lines NP460 and NP69 were provided by Professor George Sai-Wah Tsao at the University of Hong Kong. The 293FT cell line were purchased from Invitrogen (R70007). All NPC and NPE cell lines were cultured in RPMI-1640 (Gibco) supplemented with 10% FBS (Gibco) and 1% PS (Gibco), and the 293FT cell line were cultured in DMEM (Gibco) supplemented with 10% FBS (Gibco), 1% PS (Gibco), 1× MEM amino acids solution (Gibco) and 1 mM sodium pyruvate (Gibco). All cell lines were incubated in a humidified incubator at 37 °C with 5% CO2 supply. All the cell lines used in the study have been tested to be authentic by using STR profiling and free of mycoplasma contamination.

### Establishment of in vitro transwell-based and direct co-culture systems
NPC and NPE cells were seeded into a 24-well plate with a density of 5×10^4 cells/well in 1 mL complete RPMI-1640 24 hours prior to adding CD4+ naïve T cells. For the transwell-based co-culture system, 1 × 10^5 naïve CD4+ T cells were seeded into a 0.3 cm^2 transwell insert with 1 μm pore size, avoiding T cell migration but allowing cytokine exchange. For the direct co-culture system, 2 × 10^5 naïve CD4 + T cells were directly seeded into the 1.9 cm^2 24-well plate together with C666 cells or NP69/NP460 cells. The negative control group has only 2 × 10^5 CD4 + naïve T cells per well without any NPC and NPE cells seeded in the 24-well plate. Cells were cultured in 1.5 mL TexMACS™ GMP

medium (Miltenyi) supplemented with 10% HI-FBS, 1% PS, 50 µM 2-mercaptoethanol (Gibco), 25 IU/mL IL-2 (Miltenyi) and 2 ng/mL TGF-β1 (PerproTech) for 3 days.

### Immunophenotyping of CD4+ and CD8+ T cells by flow cytometry

After 3-day co-culture, CD4+ T cells were collected from the supernatant, washed with cell staining buffer (BioLegend), and centrifuged at 300× *g* for 5 mins. Subsequently, the T cells were resuspended in cell staining buffer and stained with desired surface marker antibodies for 20 mins on ice. T cells were washed with 2 mL cell staining buffer and centrifuged at 300× *g* for 5 mins. Subsequently, the T cells were fixed and permeabilized using the true-nuclear transcription factor buffer set (BioLegend), as per manufacturer's instructions. Nuclear factor FOXP3 and intracellular CTLA4 were stained with the PE/Pacific Blue anti-human FOXP3 antibody (BioLegend) and APC anti-human CTLA4 antibody (Biolegend), respectively, as per manufacturer's instructions. The stained cells were sequentially washed with 1× Perm buffer and cell staining buffer (BioLegend) and then suspended in cell staining buffer for flow cytometry analysis. Surface GITR, ICOS, and PD-1 were stained with PE/Cy7 anti-human GITR (BioLegend), FITC anti-human ICOS (BioLegend), and APC anti-human PD-1 (BioLegend), as per manufacturer's instructions.

For evaluating cytotoxicity of CD8+ T cells in the PBMC co-culture system, co-cultured PBMCs were resuspended in cell staining buffer and stained with APC anti-human CD8 (BioLegend) for 20 mins on ice, and then fixed and permeabilized using the Cyto-Fast fix-perm buffer set (BioLegend), as per manufacturer's instructions. Intracellular Granzyme A and Perforin were stained with PE anti-human granzyme A antibody (BioLegend) and PE anti-human perforin antibody (BioLegend) separately, as per manufacturer's instructions. NovoCyte Quanteon flow cytometer (Agilent) was used to detect conjugated fluorochrome intensity, and NovoExpress software (v 1.5.6, Agilent) was used to analyze the flow cytometry data. Detailed information on antibody catalog numbers, clones, and concentrations is listed in Supplementary Table 5.

### Enzyme-linked immunosorbent assay (ELISA)

Cell culture supernatants from co-culture assays were collected after 48- or 78-hour co-culture. Subcutaneous tumors were dissected from humanized mice and snap-frozen in liquid nitrogen. The frozen tumors were minced in liquid nitrogen and dissociated in RIPA buffer (Sigma) + protease inhibitor (Sigma). The total protein in the tumor lysates was quantified using the BCA method by the CLARIOstar Plus fluorescence plate reader. Standard curves for our interested cytokines, including IL-10 (Abcam), TGF-β (Invitrogen), TNF-α (Invitrogen), IFN-γ (Abcam), and CD27 (Invitrogen), were determined by the standards supplemented by the manufactures. The concentration of our interested cytokines in cell culture supernatants and tumor lysates from different experimental groups were quantified based on a measurement wavelength of 450 nm, a reference wavelength of 620 nm, and their corresponding standard curves.

### RNA extraction and quantitative reverse transcription PCR (qRT-PCR)

Total RNA was extracted using the TRIZOL Reagent (Takara) and cDNA was synthesized by reverse transcription using the PrimeScript RT Reagent Kit with gDNA eraser (Takara). 0.5 ng cDNA was mixed with qRT-PCR primers (BGI genomics) and SYRB Green (Takara) in a 10 µL volume and analyzed by LightCycler 480 II Real-time PCR Detector (Roche). The normalized mRNA fold change $2^{-\Delta\Delta Ct}$ was calculated by normalizing Ct values of corresponding genes to the internal reference gene 18s, and then normalizing to ΔCt values in the negative control group. The primer sequences were listed in Supplementary Table 3.

### Cell-cell communication analysis

CellphoneDB (v 2.0) and CellChat (v 1.0) were applied to quantify and visualize the statistically and biologically enriched cell-cell communication via ligand-receptor bindings among CD4+ naïve T cells, Treg subtypes, NPC, and NPE cells. The count file and meta file for CellPhoneDB were constructed by extracting the raw gene count and cell annotation from the Seurat object. Single-cell expression matrix and cluster annotation of the selected subtypes were processed by CellphoneDB in python (v 3.6). CellChat utilized the Seurat object in the R environment to identify overexpressed interacting pairs in each cluster and compute communication probability using the law of mass action. Cell-cell interacting pairs identified by CellChat in fewer than 10 cells were removed from the downstream analysis. To compute total cell-cell communication strength and differentially expressed interaction signals in Treg subtypes from primary NPC, INP, and NPC peripheral samples, CellChat analysis was performed on scRNA-seq data from these three sample origins separately, and merged via the mergeCellChat function. The default parameters were used throughout our interaction analysis described in https://github.com/Teichlab/cellphonedb, https://htmlpreview.github.io/?https://github.com/sqjin/CellChat/blob/master/tutorial/CellChat-vignette.html.

### Spatial transcriptome sequencing and analysis

Visium spatial transcriptome sequencing was performed on frozen tissue sections obtained from 2 patients with primary and treatment-naïve NPC. Hematoxylin and eosin (H&E) staining was performed and imaged on each frozen section to determine the optimal sequencing location and used as a reference for the downstream spatial analysis. Then, tissue optimization experiments were conducted using the Visium spatial tissue optimization slide & reagents kit (10× Genomics) to determine the RNA quality (RIN > 7.0) and optimal permeabilization time. 6-minute tissue permeabilization yielded the highest fluorescence signal in both frozen sections. The frozen tissues were placed onto Visium spatial gene expression slide (10× Genomics), fixed, and permeabilized to release RNA, which bonded to adjacent capture probes. Next, RNA reverse transcription, cDNA synthesis, denaturation and amplification, quality control as well as spatial gene library construction were performed using Visium spatial gene expression reagents kit (10× Genomics). The spatial transcriptome library was sequenced on a NovaSeq 6000 (Illumina) and mapped to the human genome (hg38) using Spatial Ranger (v 1.3.1, 10× Genomics). The processed gene-voxel matrices contained spatial information of 1,089 (patient 1) and 1,331 (patient 2) spots and were further analyzed in Seurat (v 4.0). The spatial sequencing data were normalized using the SCTransform function followed by PCA reduction. Since each spatial spot contained multiple cells with varied lineages, we deconvoluted the spatial data using the SCTransformed NPC scRNA-seq data as a reference and estimated the underlying cellular composition in each voxel via the FindTransferAnchors function (normalization method = "SCT") and the TransferData function (anchorset = anchors identified by the FindTransferAnchors function), implemented in Seurat (v 4.0). The fractions of NPC cells and Tregs in each spatial spot were projected on the slide image. The tumor core was collectively determined by scRNA-seq-based spatial deconvolution analysis and pathological examination of the H&E staining images by two professional pathologists (JH and YW). Pearson's correlation analysis was performed to evaluate the spatial co-localization between NPC cells/CD70 and Tregs.

To spatially map cell types by integrating Visium spatial data with scRNA-seq reference, we used cell2location (version 0.1). Briefly, cell2location decompose multi-cell spatial transcriptomics data into the abundance estimation of reference cell types in a spatially coordinates by applying a principled Bayesian model. First, we build a high-quality reference of cell annotations using our integrated NPC scRNA-seq data, down-sampled to reference scRNA-seq data containing 1000

cells per cluster and feed it to cell2location as the input. Second, cell2location perform gene selection and define expression signatures of each cell type. Next, we use the default option of negative binomial regression model in cell2location to estimate the reference cell type signatures and to obtain cell-type locations. Each Visium section was analyzed separately with default parameters, except train_args = ' max_spochs': 300; 'N_cells_per_location': 30; 'detection_alpha': 200. A 5% percentile of the posterior distribution of mRNA counts was used to represent the number of mRNA molecules contributed by each cell type to each spot and to be plotted for visualization.

### Immunohistochemistry (IHC) and immunofluorescence (IF) staining

The formalin-fixed paraffin-embedded (FFPE) NPC tissues used for IHC staining were collected from the Department of Pathology, The University of Hong Kong-Shenzhen Hospital. The tissue slide was placed in a 70 °C oven overnight, and sequentially immersed in xylene, 1:1 xylene: ethanol, 100% ethanol, 95% ethanol, 75% ethanol, 50% ethanol, and tap water for 2 mins each step to completely dewax. The slides were immersed in 1× antigen retrieval buffer (Dako) for 45 mins at 95 °C, and then cooled down naturally for at least 2 hours. The slides were washed with PBS three times and incubated with 3% $H_2O_2$ for 1 hour at room temperature. The slides were additionally washed with PBS and incubated with blocking buffer for 2 hours at room temperature. The slides were incubated with anti-CD70 antibody (Cell Signaling) in PBS containing 1% bovine serum albumin (BSA) at 4 °C overnight. After washing with PBS, the slides were treated with labeled polymer-HRP and DAB substrate using the Real™ EnVision Detection SystemsPeroxidase/DAB, Rabbit/Mouse (Dako), as per manufacturer's instructions. Subsequently, the slides were washed, stained with hematoxylin (Thermofisher), and dehydrated by sequentially immersing the slides into 50% ethanol, 75% ethanol, 95% ethanol, 100% ethanol, 1:1 xylene: ethanol, and xylene. The slides were sealed with DPX mounting medium (Sigma), and the images were taken under a bright-field microscope.

For IF staining, the slides were blocked with 3% BSA in PBS for 1 hour at room temperature, and incubated with primary antibodies, including mouse anti-human FOXP3 (Abcam) and rabbit anti-human CTLA4 (Abcam) antibodies at 4 °C overnight. The slides were washed with PBS three times and incubated with secondary antibodies, including Alexa 488 goat anti-mouse (Invitrogen) and Alexa 555 donkey anti-rabbit (Invitrogen) antibodies, for 1 hour at room temperature. After being washed with PBS, the slides were incubated with 1:1000 DAPI for 10 mins at room temperature. The slides were sealed with Fluoromount™ (Sigma), and the images were taken by the Nikon's Digital Eclipse C1 Microscope System. The primary antibodies used for IHC and IF staining are listed in Supplementary Table 5.

### CRISPR-Cas9 CD70 knockout in NPC cells

sgRNA sequences specifically targeting human and mouse *CD70* were designed from https://cctop.cos.uni-heidelberg.de/. The forward and reverse sgRNAs were synthesized by Genewiz (China) and diluted to 100 µm in distilled water. The sgRNA was phosphorylated and annelid using the Anza™ T4 PNK kit (Invitrogen), as per manufacturer's instructions. Then, CRISPR-V2 plasmid underwent enzymatic digestion using the Fast Digest Esp3l (Thermofisher) and was ligated with the phosphorylated sgRNA duplex using the T4 ligase kit (Takara), as per manufacturer's instructions. The CD70-targeting CRISPR plasmid with ampicillin resistance was transformed in stabl3 (Invitrogen) grown on the ampicillin+ LB agar plate (Invitrogen). The transformed plasmid was extracted and purified using the plasmid extraction kit (Qiagen), as per manufacturer's instructions. CRISPR-NC plasmid containing a non-specific sgRNA was transformed, extracted, and purified using the same method.

$1 \times 10^6$ 293FT cells were transfected with the CRISPR plasmids with two packaging plasmids, including PsPAX2 and pMD2.g, at a ratio of 3 µg: 2 µg: 1 µg, mixed with 300 µL Opti-MEM™, GlutaMAX™ supplement (Thermofisher), 6 µL Lipofectamine 2000 (Invitrogen) and 700 µL Dulbecco's Modified Eagle medium (Thermofisher) supplemented with 10% FBS (Gibco), 1× MEM amino acids solution (Gibco) and 1× sodium pyruvate (Gibco). After 72-hour transfection, lentivirus-containing supernatants were collected and centrifuged at 1100× g at 4 °C for 15 mins. C666 cells/B16-F10 cells were transduced with lentivirus containing the CD70-targeting CRISPR plasmid or the CRISPR-NC plasmid in complete RMPI-1640 for 48 hours. 10 mM puromycin (Sigma) was used to select for stably transduced C666 cells/B16-F10 cells. Single clones for CD70- C666 cells/CD70- B16-F10 cells were sorted by BD FACSMelody (BD Biosciences) using the single-cell purity sorting mode. CD70-targeting sgRNA sequences used in C666 and B16-F10 cells are listed in Supplementary Table 4.

### The Treg suppression assay

CD4+ naïve T cells were co-cultured with CD70-NC/CD70-KO C666 cells or IgG-treated/cusatuzumab-treated C666 cells for 3 days in 1.5 mL complete T cell culture medium supplemented with 25 IU/mL IL-2 (Miltenyi) and 2 ng/mL TGF-β1 (PerproTech). After 3-day co-culture, paired CD8+ T cells were freshly isolated from the same donor and labelled with CellTrace™ pacific blue (Invitrogen), and were mixed with the co-cultured CD4+ T cells at a 1:1 ratio in T cell culture medium supplemented with 1:100 T cell Transact (Miltenyi) and 25 IU/mL IL-2 (Miltenyi). After 48-hour co-culture between CD4+ T cells and paired CD8+ T cells, the proliferation of CD8+ T cells was determined by pacific blue intensity in the CD8+ T cell population via NovoCyte Quanteon Flow Cytometer, and proliferation index was quantified by the built-in CFSE analysis module in NovoExpress software.

### Cell proliferation and survival analysis by the colorimetric XTT assay

Without PBMC co-culture, NPC cells were seeded in a 96-well plate at a density of 1000 cells/well/100 µL RPMI-1640 complete medium. The cell proliferation was evaluated by using the XTT assay (Roche), as per manufacturer's instructions, for a consecutive 5 days. With PBMC co-culture, PBMCs in the supernatant were removed, and the tumor cells were washed with PBS twice to remove the remaining PBMCs and dying cells. Fresh RPMI-1640 complete medium was added to each well with adherent tumor cells. The read absorbance (492 nm as the absorbance wavelength and 620 nm as the reference wavelength) was evaluated by using the XTT assay (Roche), as per manufacturer's instructions, at 24 hours and 48 hours post-PBMC co-culture. The relative absorbance was calculated by subtracting the read absorbance from the absorbance of an empty well. Relative cell survival was normalized by the relative absorbance of viable cells without PBMC co-culture.

### Establishment of the NPC-PBMC co-culture system and apoptosis analysis

CD70-NC and CD70-KO C666 cells were seeded into a 24-well plate with a density of $1 \times 10^5$ cells/well 24 hours prior to adding activated PBMCs. $5 \times 10^5$ PBMCs were seeded into the 24-well plate in TexMACS™ GMP medium (Miltenyi) supplemented with 10% HI-FBS, 1% PS, 50 µM 2-mercaptoethanol and 25 IU/mL IL-2 (Miltenyi), for 48 hours. After 48 hours, the dead tumor cells were collected and viable adherent tumor cells were dissociated by TrypLE™ (Thermofisher) at 37 °C for 3 mins. The apoptosis of CD45- cells was determined by the PE annexin V apoptosis detection kit I (BD Biosciences) + FITC-CD45 anti-human antibody (BioLegend), as per manufacturer's instructions. The cellular intensities of FITC-CD45, PE-annexin V and APC-7-AAD was compensated and measured by NovoCyte Quanteon flow cytometer and analyzed by the NovoExpress software.

## CFSE staining and analysis

CellTrace™ CFSE cell proliferation kit (Invitrogen) was used to stain activated PBMCs with CFSE prior to co-culture with NPC cells, as per manufacturer's instructions. After 48-hour co-culture, the proliferation of CD8+ T cells was determined by staining the APC anti-human CD8 antibody (BioLegend), and CFSE intensity in the CD8+ subpopulation was measured by NovoCyte Quanteon Flow Cytometer, and proliferation index was quantified by the built-in CFSE analysis module in NovoExpress software.

## Anti-CD19 CAR lentivirus packaging and concentration

293FT cells were seeded at the density of $1 \times 10^6$ cells/well in a six-well plate in the DMEM transfection medium. Cells were transfected with the packaging plasmids psPAX2 (Addgene,) and pMD2.G (Addgene) together with the CAR overexpression plasmid (pLV[Exp]-EGFP-EF1A-CD19-CAR) in a ratio of 1:2:4 using the Lipofectamine 3000 transfection reagent (Invitrogen) according to the manufacturer's protocol. The lipid-DNA complex was incubated for 6 hours at 37 °C, 5% $CO_2$. Next, the cell medium was replaced with a fresh lentivirus packaging medium supplemented with 1× viral boost reagent (Alstem), following the manufacturer's instructions.

The lentivirus was collected at 48 hours and 72 hours post-transfection. Cell medium containing the lentivirus was centrifuged at 1,100×g, 4 °C for 15 min to remove the cell debris. To concentrate the lentivirus particles, we mixed the lentivirus with the precipitation solution (TaKaRa, Lenti-X Concentrator), as per the manufacturer's protocols. The mixture was incubated at 4 °C for 30 minutes and centrifuged at 1500×g for 45 minutes at 4 °C. The lentivirus pellet was resuspended in cold PBS to make the 100× concentrated virus.

## Anti-CD19 CAR-T cells production

Fresh CD3+ T cells isolated from human PBMCs were activated with T cell TransACT (Miltenyi) before anti-CD19 CAR lentivirus transfection. Activated human T cells with a cell density of $2 \times 10^6$ /mL were cultured with the concentrated lentivirus (50 μL/mL) supplemented with 8 μg/mL polybrene and centrifuged at 500× g at 32 °C for 90 min. The cell culture medium was refreshed 24 hours later and the transfected T cells were subsequently cultured in a complete T cell culture medium supplemented with 25 IU/mL IL-2 for additional 48 hours. Anti-CD19-CAR-expressing T cells sorted based on EGFP expression using BD FACS Aria SORP.

## Inhibition of CD70 by cusatuzumab and CD70-blocking-only antibody treatment

The humanized monoclonal anti-CD70 antibody, cusatuzumab and anti-CD70 blocking-only antibody, was purchased from MyBioSource (MBS156578) and AdipoGen (ANC-222-050), respectively. Varied cusatuzumab concentrations, including 1 μg/mL, 2.5 μg/mL, 5 μg/mL, and 10 μg/mL, were tested in the in vitro co-culture systems. 5 μg/mL cusatuzumab and the anti-CD70 blocking-only antibody diluted in the complete T cell culture medium was determined as the optimal concentration to inhibit CD70-CD27 interaction between C666 cells with CD4+ naïve T cells and Tregs. 5 μg/mL IgG1, λ antibody treatment was used as vehicle control in vitro. In addition, 5 mg/kg cusatuzumab was determined as the optimal concentration in the in vivo settings.

## Mice

All animal experiments were approved by the Committee of the Use of Live Animals in Teaching and Research at the University of Hong Kong under protocol 4924-19. NOD.Cg-Prkdc$^{scid}$Il2rg$^{tm1Wjl}$/SzJ (NSG) mice and C57BL/6J mice were purchased from the Centre for Comparative Medicine Research, the University of Hong Kong. Female NSG mice at 6 weeks of age were used for PBMC-engrafted immune humanization. Male or female C57BL/6J mice at 4-6 weeks of age were used for melanoma inoculation. The maximal end-point tumor size permitted by

the ethics committee is 2 cm in diameter and no tumor burden exceeded this limit in our study. Mice were housed on 12-hour light-dark cycles with the ambient temperature at 20–25 °C and 40–60% humidity at the Centre for Comparative Medicine Research, the University of Hong Kong.

## Establishment of the PBMC humanized mouse model and xenotransplantation

Freshly isolated PBMCs from healthy donors were activated with T Cell TransAct™, human (Miltenyi) for 3 days, and expanded in TexMACS™ GMP medium (Miltenyi) supplemented with 10% HI-FBS and 1% PS, 50 IU/mL IL-2 (Miltenyi) for additional 3 days. $1 \times 10^7$ activated PBMCs in 200 μL PBS were injected into 6-week-old female NSG mice via tail-vein injection. $3 \times 10^6$ CD70-NC and CD70-KO C666 cells were injected subcutaneously into humanized mice on day 9 post-PBMC injection. The humanized immunity started to re-establish one-week post-PBMC injection and was determined on day 14 by staining mouse PBMCs with FITC anti-human CD45 antibody.

For the xenotransplanted humanized mouse model, Xeno76 grown on nude mice was dissected into small chunks with an identical volume (2 mm × 2 mm × 2 mm). 4-week-old female NSG mice were anesthetized by intraperitoneal injection of 150 μL ketamine and xylazine mixture. The left dorsal flank of each mouse was cut with sterile scissors, and the PDX was placed inside the mouse's body. The incision was closed by Nylon. After 2 weeks when the PDX started to grow, $1 \times 10^7$ activated PBMCs in 200 μL PBS were injected into the xenotransplanted NSG mice. Different regimens, including 5 mg/kg IgG1, λ antibody, 5 mg/kg cusatuzumab, 3 mg/kg camrelizumab, and 5 mg/kg cusatuzumab + 3 mg/kg camrelizumab, were given to the PDX-bearing mice intraperitoneally once every three days, starting on day 7 post-PBMC injection.

## Primary NPC cell culture

After the endoscopic biopsy, the fresh NPC tissue samples (2–3 mm$^3$) were rinsed with RPMI-1640 medium supplemented with 10% FBS and 2% PS three times on ice, and then placed on a 6-well culture plate in 2 mL RPMI-1640 medium supplemented with 10% FBS, 1% PS and 4 μM Y-27632 (MCE). The adherent primary NPC cells were cultured for 3 days. Meanwhile, autologous PBMCs were isolated from the paired peripheral blood using the Ficoll-Paque method mentioned above and activated in T cell medium supplemented with 25 IU/mL IL-2 for 3 days. On day 3, the activated PBMCs were added to co-culture with primary NPC cells at a ratio of 4:1 in the T cell medium supplemented with 25 IU/mL IL-2 and 5 μg/mL cusatuzumab (MyBioSource) or 5 μg/mL IgG1, λ (Sigma) for 48 hours. After then, dead cells were gently washed away by PBS three times, and the viable adherent cells were measured by the colorimetric XTT assay. The immune cells were collected and stained with FITC-CD4, Pacific Blue-FOXP3, and APC-CTLA4 to evaluate Treg fraction changes after cusatuzumab treatment.

## Establish NPC organoids and the organoid-PBMC co-culture system

Xeno76 grown on nude mice was dissociated into single cells by using Liberase™ (Sigma) containing 0.5 μg/mL DNAse I (Sigma) at 37 °C for 30 min. Dissociated cells were mixed with ADF+/+/+ (AdDMEM/F12 supplemented with HEPES, GlutaMax, and Penicillin-Streptomycin) and BME with a ratio of 1:3 and seeded per 24 wells. After solidification, the organoid medium was added. Organoid medium contained B27 supplement, Primocin, 1.0 mmol/L N-acetyl-l-cysteine, 10 mmol/L Nicotinamide, 50 ng/mL human EGF, 500 nmol/L A83-01, 10 ng/mL human FGF10, 5 ng/mL human FGF2, 1 μmol/L Prostaglandin E2, 3 μmol/L CHIR 99021, 1 μmol/L Forskolin, 10 μM Rho kinase inhibitor Y-27632 (MCE), 4% R-spondin, and 4% Noggin. During culturing, the medium was refreshed at most every three days. For passaging, organoids were collected from the plate by disrupting the BME droplets with a P1000, collecting and washing in 10 mL ADF+/+/+. Pellet

was resuspended in 1 mL of 10X TrypLE Express (Thermofisher) and incubated at 37 °C. Digestion was closely monitored, and the suspension was pipetted up and down every 5 minutes to aid disruption of the organoids. TrypLE digestion was stopped when organoids were disrupted into single cells by adding 10 mL ADF+/+/+. Cells were subsequently resuspended in ice-cold 75% BME in ADF+/+/+ and plated with a ratio of 1:3 to allow efficient outgrowth of new organoids. The medium was changed every 2–3 days, and organoids were usually passaged with a split ratio of 1:3 every 7–10 days.

The organoids were gently collected, centrifuged at 300× *g* for 10 mins at 4 °C, and subsequently mixed with $1 \times 10^5$ activated PBMCs in 1.5 mL AdDMEM/F12 supplemented with 10% HI-FBS, 1% PS, 50 μM 2-mercaptoethanol, 25 IU/mL IL-2. After 48-hour co-culture, the organoids were dissociated into single-cell suspension using TrypLE Express (Thermofisher). Apoptosis of CD45- organoid-derived NPC cells was determined by the PE annexin V apoptosis detection kit I (BD Biosciences) + FITC-CD45 anti-human antibody (BioLegend), as per manufacturer's instructions. The cellular intensities of FITC-CD45, PE-annexin V, and APC-7-AAD were compensated and measured by the NovoCyte Quanteon flow cytometer and analyzed by the NovoExpress software.

### Single-cell sequencing of PBMCs
The PBMCs co-cultured with C666-NC and C666-KO NPC cells after 48 hours were isolated from the culture medium by centrifuging at 300× *g* for 5 mins, and re-suspended in PBS before single-cell encapsulation. The single-cell encapsulation and library preparation was done at the Centre for PanorOmic Sciences, the University of Hong Kong. The single-cell suspension was converted to uniquely barcoded RNA, using the Chromium single cell 3′ library, gel Bead & multiplex kit, and Chromium single cell chip kit (10× Genomics), as per manufacturer's instructions. The libraries were sequenced on a NovaSeq 6000, and mapped to the human genome (hg38) using Cell Ranger (v 6.1.2, 10x Genomics). Gene positions were annotated as per Ensembl build 85 and filtered for biotype. The sample aggregation and analysis were performed using the aforementioned methods.

### GSEA analysis on Tregs and CD8+ T cells
We constructed gct and cls files for GSEA analysis based on the single-cell expression matrix and metadata of NC-Tregs and KO-Tregs. GSEA for Windows (v 4.2.3) was downloaded from https://www.gsea-msigdb.org/gsea/downloads.jsp. The pathway enrichment scores in hallmark, KEGG, REATOME, and GO:BP gene sets were computed based on Signal2Noise analysis.

### Oil O red staining and quantification
For staining of total intracellular lipid in CD4+ T cells co-cultured with CD70-NC and CD70-KO C666 cells or IgG-treated and cusatuzumab-treated C666 cells, CD4+ naïve T cells after 3-day co-culture were washed by PBS and fixed with 10% formalin for 30 mins at room temperature. The fixed T cells were then washed with 60% isopropanol for 5 mins and stained with 0.5% Oil O red staining solution for 20 mins at room temperature. Then, T cells were washed with ddH₂O and 60% isopropanol. Lastly, the Oil O red stain in T cells were extracted by 100% isopropanol for 5 mins. The absorbance of Oil O red was quantified on a multi-plate reader at a 492 nm measurement wavelength and a 620 nm reference wavelength.

### Cellular cholesterol content measurement
Cellular-free cholesterol content was measured using the cholesterol cell-based detection assay kit (Cayman). Co-cultured CD4+ T cells were fixed and permeabilized using the true-nuclear transcription factor buffer set (BioLegend), as per manufacturer's instructions. Then the cells were stained with filipin III (Cayman), PE anti-human FOXP3 (BioLegend), and APC anti-human CTLA4 (BioLegend). The

median filipin III intensity in FOXP3-/CTLA4-, FOXP3 + /CTLA4- and FOXP3+/CTLA4+ cells was measured by the BD Aria SORP flow cytometer. For the oxidation-based quantification of cholesterol, total cholesterol was extracted using the cholesterol extraction kit (Sigma), and then oxidized using the Amplex Red Cholesterol Assay Kit (Invitrogen), per manufacturer's instructions. The oxidized cholesterol was analyzed by the CLARIOstar Plus fluorescence plate reader.

### Mass spectrometry-based metabolomics in co-cultured CD4+ naïve T cells and supernatant
Sample processing and gas chromatography-mass spectrometry (GC-MS) analysis were performed at the Centre for PanorOmic Sciences-Proteomics and Metabolomics Core, the University of Hong Kong. The number of co-cultured CD4+ naïve T cells in each sample was determined by the Countess 3 Cell Counter (Thermofisher). $4\text{-}5 \times 10^5$ co-cultured CD4+ naïve T cells were collected and washed with ice-cold sterile 0.9% saline 3 times and snap-frozen in liquid nitrogen. 1 mL of co-cultured supernatant was collected and stored in −80°C prior to sample submission. The input cell pellet was then extracted in 2 ml of 0.02 mg/L internal standard in 80% ice-cold methanol in dry ice. The lysate was then centrifuged at 14,000× *g* for 20 min at 4°C for protein precipitation. 1 ml of supernatant was dried under a gentle stream of nitrogen at room temperature. The dried aliquot was then re-suspended with 500 μl 50% acetonitrile. Each sample was vortexed for 10 s and centrifuged for 2 min at 14,000× *g*. The supernatant was dried under a gentle stream of nitrogen at room temperature and submitted to derivatization. The dried residue was redissolved and derivatized for 2 hours at 37 °C in 40 μl of methoxylamine hydrochloride (30 mg/ml in pyridine) followed by trimethylsilylation for 1 hour at 37 °C in 70 μl MSTFA with 1% TMCS. Up to 1 μl sample was injected for GC-MS/MS analysis.

Data analysis was performed using the Agilent MassHunter Workstation Quantitative Analysis Software (v B.04.xx). Linear calibration curves for each analyte were generated by plotting the peak area ratio of external/internal standard against standard concentration at different concentration levels. Analytes were confirmed by comparing the ratio of characteristic fragment ions in the sample and standard. The final intracellular concentration of targeted metabolites was normalized by the total number of input cells in each sample. The mass spectrometry results are provided in Supplementary Data 3.

### Electron microscopy scanning
CD4+ naïve T cells co-cultured with CD70-NC and CD70-KO C666 cells in the lipid-depleted or normal co-culture culture system after 3 days were washed, gently centrifuged, and re-suspended in PBS, then cells were fixed in 2.5% glutaraldehyde (Sigma-Aldrich), postfixed in 2% osmium tetroxide (Sigma-Aldrich), stained with uranyl acetate (Sigma-Aldrich), and dehydrated with a graded ethanol series. Sections were cut after embedding in Epon Araldite (Sigma-Aldrich), collected on uncoated nickel grids after sectioning and the mitochondria were observed and photographed using Philips CM-100 Transmission Electron Microscope. The sample preparation and electron microscopy scanning were performed by the Electron Microscope Unit at the University of Hong Kong.

### Seahorse real-time ATP rate assay
CD4+ naïve T cells were co-cultured with CD70-NC and CD70-KO C666 cells in the complete T cell culture system for 3 days. 16-hour prior to the assay, XFe96 sensor cartridge (Agilent) was hydrated with autoclaved water in a non-CO₂ 37 °C incubator overnight. 1-hour prior to the assay, the XFe96 cell culture plate (Agilent) was coated with 30 μL 0.1 mg/mL of poly-D-lysine (Gibco) for T cell adhesion. After 3-day co-culture, $1 \times 10^5$ CD4⁺ T cells were seeded into the cell culture plate with 200 μL Seahorse XF RPMI medium (Agilent) supplemented with

10 mM of XF glucose (Agilent), 1 mM of XF pyruvate (Agilent) and 2 mM of XF glutamine (Agilent) and incubated in a non-$CO_2$ 37 °C incubator for 60 mins. Subsequently, the T cells were washed with the complete Seahorse XF RPMI medium to a final volume of 180 μL. The XFe96 sensor cartridge was added with 200 μL Seahorse XFe96 Calibrant/well. Following the manufacturer's instructions on the Seahorse XF Real-Time ATP Rate Assay (Agilent), 20 μL diluted Oligomycin and 22 μL diluted Rotenone + Antimycin A were added into the injection port A and B, respectively. The XFe96 sensor cartridge covered with the injection plate was placed into the Agilent Seahorse XFe96 Analyzer for calibration. When the calibration was finished, replaced the utility plate containing the Seahorse XFe96 Calibrant with the cell culture plate. The ATP production rate in Tregs was dynamically measured by the Agilent Seahorse XFe96 Analyzer. ATP production via OXPHOS and glycolysis was evaluated based on the built-in analysis software.

### Evaluation of mitochondrial potential via JC-1 staining
CD4+ naïve T cells co-cultured with CD70-NC and CD70-KO C666 cells in lipid-depleted or normal culture medium after 3 days were stained with JC-1 dye (Sigma) for 20 mins in the dark at 37 °C, as per manufacturer's instructions. T cells were washed twice with cold washing buffer and centrifuged at 800× g for 5 min. After the final wash, T cells were re-suspend in 0.5 ml cold washing buffer for acquiring and analyzing the PE (JC-1 aggregates) and FITC (JC-1 monomers) fluorescence by ACEA Novocyte Quanteon.

### Establishment of the lipid-depleted co-culture system
For experiments conducted in the lipid-depleted co-culture system, the complete T cell medium was supplemented with 10% heat-inactivated lipid-depleted FBS (Neuromics) instead of regular FBS (Gibco). CD4+ naïve T cells, Treg and PBMC co-culture assays with CD70-NC and CD70-KO C666 cells were performed as described above.

### Assay for Transposase-Accessible Chromatin using sequencing on NPC and NPE cells
50,000 cells were used for each experiment. Transposase Tn5 enzymatic digestion was undergone at 37 °C for 30 mins using Illumina Tagment DNA Enzyme and Buffer Small Kit (Cat no. 20034197) following the protocol previously described. Sequencing was performed at 150 bp PE Illumina X-Ten platform. The ENCODE ATAC-seq data analysis standards and processing pipeline was followed, and TOBIAS was used for the motif analysis. The signals were adjusted and merged based on the conditions to estimate the log-transformed fold changes of the footprint scores for comparison among the EBV$^+$ NPC cells, EBV$^-$ NPC cells, and normal NPE cells. The differential motif analysis was performed based on the JASPAR database. The peaks were identified using the ENCODE analysis pipeline. The motif analysis using TOBIAS based on the JASPAR database was performed to evaluate the binding of the potential transcription factors.

### Site-directed mutagenesis and the dual-luciferase reporter assay
2000 bp (-1500 bp- 500 bp) of the promoter sequence of *CD70* were cloned into the pGL3-basic vector (Promega, E1751). For site-directed mutagenesis of the potential binding motif, the identified NFKB2 binding site (GGGGATTCCCC) on the *CD70* promoter was mutated with a QuikChange II site-directed mutagenesis kit (Stratagene). The mutation of the construct was confirmed by Sanger sequencing. 293FT cells were seeded in a 24-well plate at the density of 1×10$^5$/well. The luciferase reporter constructs (5:5:1 mixture of *CD70*-promoter-luc or *CD70*-promoter-mut-luc, NFKB2-OE or NFKB2-NC and Renilla plasmid (Promega)) were transfected together into 293FT cells using X-tremeGENE HP reagent (Roche) according to the manufacturer's instructions. pGL3-basic vector was used as the negative control. 48 hours

later, the culture supernatant was aspirated and replaced with 1× passive lysis buffer (Promega), the plate was then incubated at 4 °C for 20 mins for complete cell lysis. Subsequently, 40 μL cell lysate was mixed with 20 μL luciferase assay reagent (Promega) in the Lockwell maxisorp plate, and the firefly luminescence was immediately detected using the microplate reader. Dual-Glo® Stop & Glo® reagent (Promega) was added to the plate, mixed, and incubated for 3 mins before measuring the Renilla luminescence. The firefly luminescence was normalized to the Renilla signals.

### *NFKB2* knockdown via shRNA
To characterize the role of NFKB2 on transcriptional regulation of CD70 expression, we performed the small hairpin RNA (shRNA) knockdown in C666 cells. shRNA that specifically targeted *NFKB2* was designed, synthesized and inserted into the lentiviral vector pLKO.1. Lentivirus were packaged using 293FT cells as described above. C666 cells were infected with shNC and sh*NFKB2* lentivirus for 48 hours, supplemented with 10 μg/mL polybrene (Sigma). Following the lentivirus infection, puromycin was added in the cell culture medium for the selection and maintenance of shNC and *NFKB2*-knockdown C666 cells. *NFKB2*-targeting shRNA sequences used are listed in Supplementary Table 6.

### Statistical analysis and reproducibility
Statistical analysis was performed, and plots were generated using GraphPad Prism 8 and IBM SPSS Statistics 22. For data with a sample size ≤6, bar plots were used and all data points were shown. For data with a sample size >6, box & whiskers plots were used to visualize the data distribution. Bar plot data were presented as the mean values ± SD. Box & whiskers plots were presented as the median values ± IQR (25$^{th}$ and 75$^{th}$ percentiles) and whiskers encompass 1.5 times the IQR, as determined by the Tukey method. Violin plots were presented as kernel density estimations (KDE) annotated with median values ± IQR. For parametric data, $P$ values were evaluated by the two-sided Student's t-test. For non-parametric data from scRNA-seq data, $P$ values were evaluated by the two-sided Wilcoxon signed-rank test or Kruskal-Wallis one-way analysis of variance. Two-sided Pearson and Spearman correlation analysis was used to compute the correlation and statistical significance between two variables. In GSEA, both false discovery rates (FDR) and $P$ values were used to determine statistical significance. In survival analysis, $P$ values were evaluated by the two-sided log-rank test. The microscopic images shown in Figs. 2f, g, 4k, 6e, x and Supplementary Figs. 2g and 4a are representative for three independent experiments which generated similar results.

### Reporting summary
Further information on research design is available in the Nature Portfolio Reporting Summary linked to this article.

## Data availability
The NPC single-cell sequencing data used in the study are publicly available in Gene Expression Omnibus (GEO) under accession numbers GSE150825, GSE150430, and GSE162025. The NPC bulk RNA sequencing data used in the study are publicly available in GEO under accession numbers GSE68799, GSE102349, GSE53819, GSE13597, and GSE118719. The RNA sequencing data for time-resolved induced Treg differentiation used in the study is publicly available in GEO under accession number GSE96538. The HNSCC single-cell sequencing data used in the study are publicly available in GEO under accession numbers GSE139324 and GSE164690. The raw and processed Visium spatial sequencing data of primary NPC tissues and single-cell sequencing data of the co-cultured PBMCs have been deposited in GEO under

accession numbers GSE200310 and GSE200315. Source data are provided with this paper. The remaining data are available within the Article, Supplementary Information or Source Data file. Source data are provided with this paper.

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

## Acknowledgements

This work was supported by grants from the Hong Kong Research Grant Council (RGC), including Collaborative Research Funds (C7065-18GF, C7026-18GF, and C4039-19GF, X.G.), Research Impact Funds (R4017-18, R1020-18F and R7022-20, X.G.), Theme-based Research Funds (TRS (T12-703/22-R), X.G.), National Natural Science Foundation of China (81772554, X.G., 82072738, X.G., 82103479, L.G.), Natural Science Foundation of Guangdong Province (2023A1515010109, Z.Y.), Shenzhen Key Laboratory Program (ZDSYS20210623091811035, X.G.), The Shenzhen Science and Technology program (KQTD20180411185028798, X.G.) and the Program for Guangdong Introducing Innovative and Entrepreneurial Teams (2019BT02Y198, X.G.). Xin-Yuan Guan is the Sophie YM Chan Professor in Cancer Research.

## Author contributions

L.G., and X.G. designed the study. L.G., and X.G. wrote the manuscript. D.L.K., P.W., and S.L. supervised the sample collection. L.G., J.L., Z.Y., Y.Y., and X.F. performed the experiments. L.G., W.D., B.F., L.C., and D.C.

analyzed the data. J.L.H., and Y.W. collected the FFPE slides and performed the pathological examination. G.T. provided the valuable PDX. A.W.L., G.T., J.H., C.H., Q.L, L.B., and Y.T. participated in the discussion. X.G. supervised the study. All the authors have read and approved the manuscript.

## Competing interests

The authors declare no competing interests.

## Additional information

**Peer review information** : *Nature Communications* thanks Adrian Ochsenbein, Sander de Kivit, Qin Ma and the other, anonymous, reviewer(s) for their contribution to the peer review of this work. Peer reviewer reports are available.

[1]Department of Clinical Oncology, Li Ka Shing Faculty of Medicine, The University of Hong Kong, Hong Kong, China. [2]Department of Clinical Oncology, The University of Hong Kong-Shenzhen Hospital, Shenzhen, China. [3]Department of Pediatric Oncology, Sun Yat-sen University Cancer Center, Guangzhou, China. [4]State Key Laboratory of Oncology in South China, Guangdong Key Laboratory of Nasopharyngeal Carcinoma Diagnosis and Therapy, Sun Yat-sen University Cancer Center, Guangzhou, China. [5]Collaborative Innovation Center for Cancer Medicine, Sun Yat-sen University Cancer Center, Guangzhou, China. [6]Department of Surgery, The University of Hong Kong-Shenzhen Hospital, Shenzhen, China. [7]Department of Pathology, The University of Hong Kong-Shenzhen Hospital, Shenzhen, China. [8]School of Biomedical Sciences, Li Ka Shing Faculty of Medicine, The University of Hong Kong, Hong Kong, China. [9]Advanced Energy Science and Technology Guangdong Laboratory, Huizhou, China. [10]These authors contributed equally: Lanqi Gong, Jie Luo, Yu Zhang. ✉e-mail: xyguan@hku.hk

