## [Peer Review File · Nature Communications]

Nasopharyngeal carcinoma cells promote regulatory T cell development and suppressive activity via CD70-CD27 interactionReviewers' comments:

Reviewer #1 (Remarks to the Author): with expertise in nasopharyngeal carcinoma, cancer immunology

General comments:

1. The authors claim based on scRNAseq analyses that CD70 contribution in the TME is primarily by tumor cells. They never truly validate this claim by multi-color IF, Visium spatial profiling and/or flow cytometry. It is well known that CD70 can be expressed by various APC, like macrophages which are abundant in HNC.
2. At no point do authors really show DIRECT effect of CD70 on Treg differentiation. They need to treat naïve CD4 T cells with recombinant CD70 (or agonistic anti-CD27 antibody) + anti-CD3 antibody and see if they can induce Treg differentiation. Also, they cannot just show FoxP3 and CTLA4 expression as surrogate markers of Treg differentiation. These markers are also upregulated on transiently activated T cells. They need to prove Treg profile using functional assays.
3. Their in vitro killing assay is flawed. The authors are measuring cell death by annexin-V/7-AAD co-staining following a 48 h killing of adherent tumors. That means that they have to detach tumor cells prior to staining, which likely affects annexin-V/7-AAD staining. Furthermore, apoptosis is classically measured in 3-5 h killing assays. By 48 h, the authors should only detect necrosis. The authors need to use a different killing assay, one that is more tailored for attached cells (e.g. 3H-thymidine release, MTT or 51Cr release). Furthermore, the authors need to do a sequential stimulation, where they first induce CD4+ Treg, then collect and combine them with autologous CD8s and then measure whether there is suppression of tumor killing.
4. While the focus of this study is CD70-driven differentiation of CD4+ T cells, what is the impact of CD70 on CD8+ T cells? It was reported that 80% of EBV-specific CD8+ T cells express CD27+ memory phenotype (Hiroko Tomiyama, et al. J Immunol. 2002).
5. As their in vitro and in vivo assays where they use PBMC instead of sorted CD4s and CD8s, it is unclear if the effect they are observing is mediated directly or indirectly through another immune cell type (e.g. monocytes or B cells). More refined in vitro assays are needed.
6. For their humanized mouse models, the authors are activating PBMC for 3 days with CD3/CD28 before injecting the cells into mice. Why are they doing this? To allow for improved tissue infiltration?
7. In Fig. 3J the authors are showing increased frequencies of granzyme A+ and perforin+ tumor-infiltrating CD8+ T cells in mice inoculated with CD70 KO tumors and claim enhanced cytotoxic capacity. This is not proof of such; it may show that there was less degranulation by T cells in CD70 KO tumors. The authors should evaluate and show the following: a) overall level of immune infiltrate in these tumors by H&E [i.e. are CD70 KO tumors more inflamed], what is the actual immune infiltrate content of these tumors [e.g. CD4+ T cell FOXP3+ (Treg), CD4+ T cell FOXP3-, CD8+ T cell, NK cell, B cell and myeloid frequencies per mg of tumor] and whether tumor infiltrating T cell subsets have decreased expression of various checkpoint receptors in CD70 KO tumors.
8. For this study, the authors evaluated three different scRNAseq datasets generated in Chinese centers, which is why they ended up focusing on EBV-driven disease. Is CD70 also found in Western scRNAseq cohorts (e.g. Cillo Immunity 2019; Kurten Nat Commun 2021), where EBV is not a common cause of HNC?
9. Figure legends lack detail. How many times are experiments performed, which statistical tests are used to evaluate the data, etc.

Detailed comments:

It is a T cell centric paper and they appear to be ignoring other immune cell types. Given the role of CD27 in B cells and their importance in EBV infection did it seem odd, inappropriate and narrow that there was no study or profiling of the B cells in this manuscript and the only R:L pair analyzed was between Treg and NPC cells?

Two things that are quite noticeable ... 1. they did not validate CD70 protein expression by tumors. All their conclusions are based on scRNAseq. 2. They claim that they are inducing differentiation of naïve T cells into Treg using their IVS model based on CTLA4 and FoxP3 expression and not functional

validation. It is quite possible that in their co-culture system they are inducing transient T cell activation by allogeneic tumors.

Does the presence of Tregs correlate with the suppressive score calculated and independently show the same effect on survival?

MVA to prove the prognostic value of the suppressive score in addition to known pathologic variables commonly associated with poor prognosis.

In Fig. 1, the authors are combining three different datasets that is made up of 36 NPC tissues, 10 paired NPC peripheral blood samples and 4 INP tissues. Subsequently, they mix these unpaired samples for statistical analysis of cell fractions. Due to lack of matching specimens, not to mention substantial imbalance in the number of NPC (46), PBMC (10) and INP (4) specimens, this analysis is likely skewed.

Comments on the single cell and bulk RNASeq:

Survival analysis is Fig 1F: Since the survival analysis is done on bulk data, can the authors prove the presence of Tregs in this cohort using deconvolution methods?

In Figure 1J, Is CD70-CD27 interaction drive Treg development or CTLA4 expression on Treg? Is this sequential or concurrent event? Because CD70-CD27 interaction appear to promote CTLA4+ Treg population in direct co-culture experiment only. Both trans-well and direct co-culture show an increased Foxp3+ Treg. Moreover, would blocking CTLA4 and CD70 help minimize Treg suppression and improve anti-tumor immunity?

Figure 1I. NPC cell line C666 has been used throughout the paper. What is the expression level of CD70 in other human NPC cell lines? Are findings in Figure 1I, and Figure 4A verified in another NPC cell line?

R-L Analysis in Fig 2A &B:

This analysis needs to be done in each tissue type separately (tumor, normal, blood) if these came from separate samples. Could the authors clarify which cells were used in this analysis ?

Combining all the cells into one analysis is inaccurate (cells from Pt1 can show an interaction signal with cells from Pt2 which is not biologically possible). The authors should perform the analysis on cells from one patient at a time and report a summary of the results. This will also elaborate on the strength of the interaction from each patient and demonstrate reproducibility of the finding.

Figure 2E. Some studies show that CD70 can be expressed by human Tregs and T cells after prolonged stimulation and activation respectively (Arroyo et. al Nat Comm 2020, O'Neill et. al JImmunol 2017). In the scRNA data figure shown, authors only show CD70 expression in very general immune cell types. What about CD70 expression in the 3 different Treg clusters, as well as activated or effector T cells? It is possible that some of the CD70 expression is coming from activated T cells themselves that can act as a negative feedback loop.

From figure 2E-H, the correlation authors demonstrated in the paper are mostly based on RNAseq data. There's a lack of evidence in direct measurement of CD70+ tumor cells and CTLA4+Treg at protein level in NPC patients. Can author verify the positive correlation between CD70 expression and suppressive Treg proportion in patients via flow cytometry/or RT-qPCR?

Figure 2F. How were patients grouped into CD70high and CD70 low groups? Please add description/methods.

Can author provide a direct measurement of Treg activity? Such as IL-10, TGFb secretion from Treg after CD70 KO or antibody blockade experiments? CD8+ proliferation and effector molecule productions (such as level of granzyme A and perforin production) are indirect measuring potential Treg suppressive activity. The authors need to show direct evidence from Treg themselves.

Figure 2I&J. What is the normalized CD70 expression in dataset GSE102349 used to generate PFS

in Figure 2J?

In addition, what is the trend of PFS for the datasets used in Figure 2I in CD70^{high} vs CD70^{low} patients?

Figure 3: they do try to validate the lack of Treg differentiation, but it's a highly flawed killing assay (it's a 24-well co-culture system with tumors and PBMC; they perform a 48 h killing assay at the end of which they measure induction of apoptosis and necrosis by flow). They never describe how they detach tumor targets from the plates. They simply cannot effectively measure apoptosis in that system.

Figures 3E, 3F and Figure 4. Use of NPC cell line C666 with PBMC co-culture system as described in materials and methods. PBMC contains different immune cells in addition to T cells that can indirectly or directly influence results. It is recommended to repeat experiments using purified CD4 or CD8 T cells to remove possible effects of other immune cells in the system (similar to what was done in Figure 1).

Figure 3J, 4A, 4B, 4P, and 5B. Authors claim that knocking out or blocking CD70 in tumor cells in the NPC-humanized mouse model in Figures 3 and 4, "severely impaired immunosuppression". However, authors consistently show only lower frequencies/proportions of Tregs in TME and did not show any functional Treg immunosuppression assays. Does the Treg/CD8 T cell ratio differ between CD70-NC and CD70-KO tumors as well as cusatuzumab tx and untreated tumors? Is the anti-tumor effect of knocking down or blocking CD70 only due to less Tregs cells present in the tumor or is it also due to presence of dysfunctional Tregs (less immunosuppressive capacity of Tregs)? Please clarify such statements/conclusions accordingly in the manuscript.

In figure 4, the author showed CD70 and PD-1 inhibitors synergistically inhibit tumor growth in their humanized mice implanted with tumor cells/or NPC organoid. However, in the subsequent experiment and analysis. The authors performed single cell RNAseq on cells from the previous introduced co-culture system. In vivo mouse experiment may reflect a more comprehensive tumor microenvironment compared to co-cultured cell lines, so perform single cell on cells isolated from mouse tumor allows to interrogate the immune landscape from a relatively live tumor microenvironment. Can authors justify the advantage of using in vitro CRISPR KO cell lines over in vivo mouse experiments for single cell?

Figure 5. Data shown is based only on scRNA data. The authors did not show functional metabolic assays such as Seahorse data to support claims of reprogrammed metabolic profile differences (increased OXPHOS and FAO) in Tregs obtained from a CD70-rich TME vs CD70-KO TME.

Minor concern: the order of experimental groups needs to be consistent in the figures. For example, in Figure 5C&D, the order of CD70-NC and CD70-KO are switched in some comparisons

Figure 6D-6F. Data shown is based only on scRNA data. Gene expression does not necessarily correlate to protein expression and/or pathway activation. Generalizations and conclusions are not fully supported by data shown.

Figure 7I, can author provide experimental evidence that NFKB2 is driving CD70 expression in EBV+ cells? The experimental using luciferase reporter assay to demonstrate NFKB2 driving CD70 expression is good but doesn't mimic an EBV-infected cellular environment. It would be completed if observed NFKB2 activity in driving CD70 expression in EBV+ cells.

Reviewer #2 (Remarks to the Author): with expertise in regulatory T cells, CD70:CD27

Gong L. et. al. analyze in their established multi-center single-cell cohort T cell differentiation in samples of nasopharyngeal carcinoma (NPC) and provide indications for an increase in regulatory T cells with a suppressive function. In silico analysis of cell-cell interactions indicated a preferential role of the CD70-CD27 interaction in this process. They provide functional evidence in co-culture assays and in humanized mice that CD70 expressed on nasopharyngeal carcinoma cells induce suppressive

Tregs that in turn inhibit CD8+ T-cell function. Mechanistically, they provide evidence that CD70 knockout in NPC inhibits a lipid signaling network involving mitochondrial complex stability and fatty acid metabolism. In the last part, they provide evidence that NFkB2 binds to the CD70 promotor. This may explain the observed increased CD70 expression on EBV+ nasopharyngeal carcinoma. The strength of the study is a combination of single-cell sequencing data from patients and functional assays in vivo and in vitro. The findings support previous observations in mouse models that CD70-CD27 interaction induces regulatory T cells and inhibits the antitumor immune response. However, several major limitations limit my enthusiasm for the manuscript in its current form.

Major criticism

Induction of Tregs in vitro. A major limitation of the manuscript is the experimental system to induce regulatory T cells. PBMCs or CD4+ T cells are co-cultivated with allogenic tumor cells. This may not reflect the situation in vivo, where the differentiation of nTregs is induced by a TCR interaction with MHC class 1 on the tumor cell. Key experiments analyzing the role of CD70/CD27 signaling have to be repeated in a system using HLA-compatible CD4+ and cancer cells (preferentially CD4+ T cells and cancer cells from the same patient). Moreover, the system does not allow to distinguish between the differentiation of naïve T cells to regulatory T cells (as suggested by the authors) or an expansion of pre-existing regulatory T cells. Experiments with sorted naïve T cells have to be included.

CD70 blocking by Cusatuzumab. In Figure 4, Cusatuzumab is used to block the CD70-CD27 interaction. However, Cusatuzumab is an ADCC optimized antibody using Potelligent® technology. Thus, in addition to its blocking effect, Cusatuzumab activates FC gamma mediated ADCC by NK cells and monocytes in PBMC co-cultures and in the in vivo mouse model. Thus, the antitumor effect by Cusatuzumab may stem from ADCC mediated by NK cells or monocytes and not or not only, as suggested by the authors, by reduction of Treg cell formation.

Lipid signaling cause versus consequence. The experiments provided in Figure 5 and 6 suggest a role of lipid signaling in CD70/CD27 mediated induction of Treg cells. However, proliferation and expansion of cells always includes increased metabolic activity. The author should analyze the known CD27 signaling pathway including NFkB activation and analyze the cell cycling of Treg cells. Most probably, the increased metabolic activity reflects an activation of Treg cells with increased expansion and increased metabolic activity. Thus, the described lipid signaling is rather a consequence of the activation than the cause of the suppressive activity.

Regulation of CD70 expression. The data provided in Figure 7 documents that NFkB2 binds to the CD70 promotor. In addition, CD70 expression correlates with NFkB2 expression. However, there is no functional link provided. To conclude that EBV-mediated NFkB2 activation is the cause of the observed high expression of CD70 on nasopharyngeal carcinoma cells an experiment with NFkB pathway inhibition is needed.

Minor points

There are various typos and misspellings throughout the manuscript, i.e. quiescent Treg are misspelled in most figures.

Data on the expression of CD27 on Treg cells (FACS analysis) is missing. In addition, the expression of PD1 on tumor infiltrating CD8+ T cells and the expression of PD-L1 on tumor cells is missing.

The proposed classification of Treg cells into naïve Treg, quiescent Treg and suppressive Tregs is not a standard and it is not described in the given references 10 and 15 (line 106). If this classification is used throughout the paper, the functional effect of naïve Treg cells, quiescent Treg cells and suppressive Treg cells should be analyzed functionally.

The expression of CD70 on non-tumoral cells is ignored. Figure 2e clearly shows a higher expression of CD70 in tumor-infiltrating B and T cells than on tumor cells. In addition, Figure 2d documents a high CD70 expression in TILs. The limitation that the in vivo CD70 ligation might come from tumor cells or tumor infiltrating lymphocytes (and not only from cancer cells) should at least be discussed.

In Figure 3, co-cultivation experiments with purified CD4+ cells and PBMCs are provided. A similar experiment should be performed with purified CD8+ T cells. It is expected that CD27 signaling on CD8+ T cells might expand effector T cells.

Statistical tests should be included in all figure legends. For example, what statistical test has been used to compare tumor growth curves provided in figure 4o.

Reviewer #3 (Remarks to the Author): with expertise in regulatory T cells, metabolism

The manuscript by Guan et al. describes how CD70 expressed by nasopharyngeal carcinoma cells (NPC) drives Treg development. The authors have shown using multiple approaches and cutting-edge techniques that knock down of CD70 or anti-CD70 treatment can abrogate NPC-induced Treg induction and promote tumor eradication through CD8+ T cells. Mechanistically, the authors identified that CD27-CD70 signaling supports Treg function by activating lipid metabolism in these cells. Finally, it was shown that CD70 expression by NPC is driven by EBV-dependent chromatin remodelling.

This work is highly relevant to the field and may provide new insights into the mode of actions of targeting costimulatory molecules using antibody-based strategies to promote anti-tumor immunity. In particular, the results are of great importance, since many of these receptors are expressed both on Tregs and conventional CD4/CD8 cells and it is not clear how activation of these receptors affects their respective functions.

However, I have several major concerns that should be addressed:

- The authors classify Tregs based on naïve, quiescent and suppressive Tregs. To me, it is unclear how the authors come to this classification based on the presented analysis. Since these data are derived from RNA sequencing, the genes that are differentially expressed in each of these clusters need to be provided. To me, it seems like the authors distinguish Treg subsets based on effector status, rather than being 'quiescent' or 'suppressive'.

- Overall, the flow cytometry analyses are poor. The data provided showing that NPC drive Treg through CD27-70 signaling are not at all convincing. In particular, the expression of CTLA4 induced by NPC in the culture set up is extremely modest (if not absent), but is used to make the strong conclusion that CD70+ NPC, but not CD70- cells convert naïve CD4 T cells into the 'suppressive Treg' phenotype (identified as FOXP3+/CTLA4+ cells). The authors also draw the conclusion that direct cell-cell contact would be required for sTreg development. However, FOXP3 expression increases compared to T cell only cultures both in a transwell and a direct contact setting (both using NPC that do or do not express CD70). FOXP3 expression even seems to be more upregulated in a transwell setting rather than the cell contact setting, which is a strong indicator of Treg conversion. How can the authors exclude the possibility in this culture set up that soluble mediators derived from the tumor cells further upregulate FOXP3 expression (which would also imply that these cells become sTreg based on the trajectory analysis that shows FOXP3 expression increases during transition of nTregs to sTregs).

Furthermore, how are other key Treg molecules such as ICOS, TIGIT, GITR, 4-1BB, PD-1 affected in these cultures? These markers could have easily been addressed by flow cytometry. Also, do the cells indeed become more suppressive after coculture with NPC and can CD70 blockade or CD70-KO inhibit the suppressive activity of these cells? This is particularly important given the very modest changes in CTLA4 expression levels shown.

Also, the cultures were performed in the presence of TGF- β . Do NPC also induce Treg differentiation

without adding exogenous TGF- β to the culture? And does interfering in CD27-CD70 interaction still revert Treg induction? Additionally, what would be the effect of CD70 on NPC on natural/thymus-derived Tregs (CD25^{hi}CD127^{low} T cells) in terms of phenotype/suppressive activity instead of using TGF- β -polarizing cultures?

- In line with previous comment, the flow cytometric data regarding the enhanced CD8 T cell activation is very marginal, for instance in Figure 3E. Did the authors also analyse the expression of CD107a or the production of IFN- γ /TNF- α by the CD8 T cells? Moreover, in the organoid co-cultures, vehicle controls already contain >50% of dead tumor cells, which makes it questionable whether the effect caused by anti-CD70/anti-PD1 treatment is due to enhanced CD8-mediated killing. It would also be interesting to see in the in vivo xenotransplantation model whether CD70- NPC cells would be ignored when deliberate CD27 costimulation is provided using agonistic antibodies. In addition, authors make general statements based on their in vivo xenograft model, while melanoma cells for instance also express CD70 (line 276). Melanoma mouse models are well established, and antigen-specific responses can be assessed. The authors need to therefore show that similar phenomena occur in another well-established in vivo model. This would also link the present study to current literature stating that CD27 engagement on T cells has strong anti-tumor efficacy by promoting CD8 T cell priming, which authors do not touch upon in their discussion.

- The metabolic signature induced by CD27-CD70 interaction heavily relies on RNA sequencing data and only little on actual metabolomics analyses. Moreover, the metabolomics analysis is very limited. Differential metabolite levels and lipid species should at least be shown in an unbiased manner, for instance using volcano plots and/or a heatmap to evaluate the complete difference in the metabolome/lipidome in Tregs obtained from cocultures with CD70+ or CD70-KO NPC. This is particularly important since gene expression data do not always correlate with metabolic activity. Importantly, the authors should show that the observed differences are linked to the functionality of Tregs. Do Tregs cocultures with CD70+ NPC lose their suppressive function when lipid metabolism is impaired? Or does deliberate CD27 signaling in Tregs install a lipid metabolism program that supports their suppressive activity?

The authors make the bold statement that CD70-induced Tregs rely more on FAO-driven OxPHOS rather than glycolysis. This should at least be shown functionally using Seahorse analyses. Furthermore, authors claim that CD27-CD70 signaling promotes de novo fatty acid uptake, which could be shown by flow cytometry using fluorescent lipid analogs like BODIPY-C12/C16. Finally, it is not at all proven that FAO-driven OxPHOS affects mitochondrial stability, resulting in increased expression of STAT5A or TBX21. Authors only provide indirect proof based on RNA sequencing data. Similar to my previous comment, the authors do not provide any evidence that CD27-CD70 signaling in Tregs stabilizes mitochondrial integrity, for instance by evaluating mitochondrial morphology, mitochondrial mass/membrane potential using flow cytometric dyes and/or expression of ETC complex proteins. Furthermore, authors need to show that interfering in lipid metabolism instructed via CD27-CD70 signaling prevents the upregulation of key transcription factors, in order to conclude that CD70+ NPC provide a metabolic switch for Treg development and suppressive activity (which is the main conclusion of the study).

- The authors nicely demonstrate that EBV infection drives CD70 expression in NPC via NF- κ B2. However, it is unclear to this reviewer how authors conclude that NF- κ B2 was correlating the most with CD70 expression by NPC, especially since NF- κ B2 is absent from figure 7D. Also, it would be highly interesting to see in the luciferase reporter assay whether other NF- κ B signaling components identified in Figure 7D are less efficient in driving CD70 expression

- Overall, in many occasions, it is unclear how the data were statistically tested and what the sample size is. Also, some figures are unclear as figure legends are sometimes missing and similar color coding is used for different experimental conditions throughout the manuscript.

Reviewer #4 (Remarks to the Author): with expertise in transcriptomics

In this paper, the authors carried out a series of experiments to discover how tumor microenvironment characteristics alter immune homeostasis and immunotherapy efficacy in nasopharyngeal carcinoma (NPC). By analyzing single-cell RNA-seq data of 357,206 cells from 50 patient samples, they revealed that CD70-CD27 interaction can enhance NPC cell differentiation and suppress Tregs functions. They further validated the CD70's role in revitalizing CD8+ T-cell via CD70 knockout and anti-CD70 treatment, and inhibiting a collective lipid signaling network in Tregs. The ATAC-seq analysis proved transcriptionally regulation of CD70 via an Epstein-Barr virus (EBV)-dependent epigenetic modification in NPC cells. It is also interesting to see that the combination therapy of anti-CD70 and anti-PD-1 exhibits an improved tumor-killing efficacy compared to monotherapy. Overall, this manuscript targets a very interesting and significant topic regarding NPC microenvironment and immunotherapy. The manuscript is intact with solid evidence to support their results and conclusions. Nevertheless, there are several remaining concerns.

- In terms of deconvolution analysis of Visium data (Figure 2C), the author used CD70 expression to showcase CD70+ cancer cells distribution and then correlated it with Treg abundance to showcase co-localization of Treg and CD70+ cancer cells. Therefore, directly calculating cell type proportion for CD70+ tumor and Treg cells will be persuasive and intuitive. Refer to this paper (<https://www.nature.com/articles/s41592-022-01480-9>) and consider using cell2location to analyze cell proportion for identified cell types (CD70+ tumor and Treg cells). In addition, please provide more details in the method section regarding Visium analysis, including functions and parameters.
- Some terminologies should be clarified more clearly. For example, (1) spatial-seq is not a standard term for spatial transcriptomics. Therefore, authors may directly use Visium instead of spatial-seq in the manuscript. (2) Spatial voxel or spots? Visium's measuring units are spots rather than voxels (voxels usually were used for 3D spatial points).
- Line 175, I suppose it should be naïve CD4+ T cells?
- Monocle 3 should be cited somewhere relevant.
- Detailed descriptions of CellphoneDB and CellChat should be mentioned with parameter usage.

Reviewers' comments:

Reviewer #1 (Remarks to the Author): with expertise in nasopharyngeal carcinoma, cancer immunology

Dear reviewer,

We highly appreciate your comments and suggestions that have helped us improve the scientific soundness and novelty of our manuscript. We take your questions and advice very seriously, thus we have performed additional *in vitro/in vivo* experiments and computational analysis, and revised our manuscript accordingly. Below are the detailed revisions that we have made to address your concerns. Thank you very much again for your time and efforts.

General comments:

1. The authors claim based on scRNAseq analyses that CD70 contribution in the TME is primarily by tumor cells. They never truly validate this claim by multi-color IF, Visium spatial profiling and/or flow cytometry. It is well known that CD70 can be expressed by various APC, like macrophages which are abundant in HNC.

Reply: Thank you very much for your valuable suggestions. We evaluated the EPCAM⁺CD70⁺ fraction and the EPCAM⁻CD70⁺ fraction in additional primary NPC biopsies (61.1% vs. 13.8%, Figure 2E). We also performed IHC staining of CD70 on additional FFPE samples of primary NPC and inflammatory nasopharyngeal tissues (Figure 2F). These results validated that CD70 was highly expressed by NPC cells rather than immune cells in the TME or inflammatory microenvironment, which were consistent with our scRNA-seq analysis. Additional analysis of Visium spatial data and IHC/IF staining also confirmed a strong co-localization between NPC cells, CD70 expression and Tregs (Figures 2C, 2D, 2F and 2G). Unlike other head and neck malignancies, NPC does not have a high infiltration of DCs and macrophages in the TME, whereas T cells, especially Tregs, are the most abundant immune cells (Gong et al., 2021; Liu et al., 2021) (Figure 2G). Thus, we corroborated that NPC cells were the major source of CD70 in the TME.

2. *At no point do authors really show DIRECT effect of CD70 on Treg differentiation. They need to treat naïve CD4 T cells with recombinant CD70 (or agonistic anti-CD27 antibody) + anti-CD3 antibody and see if they can induce Treg differentiation. Also, they cannot just show FoxP3 and CTLA4 expression as surrogate markers of Treg differentiation. These markers are also upregulated on transiently activated T cells. They need to prove Treg profile using functional assays.*

Reply: Thank you very much for your feedback. To address your concern, we treated CD4⁺ naïve T cells with recombinant CD70 (ab271444, Abcam) + anti-CD3 antibody (40-0038-U100, TONBO) and subsequently evaluated FOXP3/CTLA4/CD27 expression via flow cytometry (Figures 3A to 3C) and CD27/IL-10/TGF-β/adenosine secretion via ELISA and mass spectrometry (Figures 3C and 3D). The new results in the revised manuscript showed that CD70 directly enhanced Treg differentiation, activation and suppressive activity via interacting with CD27.

3. *Their in vitro killing assay is flawed. The authors are measuring cell death by annexin-V/7-AAD co-staining following a 48 h killing of adherent tumors. That means that they have to detach tumor cells prior to staining, which likely affects annexin-V/7-AAD staining. Furthermore, apoptosis is classically measured in 3-5 h killing assays. By 48 h, the authors should only detect necrosis. The authors need to use a different killing assay, one that is more tailored for attached cells (e.g. 3H-thymidine release, MTT or 51Cr release). Furthermore, the authors need to do a sequential stimulation, where they first induce CD4+ Treg, then collect and combine them with autologous CD8s and then measure whether there is suppression of tumor killing.*

Reply: Thank you very much for pointing out the question. Prior to annexin V/7-AAD staining, we used TrypLE™ Express (Thermofisher, 12604013), a gentler substitute for Trypsin, to dissociate the viable adherent tumor cells from the cell culture plate at 37°C for 3 minutes. We used the same cell dissociation protocol for both CD70-NC and CD70-KO NPC cells, thus the influence of the dissociation process on annexin V/7-AAD staining was minimized. We agree with your comment stating that 24 and 48 hours might not be ideal for apoptosis measurement, but for necrosis measurement. In the original manuscript, we chose these time points to quantify cell death induced by PBMCs because lower than 12-hour co-culture (for example, 8-hour co-culture) did not yield an obvious cell death yet. Considering that our PBMC co-culture contained principally CD4⁺ and CD8⁺ T cells, the impaired T-cell tumor killing was primarily mediated by a higher fraction of activated Tregs during co-culture with CD70-NC C666 cells.

Consistent with your comment, the flow cytometry results in Figures 3J, 3K, 4E and 4F, show that most CD70-KO cells were 7-AAD⁺ cells, indicating that they were late apoptotic/necrotic cells. Hence, we have changed the term “apoptosis” into “tumor cell death” in the revised manuscript wherever applicable. Following your suggestion, we also performed XTT assays (Figures 3L and 4G) and nucleoside analog EdU staining to evaluate cell viability in the PBMC co-culture system, without interfering with adherent tumor cells in the culture plate.

To better confirm the enhanced cytotoxicity after CD70-KO, we also performed ELISAs to quantify secreted TNF- α and IFN- γ in the co-culture system to better characterize the anti-tumor cytotoxicity (Figures 3L, 3O, 4G and 4I).

4. While the focus of this study is CD70-driven differentiation of CD4⁺ T cells, what is the impact of CD70 on CD8⁺ T cells? It was reported that 80% of EBV-specific CD8⁺ T cells express CD27⁺ memory phenotype (Hiroko Tomiyama, et al. *J Immunol.* 2002).

Reply: We agree with you about evaluating the importance of CD70-CD27 interaction between NPC cells and CD8⁺ T cells. Recent advances demonstrate that the influence of CD70 on CD8⁺ T cells is contradictory, inducing T cell proliferation or exhaustion, due to a context-specific microenvironment and predominance of CD8⁺/CD27⁺ T cells, CD4⁺/CD27⁺ naïve T cells and Tregs (Flieswasser et al., 2022; O'Neill et al., 2017; van de Ven and Borst, 2015). In our tumor-PBMC co-culture system, we found that the proliferation and cytotoxicity of CD8⁺ T cells in co-cultured PBMCs were enhanced in the CD70-KO and cusatzumab-treated group after 48 hours, due to impaired Treg activity (Figures 3M to 3Q, 4H, 4I, and S4G). To address your concern, we evaluated whether co-culture between NPC cells and CD8⁺ T cells alone affected T cell proliferation, and showed that CD70 did not significantly affect CD8⁺ T cell proliferation after 48-hour co-culture, shown by the CFSE flow result at the lower right corner.

Moreover, we also found that CD70-responding PBMCs that we used throughout the study contained significantly more CD4⁺/CD27⁺ T cells compared to CD8⁺/CD27⁺ T cells (approximately a 3:1 ratio, Figures S3F and S3G). According to scRNA-seq analyses, the NPC microenvironment was also a CD4-enriched ecosystem and CD27 is preferentially expressed by infiltrating CD4⁺ T cells (Figures S3H and S3I). Thus, we think that CD70-CD27 interaction preferentially happens between NPC cells and CD4⁺ naïve T cells/Tregs in the TME.

5. As their *in vitro* and *in vivo* assays where they use PBMC instead of sorted CD4s and CD8s, it is unclear if the effect they are observing is mediated directly or indirectly through another immune cell type (e.g. monocytes or B cells). More refined *in vitro* assays are needed.

Reply: Thank you very much for pointing out the issue. In the revised manuscript, we evaluated the PBMC composition after 3-day activation and 2-day co-culture. Since we activated PBMCs using anti-CD3 antibodies and supplemented them with 25 IU/mL IL-2, more than 95% of PBMCs were CD4⁺ and CD8⁺ T cells (Figure S3F), whereas CD64⁺ monocytes and CD138⁺ B cells lacked. Additionally, as shown by scRNA-seq of co-cultured PBMCs, the fractions of myeloid cells and plasma B cells were only <0.5% and <5%, respectively (Figures 5A and 5B). Thus, in our NPC-PBMC co-culture system, immunosuppression and tumor-killing were overwhelmingly contributed by T cells.

6. For their humanized mouse models, the authors are activating PBMC for 3 days with CD3/CD28 before injecting the cells into mice. Why are they doing this? To allow for improved tissue infiltration?

Reply: Your understanding is correct. Based on our experience and previous publications that we referred to, the use of CD3/CD28-activated PBMCs has achieved a higher success rate for establishing humanized mice and better survival, expansion and tumor infiltration of T cells (Li and Kurlander, 2010; Qiu et al., 2021). We activated and expanded PBMCs using anti-CD3/anti-CD28 microbeads to enable a more robust activation and proliferation of T cells, since the establishment of humanized mice required a large number of healthy PBMCs (10⁷ PBMCs per mouse). The CD3/CD28 activated PBMCs (principally CD4⁺ and CD8⁺ T cells) could stably expand within the NSG mice after tail-vein injection and be re-stimulated by inoculated CD70⁺ tumors.

7. In Fig. 3J the authors are showing increased frequencies of granzyme A⁺ and perforin⁺ tumor-infiltrating CD8⁺ T cells in mice inoculated with CD70 KO tumors and claim

enhanced cytotoxic capacity. This is not proof of such; it may show that there was less degranulation by T cells in CD70 KO tumors. The authors should evaluate and show the following: a) overall level of immune infiltrate in these tumors by H&E [i.e. are CD70 KO tumors more inflamed], what is the actual immune infiltrate content of these tumors [e.g. CD4+ T cell FOXP3+ (Treg), CD4+ T cell FOXP3-, CD8+ T cell, NK cell, B cell and myeloid frequencies per mg of tumor] and whether tumor infiltrating T cell subsets have decreased expression of various checkpoint receptors in CD70 KO tumors.

Reply: We sincerely appreciate your constructive suggestions on our animal model. In the revised manuscript, we performed HE staining to evaluate the infiltration of total immune cells in CD70-NC and CD70-KO tumors, ELISAs on tumor lysates, and qRT-PCR to quantify mRNA changes in Treg-related genes. Based on H&E staining, there is no significant difference in the overall immune infiltration between CD70-NC and CD70-KO tumors (Figure S4A). However, compared to CD70-NC tumors, CD70-KO ones had a lower fraction of FOXP3⁺ Tregs (Figure 3T), lower concentrations of intratumoral immunosuppressive cytokines, including IL-10 and TGF- β (Figure 3U); higher concentrations of intratumoral cytotoxic cytokines, including TNF- α and IFN- γ (Figure 3U); lower expression of Treg lineage and activation-specific markers (Figure 3V); and decreased expression of PD-1 and TIM3 (Figures S4B, S4D and S4E). In addition, we also re-analyzed our flow cytometry results to compare the actual immune infiltrates (cell number/mg tumor), including FOXP3, granzyme A, perforin, PD-1 and TIM3, in CD70-NC and CD70-KO tumors (Figures S4C and S4E). These features indicated CD70-KO significantly impaired the Treg suppressive activity and thus enhanced anti-tumor cytotoxicity.

8. For this study, the authors evaluated three different scRNAseq datasets generated in Chinese centers, which is why they ended up focusing on EBV-driven disease. Is CD70 also found in Western scRNAseq cohorts (e.g. Cillo Immunity 2019; Kurten Nat Commun 2021), where EBV is not a common cause of HNC?

Reply: It is highly appreciated to have your suggestion on comparing the NPC microenvironment to other HNSCC microenvironments. Following your comment, we analyzed two HNSCC scRNA-seq cohorts generated by Cillo et al. (GSE139324) and Kurten et al. (GSE164690) (Figures S2M and S2N). scRNA analysis showed that the abundance of Tregs in the HNSCC microenvironment was not significantly different from that in the normal tonsil microenvironment (Figure S2O). We further demonstrated that the overall expression level of CD70 in HNC patients remains low, and only conventional DCs primarily and specifically expressed CD70 (Figure S2P). Unlike EBV infection in NPC, HPV might not drive CD70 upregulation in HNC cells, thus not enhancing the suppressive activity of Tregs in the TME. Besides, we performed survival analysis using HNS bulk-RNA seq data and it showed that CD70 expression was not correlated to worse prognosis in HNSCC patients either (Figure S2Q). Thus, these results demonstrate that CD70-driven immunosuppression is an NPC- and EBV-specific mechanism.

9. Figure legends lack detail. How many times are experiments performed, which statistical tests are used to evaluate the data, etc.

Reply: Thank you very much for pointing out the issue. We added more details in the figure legend and the method section to enhance the readiness of our manuscript. In the revised manuscript, we clearly stated how many biological independent experiments were performed and what kinds of statistical tests were used in each figure legend.

Detailed comments:

1. It is a T cell centric paper and they appear to be ignoring other immune cell types. Given the role of CD27 in B cells and their importance in EBV infection did it seem odd, inappropriate and narrow that there was no study or profiling of the B cells in this manuscript and the only R:L pair analyzed was between Treg and NPC cells?

Reply: Thank you very much for your comment. Indeed, we primarily focused on delineating the influence of NPC cells on Treg-mediated immunosuppression, which inhibited cytotoxicity of CD8⁺ T cell, because scRNA analyses suggested that Tregs played a central role in regulating immunosuppression in NPC. We agree with the high importance on investigating the effect of CD70 on CD27⁺ B cells. In this study, we developed a PBMC-engrafted humanized mouse model containing a T-cell-enriched immune system which hindered our further investigations on the interaction between NPC cells and B cells. Considering the lack of immunological studies on NPC in the past decade, owing to the incapability of EBV infection in rodents and the lack of murine NPC cell lines and spontaneous mouse models, our current PBMC-engrafted humanized mouse model has comprehensively demonstrated one key aspect of the immunosuppression mechanism in NPC and will inspire future immunological studies conducted by our peers. We consider your advice is of high importance for further delineating the pro-tumor effect of CD70-CD27 interaction in the NPC microenvironment, and facilitating clinical translation of anti-

CD70 therapy in NPC patients. Thus, we are currently focusing on developing CD34⁺ hematopoietic stem cells-engrafted mouse model with a more competent immune system to study the effects of CD70⁺ NPC cells on other immune subtypes, particularly B cells.

2. Two things that are quite noticeable ... 1. they did not validate CD70 protein expression by tumors. All their conclusions are based on scRNAseq. 2. They claim that they are inducing differentiation of naive T cells into Treg using their IVS model based on CTLA4 and FoxP3 expression and not functional validation. It is quite possible that in their co-culture system they are inducing transient T cell activation by allogeneic tumors.

Reply: Thank you very much for pointing out the issue. As you suggested, we performed flow cytometry on cells dissociated from primary NPC tissues, showing more than 60% of EPCAM⁺ cells are CD70⁺ (Figure 2E). We also performed additional IHC staining and spatial profiling that verified the expression of CD70 on NPC cells in the TME (Figures 2C and 2F).

Besides, to further validate the enhanced Treg activity induced by CD70-CD27 interaction, we performed additional ELISAs and mass spectrometry to quantify the secreted IL-10, TGF- β and adenosine in our *in vitro* co-culture systems (NPC vs. NPE; IgG vs. recombinant CD70; CD70-NC vs. CD70-KO; and IgG vs. cusatuzumab) and humanized mouse model (Figures 1N, 3D, 3H, 3U and 4D). We also performed additional qRT-PCR and flow analysis to evaluate the *in vitro* and *in vivo* changes of Treg lineage-and activation-specific markers (Figures 1M, 3I, 3V, S3B and S3C). These results altogether validated the functional profile of induced Treg differentiation and activation by CD70⁺ NPC cells in all of our experimental conditions.

3. Does the presence of Tregs correlate with the suppressive score calculated and independently show the same effect on survival?

Reply: Thank you very much for your comment. To address your concern, we performed CIBERSORTx deconvolution to estimate the Treg abundance from NPC bulk RNA-seq data (Figure S1J), using our scRNA signature matrix as the reference. The Treg abundance was highly correlated to the Treg suppressive score calculated by our scRNA-seq linear model (Figure S1H), and independently correlated to inferior prognosis in NPC patients (Figure S1I).

4. MVA to prove the prognostic value of the suppressive score in addition to known pathologic variables commonly associated with poor prognosis.

Reply: According to your suggestion, we examined the association between the Treg suppressive score and the tumor stage, age, and sex that are available in GSE68799 (Table S2).

Variables	Treg suppressive score		Statistics	
	Low	High	Chi-square	p-value
Stage				
II	4 (66.7%)	2 (33.3%)		
III	16 (44.4%)	20 (55.6%)	1.018	0.400
Age				
>60	14 (48.3%)	15 (51.7%)		
<60	6 (46.2%)	7 (53.8%)	0.016	1.000
Sex				
Female	4 (44.4%)	5 (55.6%)		
Male	16 (48.5%)	17 (51.5%)	-0.033	1.000

5. In Fig. 1, the authors are combining three different datasets that is made up of 36 NPC tissues, 10 paired NPC peripheral blood samples and 4 INP tissues. Subsequently, they mix these unpaired samples for statistical analysis of cell fractions. Due to lack of matching specimens, not to mention substantial imbalance in the number of NPC (46), PBMC (10) and INP (4) specimens, this analysis is likely skewed.

Reply: Following your question, we normalized T cell fractions within each sample and subsequently performed the abundance analysis on Tregs and CD4⁺ naïve T cells (Figure 1 D). After cross-sample normalization, the influence of imbalanced sample numbers on drawing the statistical significance was minimized. In addition, normal nasopharyngeal tissue samples are very rare and valuable, so we only obtained four samples collected by three independent cohorts. Higher Treg infiltration in NPC was validated by IF staining of primary NPC tissues and non-malignant inflammatory nasopharyngeal tissues in our previous paper (Gong et al., 2021). Thus, we are confident that the high Treg infiltration is an NPC-specific characteristic.

Comments on the single cell and bulk RNASeq:

1. Survival analysis is Fig 1F: Since the survival analysis is done on bulk data, can the authors prove the presence of Tregs in this cohort using deconvolution methods?

Reply: As you suggested, we performed CIBERSORTx deconvolution to bulk RNA-seq data (GSE102349) using the NPC scRNA signatures as the reference and performed additional survival analysis based on Treg abundance accordingly (Figures S1J to S1L).

2. In Figure 1J, Is CD70-CD27 interaction drive Treg development or CTLA4 expression on Treg? Is this sequential or concurrent event? Because CD70-CD27 interaction appear to promote CTLA4+ Treg population in direct co-culture experiment only. Both trans-well and direct co-culture show an increased Foxp3+ Treg. Moreover, would blocking CTLA4 and CD70 help minimize Treg suppression and improve anti-tumor immunity?

Reply: Thank you very much for your comments. Our study showed that CD70-CD27 enhanced Treg differentiation and activation sequentially because we performed co-culture assays using CD4⁺ naïve T cells initially, which required to differentiate into Tregs first prior to subsequent Treg activation. Only direct co-culture with NPC cells increased the frequency of FOXP3⁺ and FOXP3⁺/CTLA4⁺ Tregs (Figures 1J to 1N). The difference in Treg frequencies between transwell co-culture and direct co-culture was due to different experimental settings, such as a total number of CD4⁺ naïve T cells seeded in the 0.3 cm² transwell insert and 1.9 cm² culture plate, as well as the use of CD4⁺ naïve T cells isolated from different donors for transwell/direct co-culture. Thus, the fraction of FOXP3⁺ Tregs was not comparable between the transwell and direct settings. We are sorry for the misleading information in the figure, so we modified and provided a sufficient explanation in the revised manuscript.

We performed several times of the transwell co-culture assay using different effector: target ratios and generated consistent results showing that NPC cells did not significantly enhance Treg differentiation or Treg activation in transwell co-culture. Hence, we determined to perform the direct co-culture assay and found NPC cells significantly enhanced Treg differentiation and Treg activation via direct cell-cell contact.

In our study, we showed that blocking CD70 using cusatuzumab inhibited the suppressive activity of Tregs and revigorated anti-tumor immunity in our *in vitro* co-culture

systems synergizing with PD-1 blockade (Figures 4A to 4N).

3. Figure 11. NPC cell line C666 has been used throughout the paper. What is the expression level of CD70 in other human NPC cell lines? Are findings in Figure 11, and Figure 4A verified in another NPC cell line?

Reply: Thank you for your questions. C666 is the most widely used cell line for authentic undifferentiated EBV⁺ NPC in NPC-studying labs worldwide, because most of the NPC cell lines, including CNE1, CNE2, SUNE1, SUNE2 and 5-8F, have been reported with Hela cell contamination. We were recently provided by Prof. George Tsao from The University of Hong Kong, School of Biomedical Sciences, with another authentic NPC cell line: NPC43, with/without EBV infection. Compared to EBV⁻ NPC43 cells, EBV⁺ NPC43 cells had a high CD70⁺ fraction that was comparable to that in C666 (Figure 7A). A consistent CD70-high cell fraction was also observed in NPC PDX, Xeno76 (Figure 4J).

However, NPC43 cells were not ideal for co-culture assays because they were very easy to become unhealthy during co-culture with CD4⁺ naïve T cells and PBMCs. Instead, we tried to verify our findings in Figures 1I and 4A by co-culturing primary NPC cells with autologous PBMCs, treated with IgG or cusatuzumab, which showed a consistent result as cusatuzumab treatment in the C666-PBMC co-culture system (Figure S4J to S4M).

R-L Analysis in Fig 2A &B:

1. This analysis needs to be done in each tissue type separately (tumor, normal, blood) if these came from separate samples. Could the authors clarify which cells were used in this analysis ?

Reply: Because we are interested in delineating the immunosuppressive mechanism in the TME, we conducted the CellphoneDB and CellChat analyses based on cells selected from NPC patients. In the revised manuscript, we compared the signaling changes in Treg subtypes, particularly eTregs, derived from the TME, NPC peripheral blood and non-malignant nasopharyngeal inflammatory tissues (Figures S2B to S2D).

2. Combining all the cells into one analysis is inaccurate (cells from Pt1 can show an interaction signal with cells from Pt2 which is not biologically possible). The authors should perform the analysis on cells from one patient at a time and report a summary of the results. This will also elaborate on the strength of the interaction from each patient and demonstrate reproducibility of the finding.

Reply: Thank you very much for your suggestion, and we totally agree with your opinion that cell-cell interaction analyses performed on each patient sample individually might be more convincing. Nevertheless, one major limitation of scRNA-seq for NPC is the low tumor cell recovery rate, because NPC cells are highly vulnerable after endoscopic biopsy and immune cells are highly overnumbered the NPC cells in tissues. Therefore, we and the other two groups (Liu Y et al., 2021, and YP Chen et al., 2020, whose data were integrated in our study) did not get sufficient tumor cells from each of the NPC patient samples. The

only option for us was to combine all tumor cells and identified significant ligand-receptor pairs with CD4⁺ naïve T cells and Tregs accordingly. Although it might not be biologically convincing, it at least provided us with sufficient hints for us to perform the downstream analyses and we successfully validated the importance of CD70-CD27 interaction between NPC and immunosuppressive Tregs via functional assays. To better address your concern, we performed additional cell-cell interaction analyses based on cells extracted from NPC tissues, inflammatory nasopharyngeal tissues and NPC periphery. The results showed NPC-infiltrated Tregs, especially eTregs, had significantly higher overall interaction strength compared to Tregs infiltrated in the inflammatory tissues and peripheral blood of NPC patients (Figure S2B). Furthermore, we compared the signaling changes in eTregs from three sample types, showing that CD70-CD27 was the most significant incoming signal in NPC-infiltrated eTregs (Figures S2C and S2D).

3. *Figure 2E. Some studies show that CD70 can be expressed by human Tregs and T cells after prolonged stimulation and activation respectively (Arroyo et. al Nat Comm 2020, O'Neill et. al JImmunol 2017). In the scRNA data figure shown, authors only show CD70 expression in very general immune cell types. What about CD70 expression in the 3 different Treg clusters, as well as activated or effector T cells? It is possible that some of the CD70 expression is coming from activated T cells themselves that can act as a negative feedback loop.*

Reply: Thank you very much for raising this important question. scRNA-seq analyses illustrated that CD70 expression in three Treg subtypes, effector T cells and exhausted T cells was very low (Supplementary figure 3J). Additional flow cytometry was also performed to validate the low CD70 protein expressed by CD4⁺ and CD8⁺ T cells in the co-culture system (Supplementary figures 3K and 3L). Thus, NPC cells were verified as the major contributor of CD70 in the physiological NPC TME and our co-culture systems.

4. *From figure 2E-H, the correlation authors demonstrated in the paper are mostly based on RNAseq data. There's a lack of evidence in direct measurement of CD70+ tumor cells and CTLA4+Treg at protein level in NPC patients. Can author verify the positive*

correlation between CD70 expression and suppressive Treg proportion in patients via flow cytometry/or RT-qPCR?

Reply: As you suggested, we collected additional primary NPC biopsies and performed flow cytometry to quantify the frequency of EPCAM⁺/CD70⁺ in each sample. Subsequently, we stratified the samples into CD70-high and CD70-low groups and compared *FOXP3*, *IL2RA* and *CTLA4* expression via RT-qPCR by extracting mRNA from the leftover tissues (Figure S2H).

5. *Figure 2F. How were patients grouped into CD70high and CD70 low groups? Please add description/methods.*

Reply: The NPC patients were stratified into CD70-high and CD70-low groups based on the median CD70 expression of tumor cells in the integrated scRNA-seq cohort.

6. *Can author provide a direct measurement of Treg activity? Such as IL-10, TGFβ secretion from Treg after CD70 KO or antibody blockade experiments? CD8+ proliferation and effector molecule productions (such as level of granzyme A and perforin production) are indirect measuring potential Treg suppressive activity. The authors need to show direct evidence from Treg themselves.*

Reply: Following your suggestion, we performed ELISAs (IL-10 and TGF-β) and mass spectrometry (adenosine) in the co-culture system and tumor lysates from humanized mice to evaluate the suppressive activity of Tregs after CD70-KO and cusatuzumab treatment in C666 cells. The results showed CD70-KO and blockade significantly inhibited the secretion of immunosuppressive factors from Tregs (Figures 3D, 3H, 3U and 4D).

7. *Figure 2I&J. What is the normalized CD70 expression in dataset GSE102349 used to*

generate PFS in Figure 2J? In addition, what is the trend of PFS for the datasets used in Figure 2I in CD70^{high} vs CD70^{low} patients?

Reply: Cross-sample normalization was performed on each gene expression in NPC bulk RNA-seq data (GSE102349) to enable reliable survival analyses stratified based on gene expression levels and module scores. The normalized expression matrix for each NPC patient was uploaded as source data along with the manuscript. The CD70-high and CD70-low patients used in Figure 2I were stratified based on the median CD70 expression of tumor cells from our integrated scRNA-seq cohort. For the NPC patients enrolled in scRNA-seq studies, we do not have enough prognosis information so far, because most of them were diagnosed in late 2019 and the follow-up time is still not sufficiently long. According to our latest follow-up in September, 2022, only one patient relapsed after receiving chemo-radiotherapy.

8. *Figure 3: they do try to validate the lack of Treg differentiation, but it's a highly flawed killing assay (it's a 24-well co-culture system with tumors and PBMC; they perform a 48 h killing assay at the end of which they measure induction of apoptosis and necrosis by flow). They never describe how they detach tumor targets from the plates. They simply cannot effectively measure apoptosis in that system.*

Reply: Thank you very much for pointing out the question. Prior to annexin V/7-AAD staining, we used TrypLE™ Express (ThermoFisher, 12604013), a gentler substitute for Trypsin, to dissociate the viable adherent tumor cells from the cell culture plate at 37°C for 3 minutes. We used the same cell dissociation protocol for both CD70-NC and CD70-KO NPC cells, thus the influence of the dissociation process on annexin V/7-AAD staining was minimized. We agree with your comment stating that 24 and 48 hours might not be ideal for apoptosis measurement, but for necrosis measurement. In the original manuscript, we chose these time points to quantify cell death induced by PBMCs because lower than 12-hour co-culture (for example, 8-hour co-culture) did not yield an obvious cell death yet. Considering that our PBMC co-culture contained principally CD4⁺ and CD8⁺ T cells, the impaired T-cell tumor killing was primarily mediated by a higher fraction of activated Tregs during co-culture with CD70-NC C666 cells.

Consistent with your comment, the flow cytometry results in Figures 3J, 3K, 4E and 4F, show that most CD70-KO cells were 7-AAD⁺ cells, indicating that they were late apoptotic/necrotic cells. Hence, we have changed the term “apoptosis” into “tumor cell death” in the revised manuscript wherever applicable. Following your suggestion, we also performed XTT assays (Figures 3L and 4G) and nucleoside analog EdU staining to evaluate cell viability in the PBMC co-culture system, without interfering with adherent tumor cells in the culture plate.

To better confirm the enhanced cytotoxicity after CD70-KO, we also performed ELISAs to quantify secreted TNF- α and IFN- γ in the PBMC co-culture system to better characterize the anti-tumor cytotoxicity (Figures 3L, 3O, 4G and 4I).

9. Figures 3E, 3F and Figure 4. Use of NPC cell line C666 with PBMC co-culture system as described in materials and methods. PBMC contains different immune cells in addition to T cells that can indirectly or directly influence results. It is recommended to repeat experiments using purified CD4 or CD8 T cells to remove possible effects of other immune cells in the system (similar to what was done in Figure 1).

Reply: Thank you very much for pointing out the issue. In the revised manuscript, we evaluated the PBMC composition after 3-day activation and 2-day co-culture. Since we activated PBMCs using anti-CD3 antibodies and supplemented them with 25 IU/mL IL-2, more than 95% of PBMCs are CD4⁺ and CD8⁺ T cells (Figure S3F), whereas CD64⁺ monocytes and CD138⁺ B cells lacked. Additionally, as shown by scRNA-seq of co-cultured PBMCs, the fractions of myeloid cells and plasma B cells were only <0.5% and <5%, respectively (Figures 5A and 5B). Thus, in our NPC-PBMC co-culture system, immunosuppression and tumor killing was overwhelmingly contributed by T cells.

10. Figure 3J, 4A, 4B, 4P, and 5B. Authors claim that knocking out or blocking CD70 in tumor cells in the NPC-humanized mouse model in Figures 3 and 4, “severely impaired immunosuppression”. However, authors consistently show only lower frequencies/proportions of Tregs in TME and did not show any functional Treg immunosuppression assays. Does the Treg/CD8 T cell ratio differ between CD70-NC and CD70-KO tumors as well as cusatuzumab tx and untreated tumors? Is the anti-tumor effect of knocking down or blocking CD70 only due to less Tregs cells present in the tumor or is it also due to presence of dysfunctional Tregs (less immunosuppressive capacity of Tregs)? Please clarify such statements/conclusions accordingly in the manuscript.

Reply: Thank you very much for your comments on our animal model. As you suggested, first, we performed ELISA of IL-10, TGF- β , TNF- α and IFN- γ on CD70-NC and CD70-KO tumor lysates to evaluate immunosuppression and cytotoxicity in humanized mice (Figures 3H, 3O, 3U, 4D and 4I). We also extracted RNA from these tumors and performed qRT-PCR to determine the changes in Treg-specific markers (Figures 3V). Overall, these results validated impaired immunosuppression and enhanced anti-tumor immunity in CD70-KO tumors grown on the PBMC-engrafted humanized mice. Knockout and blocking of CD70 decreased Treg fraction in tumors and subsequently impeded immunosuppressive activity and enhanced anti-tumor immunity. Following your suggestions, we clarified the above-mentioned statements accordingly in the revised manuscript.

11. In figure 4, the author showed CD70 and PD-1 inhibitors synergistically inhibit tumor growth in their humanized mice implanted with tumor cells/or NPC organoid. However, in the subsequent experiment and analysis. The authors performed single cell RNAseq on cells from the previous introduced co-culture system. In vivo mouse experiment may reflect a more comprehensive tumor microenvironment compared to co-cultured cell lines, so perform single cell on cells isolated from mouse tumor allows to interrogate the immune landscape from a relatively live tumor microenvironment. Can authors justify the advantage of using in vitro CRISPR KO cell lines over in vivo mouse experiments for single cell?

Reply: We chose to perform single-cell sequencing on *in vitro* co-cultured PBMCs because the *in vitro* co-culture settings are much easier to be controlled and the *in vivo* models might have high inter-mouse heterogeneity on T cell subpopulations. We can easily determine a good co-culture timing and condition for *in vitro* PBMC scRNA sequencing based on immunophenotyping and killing assays, to generate consistent results using biologically independent samples. Meanwhile, we agree with your comments stating that scRNA-seq on tumor-infiltrating immune cells might be superior. To avoid a T-cell-enriched humanized immune system and enable a more comprehensive investigation of the effect of CD70 blockade, we are currently developing a CD34⁺ hematopoietic stem cell-engrafted humanized mouse model, which contains T cells, B cells and myeloid cells in the periphery. In our next study, the anti-CD70/anti-PD-1 treatment will be tested on this mouse model in the future and further decipher the microenvironmental characteristics associated to responsiveness to CD70 blockade.

12. Figure 5. Data shown is based only on scRNA data. The authors did not show functional metabolic assays such as Seahorse data to support claims of reprogrammed metabolic profile differences (increased OXHPOS and FAO) in Tregs obtained from a CD70-rich TME vs CD70-KO TME.

Reply: Following your suggestions on the metabolomics part, we performed Oil O staining, the Seahorse assay, fatty acid oxidation assay, mitochondria potential assay, and determined the mitochondrial morphology via electron microscopy, on Tregs co-cultured with CD70-

NC and CD70-KO C666 cells (Figures 6A, 6E to 6H). These results exhibited that CD70-CD27 interaction significantly enhanced intracellular lipid storage, OXPHOS, FAO and mitochondrial functions in co-cultured Tregs.

We also showed that lipid signaling was indispensable for the downstream effect of CD70-CD27 interaction to take place in Tregs, using the lipid-depleted medium for co-culture and functional assays (Figures 6P to 6X).

Minor concern:

1. The order of experimental groups needs to be consistent in the figures. For example, in Figure 5C&D, the order of CD70-NC and CD70-KO are switched in some comparisons

Reply: The order of experimental groups was changed in the revised manuscript according to your feedback.

2. Figure 6D-6F. Data shown is based only on scRNA data. Gene expression does not necessarily correlate to protein expression and/or pathway activation. Generalizations and conclusions are not fully supported by data shown.

Reply: We performed additional flow cytometry and metabolomics assays in the revised manuscript to corroborate the findings based on our scRNA-seq data, shown in Figures 6A to 6X.

3. *Figure 7I, can author provide experimental evidence that NFKB2 is driving CD70 expression in EBV+ cells? The experimental using luciferase reporter assay to demonstrate NFKB2 driving CD70 expression is good but doesn't mimic an EBV-infected cellular environment. It would be completed if observed NFKB2 activity in driving CD70 expression in EBV+ cells.*

Reply: Thank you very much for your suggestions. We performed NFKB2 knockdown in C666 cells, which inhibited transcription of CD70 mRNA shown by qRT-PCR and decreased the fraction of CD70⁺ cells shown by flow cytometry analysis (Figures 7I-7K). Besides, we treated C666 cells with an NFKB2 inhibitor: SN-52 (HY-P3229, MCE), which hampered the transcriptional activity of NFKB2 by inhibiting its nuclear translocation. The use of SN-52 exhibited a consistent effect on inhibition of CD70 mRNA and protein levels, compared to NFKB2 knockdown (Figures 7I-7K). Both NFKB2 knockdown and inhibition promoted the sensitivity of NPC cells to anti-tumor immunity (Figures 7L and 7M).

Reviewer #2 (Remarks to the Author): with expertise in regulatory T cells, CD70:CD27

Gong L. et. al. analyze in their established multi-center single-cell cohort T cell differentiation in samples of nasopharyngeal carcinoma (NPC) and provide indications for an increase in regulatory T cells with a suppressive function. In silico analysis of cell-cell interactions indicated a preferential role of the CD70-CD27 interaction in this process. They provide functional evidence in co-culture assays and in humanized mice that CD70 expressed on nasopharyngeal carcinoma cells induce suppressive Tregs that in turn inhibit CD8+ T-cell function. Mechanistically, they provide evidence that CD70 knockout in NPC inhibits a lipid signaling network involving mitochondrial complex stability and fatty acid metabolism. In the last part, they provide evidence that NFkB2 binds to the CD70 promotor. This may explain the observed increased CD70 expression on EBV+ nasopharyngeal carcinoma.

The strength of the study is a combination of single-cell sequencing data from patients and functional assays in vivo and in vitro. The findings support previous observations in mouse models that CD70-CD27 interaction induces regulatory T cells and inhibits the antitumor immune response. However, several major limitations limit my enthusiasm for the manuscript in its current form.

Dear reviewer,

We would like to thank you for your support of our findings, as well as your valuable feedback that leads to our revised manuscript with additional experimental results and computational analyses to further address the functional, metabolic and clinical importance of the CD70-CD27 axis between NPC cells and Tregs. You can look at our detailed replies below and determine whether our revision sufficiently addresses your concerns. Thank you very much again and we are looking forward to hearing from you.

Major criticism

- 1. Induction of Tregs in vitro. A major limitation of the manuscript is the experimental system to induce regulatory T cells. PBMCs or CD4+ T cells are co-cultivated with allogenic tumor cells. This may not reflect the situation in vivo, where the differentiation of nTregs is induced by ae TCR interaction with MHC class I on the tumor cell. Key experiments analyzing the role of CD70/CD27 signaling have to be repeted in a system using HLA-compatible CD4+ and cancer cells (preferentially CD4+ T cells and cancer cells from the same patient). Moreover, the system does not allow to distinguish between the differentiation of naïve T cells to regulatory T cells (as suggested by the authors) or an expansion of pre-existing regulatory T cells. Experiments with sorted naïve T cells have to be included.*

Reply: Thank you very much for your suggestions. For the co-culture assays (Figure 1I), we used CD4⁺ naïve T cells sorted from fresh PBMCs of healthy donors using the Naive CD4⁺ T Cell Isolation Kit II, human (Miltenyi, 130-094-131). Subsequently, the TCR complex of CD4⁺ naïve T cells was activated by the anti-CD3 antibody that mimicked the antigen-presenting process, prior to induction of Treg differentiation. We followed and

optimized the previously established protocols in Nature Protocols for the generation of induced Tregs (iTregs, which were genetically and functionally similar to natural thymus-derived Tregs) from CD4⁺ naïve T cells (Ellis et al., 2012; Fantini et al., 2007). In our co-culture system, CD70⁺ NPC cells only served as a signaling provider for inducing CD27 co-stimulation. In our revised manuscript, we comprehensively validated the importance of CD70-CD27 interaction on Treg differentiation and activation by varied sequencing techniques on clinical samples from multiple cohorts and cell/animal samples from our *in vitro* and *in vivo* functional assays.

Following your suggestion on using allogeneic T cells and NPC cells from the same patient, we isolated and cultured primary NPC cells collected from additional NPC biopsies. Because primary NPC cells are very difficult to culture *in vitro* and the volume of patient peripheral blood is limited, we only successfully co-cultured NPC cells from one patient with the paired PBMCs (inadequate volume for CD4⁺ naïve T cell isolation) for 2 days post-TCR activation, with the treatment of cusatuzumab and IgG. Flow cytometry results showed lower FOXP3⁺ and FOXP3⁺/CTLA4⁺ Treg fractions, higher tumor death, lower concentrations of immunosuppressive cytokine, and higher concentrations of cytotoxic cytokines in the cusatuzumab-treated co-culture system (Figures S4J to S4M).

2. *CD70 blocking by Cusatuzumab.* In Figure 4, Cusatuzumab is used to block the CD70-CD27 interaction. However, Cusatuzumab is an ADCC optimized antibody using Potelligent® technology. Thus, in addition to its blocking effect, Cusatuzumab activates FC gamma mediated ADCC by NK cells and monocytes in PBMC co-cultures and in the *in vivo* mouse model. Thus, the antitumor effect by Cusatuzumab may stem from ADCC mediated by NK cells or monocytes and not or not only, as suggested by the authors, by reduction of Treg cell formation.

Reply: Thank you very much for pointing out this issue. In our co-culture system, only T cells in PBMCs were activated and stimulated for 72 hours and co-cultured with NPC cells with supplemented IL-2 for 48 hours. Thus, after TCR activation and co-culture, more than 95% of PBMCs are CD4⁺ and CD8⁺ T cells, shown by flow cytometry (Figure S3F) and scRNA-seq (Figures 5A and 5B). B cells, NK cells and monocytes in the co-cultured

PBMCs were very rare. Thus, in our co-culture system, the anti-tumor effect by cusatuzumab is primarily came from impaired Treg formation and activation, which enhanced cytotoxicity and proliferation of CD8⁺ T cells. Following your suggestion, in our next study, we are developing a CD34⁺ hematopoietic stem cell-engrafted humanized mouse model transplanted with varied NPX PDXs, that enables the investigation of the anti-tumor effect of cusatuzumab in a more comprehensive immune system.

3. *Lipid signaling cause versus consequence. The experiments provided in Figure 5 and 6 suggest a role of lipid signaling in CD70/CD27 mediated induction of Treg cells. However, proliferation and expansion of cells always includes increased metabolic activity. The author should analyze the known CD27 signaling pathway including NFkB activation and analyze the cell cycling of Treg cells. Most probably, the increased metabolic activity reflects an activation of Treg cells with increased expansion and increased metabolic activity. Thus, the described lipid signaling is rather a consequence of the activation than the cause of the suppressive activity.*

Reply: Following your suggestions, we firstly evaluated the proliferation of Tregs in our co-culture system. After 72-hour T cell activation, CD4⁺ T naïve T cells were co-cultured with NPC cells for additional 48 hours in the presence of 25 IU/mL IL-2 and 2 ng/mL TGF-beta. The 48-hour co-culture was too short to induce a higher proliferation of Tregs via CD70-CD27 interaction, and the higher Treg fraction was mainly caused by CD70-induced Treg differentiation. Thus, increased lipid metabolism might not be caused by the elevated proliferation of Tregs. Subsequently, as you suggested, we performed scRNA-seq analyses in co-cultured Tregs, showing that neither NFkB signaling nor cell cycle-related signaling was enriched in Tregs co-cultured with CD70-NC NPC cells.

Additionally, to address your concern more comprehensively on the importance of CD70-CD27 interaction on lipid metabolism in Tregs, we used a lipid-depleted culture

medium to perform co-culture functional assays between CD70-NC/CD70-KO NPC cells and CD4⁺ naïve T cells/PBMCs. In a lipid-depleted microenvironment, CD70-CD27 interaction lost their effect on inducing Treg differentiation and activation, verified by flow analyses and ELISAs (Figures 6P to 6R). Furthermore, lipid depletion inhibited intracellular lipid contents in Tregs, mitochondrial functions and fatty acid oxidation in Tregs, even in the presence of CD70 (Figures 6T to 6X). Thus, CD70-mediated immunosuppression in the lipid-depleted microenvironment did not inhibit CD8⁺ T cell immunity (Figure 6S), showing anti-tumor immunity even higher than CD70-KO cells in a lipid-normal microenvironment, suggesting an active role of lipid metabolism in regulating CD4⁺ and CD8⁺ T cell functions, as consistent with our recent Nature Medicine paper (Zhang et al., 2022).

4. *Regulation of CD70 expression. The data provided in Figure 7 documents that NFKB2 binds to the CD70 promoter. In addition, CD70 expression correlates with NFKB2 expression. However, there is no functional link provided. To conclude that EBV-mediated NFKB2 activation is the cause of the observed high expression of CD70 on nasopharyngeal carcinoma cells an experiment with NFKB pathway inhibition is needed.*

Reply: We sincerely appreciate your constructive suggestions on the upstream mechanism of CD70 overexpression in NPC cells. First, we performed NFKB2 knockdown in C666 cells, which led to the downregulation of CD70 in both mRNA and protein levels (Figures 6I to 6K). Second, we treated C666 with SN-52, an inhibitor that specifically inhibited the nuclear translocation of NFKB2, which could also inhibit the mRNA and protein expression of CD70 (Figures 6I to 6K). Such CD70 downregulation via upstream inhibition also induced higher tumor killing efficacy in NPC-PBMC co-culture system (Figures 6L and 6M).

Minor points

1. There are various typos and misspellings throughout the manuscript, i.e. quiescent Treg are misspelled in most figures.

Reply: Thank you very much for pointing out the problem. We have corrected the typos and misspellings in the revised manuscript.

2. Data on the expression of CD27 on Treg cells (FACS analysis) is missing. In addition, the expression of PD1 on tumor infiltrating CD8⁺ T cells and the expression of PD-L1 on tumor cells is missing.

Reply: We performed additional flow cytometry analyses to evaluate CD27 expression on co-cultured CD4⁺ T cells (Figures 3C, 3G and S3B); PD-1 and TIM3 expression in CD8⁺ T cells (Figure S3M) and PD-L1 on CD70-NC and CD70-KO NPC cells (Figure S2K).

3. The proposed classification of Treg cells into naïve Treg, quiescent Treg and suppressive Tregs is not a standard and it is not described in the given references 10 and 15 (line 106). If this classification is used throughout the paper, the functional effect of naïve Treg cells, quiescent Treg cells and suppressive Treg cells should be analyzed functionally.

Reply: Thank you very much for your comments. We classified the Treg subtypes based on the expression of marker genes and signaling activities (Figures S1C and 1H). Based on previous literature, we changed the term “suppressive Tregs” into “effector Tregs (eTregs)”, “quiescent Tregs” into “resting Tregs (rTregs)”. Detailed references that support our annotation were added in the revised manuscript (Liston and Gray, 2014; Mijnheer et al., 2021).

4. The expression of CD70 on non-tumoral cells is ignored. Figure 2e clearly shows a higher expression of CD70 in tumor-infiltrating B and T cells than on tumor cells. In addition, Figure 2d documents a high CD70 expression in TILs. The limitation that the *in vivo* CD70 ligation might come from tumor cells or tumor infiltrating lymphocytes (and not only from cancer cells) should at least be discussed.

Reply: Thank you very much for your question. The violin plot shown in Figure 2D showed a higher fraction of CD70⁺ cells in the tumor cell cluster. More than 99% of the immune and stromal cells, including T cells, B cells, myeloid cells, fibroblasts and epithelial cells, did not express CD70 at all (These CD70-negative cells were shown at the bottom of the line). The straight line in each cell cluster only showed the minimum to maximum CD70 expression levels. The width of the violin plot mattered more because it stood for the frequency of cells with a certain CD70 expression level (Figure 2H). The violin plot was also ordered descending based on the average CD70 expression (from left to right).

Without causing the misunderstanding of CD70 expression distribution across tumor cells and immune cells, we did not show the single dots, including those that were outliers, and provided more detailed explanations in the figure legend. We also performed additional flow cytometry analyses on fresh NPC biopsies, showing CD70⁺ fraction was significantly higher in EPCAM⁺ tumor cells than in EPCAM⁻ non-tumor cells (Figure 2E).

5. In Figure 3, co-cultivation experiments with purified CD4⁺ cells and PBMCs are provided. A similar experiment should be performed with purified CD8⁺ T cells. It is expected that CD27 signaling on CD8⁺ T cells might expand effector T cells.

Reply: Following your suggestion, we performed the co-culture experiment between CD70-NC/CD70-KO cells and CD8⁺ T cells after 3-day TCR activation. A consistent concentration of IL-2 (25 IU/mL) was provided in the co-culture system, and we found that CD70-NC NPC cells did not expand CD8⁺ T cells more effectively, compared to CD70-KO cells, possibly the co-culture assay only lasted for 48 hours, and these T cells were potently activated and the initial concentration of IL-2 is high. According to recent reviews, CD70-CD27 signaling on CD8⁺ T cells remains controversial due to context-specific molecular mechanisms.

6. *Statistical tests should be included in all figure legends. For example, what statistical test has been used to compare tumor growth curves provided in figure 4o.*

Reply: Thank you very much for your feedback. We have added more details of what kind of tests we used to draw statistical significance in the revised manuscript.

Reviewer #3 (Remarks to the Author): with expertise in regulatory T cells, metabolism

The manuscript by Guan et al. describes how CD70 expressed by nasopharyngeal carcinoma cells (NPC) drives Treg development. The authors have shown using multiple approaches and cutting-edge techniques that knock down of CD70 or anti-CD70 treatment can abrogate NPC-induced Treg induction and promote tumor eradication through CD8+ T cells. Mechanistically, the authors identified that CD27-CD70 signaling supports Treg function by activating lipid metabolism in these cells. Finally, it was shown that CD70 expression by NPC is driven by EBV-dependent chromatin remodelling.

This work is highly relevant to the field and may provide new insights into the mode of actions of targeting costimulatory molecules using antibody-based strategies to promote anti-tumor immunity. In particular, the results are of great importance, since many of these receptors are expressed both on Tregs and conventional CD4/CD8 cells and it is not clear how activation of these receptors affects their respective functions.

Dear reviewer,

It is a pleasure to have your recognition of our findings about the molecular mechanism and clinical utility of targeting CD70-CD27 co-stimulatory signaling in the NPC microenvironment to overcome Treg-mediated immunosuppression via lipid metabolism reprogramming. Based on your comments and suggestions, we extensively revised our manuscript to address your concerns using our new results and analyses. The detailed revisions are shown below. Thank you very much again for helping us overcome the limitations raised in our original manuscript.

However, I have several major concerns that should be addressed:

- 1. - The authors classify Tregs based on naïve, quiescent and suppressive Tregs. To me, it is unclear how the authors come to this classification based on the presented analysis. Since these data are derived from RNA sequencing, the genes that are differentially expressed in each of these clusters need to be provided. To me, it seems like the authors distinguish Treg subsets based on effector status, rather than being 'quiescent' or 'suppressive'.*

Reply: Thank you very much for your suggestion. We classified the Treg subtypes based on marker genes and signaling activities (Figures S1C and 1H). Based on previous literature, we changed the term “suppressive Tregs” into “effector Tregs (eTregs)”, “quiescent Tregs” into “resting Tregs (rTregs)”. Detailed references that support our annotation were added in the revised manuscript (Liston and Gray, 2014; Mijnheer et al., 2021).

2. - Overall, the flow cytometry analyses are poor. The data provided showing that NPC drive Treg through CD27-70 signaling are not at all convincing. In particular, the expression of CTLA4 induced by NPC in the culture set up is extremely modest (if not absent), but is used to make the strong conclusion that CD70+ NPC, but not CD70- cells convert naïve CD4 T cells into the ‘suppressive Treg’ phenotype (identified as FOXP3+/CTLA4+ cells). The authors also draw the conclusion that direct cell-cell contact would be required for sTreg development. However, FOXP3 expression increases compared to T cell only cultures both in a transwell and a direct contact setting (both using NPC that do or do not express CD70). FOXP3 expression even seems to be more upregulated in a transwell setting rather than the cell contact setting, which is a strong indicator of Treg conversion. How can the authors exclude the possibility in this culture set up that soluble mediators derived from the tumor cells further upregulate FOXP3 expression (which would also imply that these cells become sTreg based on the trajectory analysis that shows FOXP3 expression increases during transition of nTregs to sTregs).

Reply: We sincerely appreciate your comments on our flow cytometry results. As you suggested, we re-analyzed our flow cytometry data. For instance, in the CD4⁺ naïve T cells co-cultured with CD70-NC/CD70-KO C666 cells, the fraction of FOXP3⁺ Tregs decreased from 61.14% to 41.33% (fold change = 0.67), FOXP3⁺CTLA4⁺ Tregs decreased from 15.09% to 4.57% (fold change = 0.30), and CD4⁺CD27⁺ cells increased from 11.59% to 46.46% (fold change = 4.01) (Figures 3E to 3G).

For cusatuzumab treatment, fraction of FOXP3⁺ Tregs decreased from 57.58% to 35.25% (fold change = 0.61), and FOXP3⁺CTLA4⁺ Tregs decreased from 9.00% to 1.10% (fold change = 0.12), and CD4⁺CD27⁺ cells increased from 23.21% to 65.44% (fold change =

2.82) (Figures 4A to 4C).

To verify our findings based on flow analyses, in the revised manuscript, we performed additional clinical examinations, RNA-seq analyses, qRT-PCR, ELISAs, mass spectrometry, and scRNA-seq that comprehensively supported the pro-immunosuppression effect of CD70⁺ NPC cells (including but not limited to Figures 1M, 1N, 3D, 3H, 3I, 3U, 3V, S3B, S3C and 4D).

We are sorry for some misunderstanding caused by our figure and the lack of explicit explanation in the corresponding texts (Figures 1J to 1L). The difference in Treg frequencies between transwell co-culture and direct co-culture was due to different experimental settings, such as the total number of CD4⁺ naïve T cells seeded in 1.3 cm² transwell and culture plate, as well as the use of CD4⁺ naïve T cells isolated from different donors for transwell/direct co-culture, respectively. Thus, the fraction of FOXP3⁺ Tregs was not comparable between the transwell and direct settings. We are sorry for the misleading information in the figure, so we modified and provided a sufficient explanation in the revised manuscript. We performed several times of the transwell co-culture assay using different effector: target ratios and generated consistent results showing that NPC cells did not significantly enhance Treg differentiation or Treg activation in transwell co-culture. Hence, we determined to perform the direct co-culture assay and found NPC cells significantly enhanced Treg differentiation and Treg activation via direct cell-cell contact.

We also performed qRT-PCR and ELISAs to validate that enhanced Treg differentiation and suppressive activity of C666 cells via cell-cell contact (Figures 1M and 1N).

3. Furthermore, how are other key Treg molecules such as ICOS, TIGIT, GITR, 4-1BB, PD-1 affected in these cultures? These markers could have easily been addressed by flow cytometry. Also, do the cells indeed become more suppressive after coculture with NPC and can CD70 blockade or CD70-KO inhibit the suppressive activity of these cells? This is particularly important given the very modest changes in CTLA4 expression levels shown.

Reply: Thank you very much for your suggestions. We performed flow cytometry and qRT-PCR analyses on ICOS, GITR, 4-1BB and PD-1 in our co-culture system. The result showed that CD70-CD27 interaction promoted these markers, except for TIGIT (Figures 3I, S3B and S3C), on co-cultured FOXP3⁺ Tregs, indicating a higher activation status of Tregs. For TIGIT, we initially performed qRT-PCR to examine the mRNA change and found CD70-CD27 did not induce a significant upregulation of TIGIT (Figure 3I). Subsequently, we read recent papers which also stated the TIGIT⁺ fractions in FOXP3⁺ Tregs did not change in CD70 (-/-) mice compared to wildtype ones (Anta et al., 2021), nor CD27⁺CD70⁻ Tregs compared to CD27⁻CD70⁺ Tregs (Arroyo Hornero et al., 2020).

To address your second concern on the suppressive ability of Tregs, we performed ELISAs and mass spectrometry on Tregs co-cultured with CD70-NC/CD70-KO cells, and C666 cells treated with IgG/cusatuzumab, as well as CD70-NC/CD70-KO tumor lysates from humanized mice. The secreted IL-10, TGF-β and adenosine in all experimental conditions were lowered in CD70-KO and cusatuzumab groups (Figures 3D, 3H, 3U and 4D).

4. Also, the cultures were performed in the presence of TGF- β . Do NPC also induce Treg differentiation without adding exogenous TGF- β to the culture? And does interfering in CD27-CD70 interaction still revert Treg induction? Additionally, what would be the effect of CD70 on NPC on natural/thymus-derived Tregs (CD25^{hi}CD127^{low} T cells) in terms of phenotype/suppressive activity instead of using TGF- β -polarizing cultures?

Reply: Following on your suggestions. We performed co-culture assays without adding exogenous TGF- β , which significantly inhibited *in vitro* Treg differentiation from CD4⁺ naïve T cells, due to low TGF- β secreted by C666 cells, as measured by ELISAs. The Treg differentiation became minimal, thus the effect of CD70-CD27 interaction was not explicit.

We also sorted natural-derived Tregs from peripheral blood using the CD4⁺CD25⁺CD127^{dim/-} Regulatory T Cell Isolation Kit II, human (Miltenyi, 130-094-775), the CD70-CD27 interaction could also induce Treg activation and higher suppressive activity, but the effect of Treg differentiation from CD4⁺ naïve T cells could not be explored in this kind of co-culture assay. The *in vitro* TGF- β -inducing Treg differentiation generated induced Tregs were genetically and functionally similar to nTreg, as stated in Fantini et al., Nature Protocols, and other well-established protocols.

5. - In line with previous comment, the flow cytometric data regarding the enhanced CD8 T cell activation is very marginal, for instance in Figure 3E. Did the authors also analyse the expression of CD107a or the production of IFN- γ /TNF- α by the CD8 T cells? Moreover, in the organoid co-cultures, vehicle controls already contain >50% of dead tumor cells, which makes it questionable whether the effect caused by anti-CD70/anti-PD1 treatment is due to enhanced CD8-mediated killing. It would also be interesting to see in the *in vivo* xenotransplantation model whether CD70- NPC cells would be ignored when deliberate CD27 costimulation is provided using agonistic antibodies.

Reply: We highly appreciate your suggestions on detecting the production of cytotoxic

cytokines in our PBMC co-culture system. We performed ELISAs on IFN- γ and TNF- α and showed PBMCs co-cultured with CD70-KO NPC cells and NPC cells treated with cusatuzumab had significantly higher concentrations of these two cytokines. A consistent ELISA result was also obtained from tumor lysates of humanized mice (Figures 3O, 3U and 4I).

For our NPC organoid co-culture system, because the organoids were very vulnerable to PBMCs, the vehicle-treated organoids also displayed a high cell death after 48-hour treatment. But both anti-CD70/anti-PD-1 monotherapy and combination therapy induced higher cell death compared to IgG treatment (Figures 4L and 4N). The high apoptosis index in the vehicle-treated group might be due to the high difficulty of organoid dissociation, thus there were some tightly connected cell organoids recognized as single cells by the flow cytometer. We also took microscopic images of organoids receiving these treatments, and quantified the size/numbers of organoids (Figures 4K and 4M). The killing efficacy was also validated in our PDX-transplanted humanized mouse model.

For the PDX transplantation procedure, we subcutaneously placed small chunks of PDX in humanized mice. If the PDXs were dissociated into single cells and performed cell sorting to sort out CD70- cells prior to transplantation, the PDX-derived cells could no longer develop subcutaneous tumors in humanized mice, even in nude mice where they were originally kept in. Using a recombinant CD70 antibody *in vitro* without co-culturing with CD70⁺ NPC cells to mimic CD70-CD27 co-stimulation showed a consistent effect on inducing Treg differentiation and activation (Figures 2A to 2D).

6. In addition, authors make general statements based on their *in vivo* xenograft model, while melanoma cells for instance also express CD70 (line 276). Melanoma mouse models are well established, and antigen-specific responses can be assessed. The authors need to therefore show that similar phenomena occur in another well-established *in vivo* model. This would also link the present study to current literature stating that CD27 engagement on T cells has strong anti-tumor efficacy by promoting CD8 T cell priming, which authors do not touch upon in their discussion.

Reply: Thank you very much. According to your suggestion, we performed CRISPR to knockout CD70 in a murine melanoma cell line B16-F10 (Supplementary figure). We orthotopically injected CD70-NC and CD70-KO B16-F10 cells into C57BL/6 mice and monitored tumor growth from day 7 post-injection. After 2-week tumor growth, the CD70-KO B16-F10 tumors grew significantly slower than the CD70-NC ones (Supplementary figures 5D to 5F). Since CD70-KO did not influence tumor cell proliferation, the tumor-inhibiting effect of CD70-KO in melanoma was also due to elevated anti-tumor immunity. As you suggested, we also discussed the potential effect of CD70-CD27 interaction on either activation or exhaustion of CD8⁺ T cells, which is a context-specific process, according to recent publications.

In addition, our next study has been initiated right after receiving your comments, we plan to send CD45⁺ cells isolated from CD70-NC tumors, CD70-KO responding tumors and CD70-KO resistant tumors for scRNA sequencing to characterize the microenvironmental characteristics that influenced the outcome of anti-CD70 treatment, which will facilitate the current clinical trials evaluating its efficacy in solid tumors. We sincerely appreciate your suggestions that have led to our new ideas on anti-CD70 therapy in solid malignancies and make our revised manuscript of higher interest to a broader audience.

7. - The metabolic signature induced by CD27-CD70 interaction heavily relies on RNA sequencing data and only little on actual metabolomics analyses. Moreover, the

metabolomics analysis is very limited. Differential metabolite levels and lipid species should at least be shown in an unbiased manner, for instance using volcano plots and/or a heatmaps to evaluate the complete difference in the metabolome/lipidome in Tregs obtained from cocultures with CD70+ or CD70-KO NPC. This is particularly important since gene expression data do not always correlate with metabolic activity.

Reply: Thank you very much for your suggestion. In the revised manuscript, we performed additional metabolic functional assays to more comprehensively characterize the changes of lipid metabolism in Tregs induced by CD70-CD27 interaction. By performing Oil O red staining, the Seahorse assay, electron microscopy, JC-1 staining and the fatty acid oxidation assay, we demonstrated that Tregs co-cultured with CD70-KO NPC cells decreased total intracellular lipids, OXPHOS, fatty acid oxidation and mitochondrial integrity and functions, compared to the ones co-cultured with CD70-NC cells (Figures 6A to 6I).

8. Importantly, the authors should show that the observed differences are linked to the functionality of Tregs. Do Tregs cocultures with CD70+ NPC lose their suppressive function when lipid metabolism is impaired? Or does deliberate CD27 signaling in Tregs install a lipid metabolism program that supports their suppressive activity?

Reply: Thank you very much for your questions. Following your comments, we performed additional co-culture assays using a lipid-depleted culture medium. The effect of CD70+ NPC cells on inducing Treg differentiation and activation was impaired in a lipid-depleted microenvironment (Figures 6P to 6R). Thus, we considered that the immunosuppression-promoting effect of CD70-CD27 interaction was dependent on downstream lipid metabolism. Lowering lipid content in the culture microenvironment no longer enabled Treg to adapt effectively to lipid reprogramming induced by CD70-CD27 signaling, thus inhibiting immunosuppression and enhancing the tumor-killing efficacy (Figures 6S to 6X).

9. The authors make the bold statement that CD70-induced Tregs rely more on FAO-driven OxPHOS rather than glycolysis. This should at least be shown functionally using Seahorse analyses. Furthermore, authors claim that CD27-CD70 signaling promotes de novo fatty acid uptake, which could be shown by flow cytometry using fluorescent lipid analogs like BODIPY-C12/C16.

Reply: Thank you very much for your suggestion. We performed the Seahorse assay on CD4⁺ T cells co-cultured with CD70-NC and CD70-KO NPC cells. The Seahorse analyses showed increased OXPHOS-driven ATP generation in the CD70-NC co-culture system. We are sorry for the incorrect statement in our original manuscript stating “CD70-induced Tregs rely more on FAO-driven OXPHOS rather than glycolysis”, we corrected the statement into “In CD70-induced Tregs, mitochondrial fatty acid oxidation-driven OXPHOS and ATP generation were significantly enhanced to maintain Treg metabolic fitness in the TME”. In addition, we performed de novo fatty acid uptake using BODIPY-C12 and the fatty acid oxidation assay. The results showed FA uptake and oxidation were increased in CD70-induced Tregs (Figures S7D and SE).

10. Finally, it is not at all proven that FAO-driven OxPHOS affects mitochondrial stability, resulting in increased expression of STAT5A or TBX21. Authors only provide indirect proof based on RNA sequencing data. Similar to my previous comment, the authors do

not provide any evidence that CD27-CD70 signaling in Tregs stabilizes mitochondrial integrity, for instance by evaluating mitochondrial morphology, mitochondrial mass/membrane potential using flow cytometric dyes and/or expression of ETC complex proteins. Furthermore, authors need to show that interfering in lipid metabolism instructed via CD27-CD70 signaling prevents the upregulation of key transcription factors, in order to conclude that CD70+ NPC provide a metabolic switch for Treg development and suppressive activity (which is the main conclusion of the study).

Reply: We sincerely appreciate your comments on our metabolic investigation. Following your suggestions, we used electron microscopy to examine the mitochondria in Tregs co-cultured with CD70-NC/CD70-KO NPC cells. The electron microscopy images demonstrated a higher number of healthier mitochondria in CD70-induced Tregs (Figure 6E). Besides, we performed flow cytometry analysis on aggregated/monomer JC-1 in Tregs, showing changes in mitochondrial potentials between NC-Tregs and KO-Tregs (Figures 6G and 6H).

We also showed that lipid depletion in the co-culture system significantly impaired the effect of CD70-CD27 signaling on modulating Treg suppressive activity via lipid metabolism reprogramming (Figures 6P to 6X).

11. - The authors nicely demonstrate that EBV infection drives CD70 expression in NPC via NFKB2. However, it is unclear to this reviewer how authors conclude that NFKB2 was correlating the most with CD70 expression by NPC, especially since NFKB2 is absent from figure 7D. Also, it would be highly interesting to see in the luciferase reporter assay whether other NF- κ B signaling components identified in Figure 7D are less efficient in driving CD70 expression

Reply: Thank you very much for your recognition of our investigations on the upstream molecular mechanism of CD70 upregulation in NPC cells. To address your concern, we performed NFKB2 knockdown and NFKB2 inhibition (using SN-52, an inhibitor that inhibits the nuclear translocation of NFKB2), which both downregulated CD70 mRNA and the CD70⁺ cell fraction in NPC cells and subsequently promoted anti-tumor immunity (Figures 7I to 7M). For other transcription factors, we added a supplementary list showing the predicted binding efficacy of these TFs (Table S8), and their correlation to CD70 mRNA level in two NPC bulk RNA-seq cohorts. NFKB2 has the highest predicted binding score to the promoter region of CD70 that had higher accessibility caused by EBV infection and highest correlation coefficient in two NPC cohorts, which was the major reason we chose NFKB2 to perform downstream validation.

12. - Overall, in many occasions, it is unclear how the data were statistically tested and what the sample size is. Also, some figures are unclear as figure legends are sometimes missing and similar color coding is used for different experimental conditions throughout the manuscript.

Reply: Thank you very much for pointing out the issue. In the revised manuscript, we added more details on how statistical results were drawn from the data. We also modified the figures with clearer labels and color coding that benefits the readers.

Reviewer #4 (Remarks to the Author): with expertise in transcriptomics

In this paper, the authors carried out a series of experiments to discover how tumor microenvironment characteristics alter immune homeostasis and immunotherapy efficacy in nasopharyngeal carcinoma (NPC). By analyzing single-cell RNA-seq data of 357,206 cells from 50 patient samples, they revealed that CD70-CD27 interaction can enhance NPC cell differentiation and suppress Tregs functions. They further validated the CD70's role in revitalizing CD8+ T-cell via CD70 knockout and anti-CD70 treatment, and inhibiting a collective lipid signaling network in Tregs. The ATAC-seq analysis proved transcriptional regulation of CD70 via an Epstein-Barr virus (EBV)-dependent epigenetic modification in NPC cells. It is also interesting to see that the combination therapy of anti-CD70 and anti-PD-1 exhibits an improved tumor-killing efficacy compared to monotherapy. Overall, this manuscript targets a very interesting and significant topic regarding NPC microenvironment and immunotherapy. The manuscript is intact with solid evidence to support their results and conclusions. Nevertheless, there are several remaining concerns.

Dear reviewer,

Thank you very much for this acknowledgement and recognition of our work. In the revised manuscript, we added more clinical and experimental evidence to support our results and conclusions. We followed your comments and suggestions very carefully and revised the manuscript accordingly. Below are the detailed replies to address your concerns. Thank you very much again for your time and efforts on our study.

- 1. In terms of deconvolution analysis of Visium data (Figure 2C), the author used CD70 expression to showcase CD70+ cancer cells distribution and then correlated it with Treg abundance to showcase co-localization of Treg and CD70+ cancer cells. Therefore, directly calculating cell type proportion for CD70+ tumor and Treg cells will be persuasive and intuitive. Refer to this paper (<https://www.nature.com/articles/s41592-022-01480-9>) and consider using cell2location to analyze cell proportion for identified cell types (CD70+ tumor and Treg cells). In addition, please provide more details in the method section regarding Visium analysis, including functions and parameters.*

Reply: Thank you very much for your detailed suggestions. We performed deconvolution on NPC tumor cells in our spatial data using Seurat and cell2location. Both deconvolution results showed high spatial correlation between NPC cells and Tregs (Figures 2C, 2D, S2E and S2F). We also cited Seurat and cell2location in the revised manuscript and added a paragraph in the method section to describe the parameters we used for cell2location analysis. As you suggested, we also added more details on how we performed Visium spatial analysis based on Seurat.

2. Some terminologies should be clarified more clearly. For example, (1) *spatial-seq* is not a standard term for spatial transcriptomics. Therefore, authors may directly use *Visium* instead of *spatial-seq* in the manuscript. (2) *Spatial voxel* or *spots*? *Visium*'s measuring units are *spots* rather than *voxels* (*voxels* usually were used for 3D spatial points).

Reply: Following your suggestions, we corrected the words in our revised manuscript.

3. Line 175, I suppose it should be naïve CD4+ T cells?

Reply: Thank you very much for pointing out the typo. Yes, it should be CD4⁺ T cells.

4. *Monocle 3* should be cited somewhere relevant.

Reply: Thank you very much for pointing out the issue. We cited *monocle 3* in our revised manuscript.

5. Detailed descriptions of *CellphoneDB* and *CellChat* should be mentioned with

parameter usage.

Reply: Thank you for your comment. We added more details in the method section to describe the parameters that we used to perform the CellphoneDB and CellChat analyses.

REFERENCES

- Arroyo Hornero, R., Georgiadis, C., Hua, P., Trzuppek, D., He, L. Z., Qasim, W., Todd, J. A., Ferreira, R. C., Wood, K. J., Issa, F., and Hester, J. (2020). CD70 expression determines the therapeutic efficacy of expanded human regulatory T cells. *Commun Biol* 3, 375.
- Ellis, G. I., Reneer, M. C., Velez-Ortega, A. C., McCool, A., and Marti, F. (2012). Generation of induced regulatory T cells from primary human naive and memory T cells. *J Vis Exp*.
- Fantini, M. C., Rizzo, A., Fina, D., Caruso, R., Becker, C., Neurath, M. F., Macdonald, T. T., Pallone, F., and Monteleone, G. (2007). IL-21 regulates experimental colitis by modulating the balance between Treg and Th17 cells. *Eur J Immunol* 37, 3155-3163.
- Flieswasser, T., Van den Eynde, A., Van Audenaerde, J., De Waele, J., Lardon, F., Riether, C., de Haard, H., Smits, E., Pauwels, P., and Jacobs, J. (2022). The CD70-CD27 axis in oncology: the new kids on the block. *J Exp Clin Cancer Res* 41, 12.
- Gong, L., Kwong, D. L., Dai, W., Wu, P., Li, S., Yan, Q., Zhang, Y., Zhang, B., Fang, X., Liu, L., *et al.* (2021). Comprehensive single-cell sequencing reveals the stromal dynamics and tumor-specific characteristics in the microenvironment of nasopharyngeal carcinoma. *Nat Commun* 12, 1540.
- Li, Y., and Kurlander, R. J. (2010). Comparison of anti-CD3 and anti-CD28-coated beads with soluble anti-CD3 for expanding human T cells: differing impact on CD8 T cell phenotype and responsiveness to restimulation. *J Transl Med* 8, 104.
- Liston, A., and Gray, D. H. (2014). Homeostatic control of regulatory T cell diversity. *Nat Rev Immunol* 14, 154-165.
- Liu, Y., He, S., Wang, X. L., Peng, W., Chen, Q. Y., Chi, D. M., Chen, J. R., Han, B. W., Lin, G. W., Li, Y. Q., *et al.* (2021). Tumour heterogeneity and intercellular networks of nasopharyngeal carcinoma at single cell resolution. *Nat Commun* 12, 741.
- Mijnheer, G., Lutter, L., Mokry, M., van der Wal, M., Scholman, R., Fleskens, V., Pandit, A., Tao, W., Wekking, M., Vervoort, S., *et al.* (2021). Conserved human effector Treg cell transcriptomic and epigenetic signature in arthritic joint inflammation. *Nat Commun* 12, 2710.
- O'Neill, R. E., Du, W., Mohammadpour, H., Alqassim, E., Qiu, J., Chen, G., McCarthy, P. L., Lee, K. P., and Cao, X. (2017). T Cell-Derived CD70 Delivers an Immune Checkpoint Function in Inflammatory T Cell Responses. *J Immunol* 199, 3700-3710.
- Qiu, X., Yang, S., Wang, S., Wu, J., Zheng, B., Wang, K., Shen, S., Jeong, S., Li, Z., Zhu, Y., *et al.* (2021). M(6)A Demethylase ALKBH5 Regulates PD-L1 Expression and Tumor Immunoenvironment in Intrahepatic Cholangiocarcinoma. *Cancer Res* 81, 4778-4793.
- van de Ven, K., and Borst, J. (2015). Targeting the T-cell co-stimulatory CD27/CD70 pathway in cancer immunotherapy: rationale and potential. *Immunotherapy* 7, 655-667.
- Zhang, Y., Vu, T., Palmer, D. C., Kishton, R. J., Gong, L., Huang, J., Nguyen, T., Chen, Z., Smith, C., Livak, F., *et al.* (2022). A T cell resilience model associated with response to immunotherapy in multiple tumor types. *Nat Med* 28, 1421-1431.

REVIEWER COMMENTS

Reviewer #1 (Remarks to the Author):

The manuscript is substantially improved and the authors have been responsive to reviewer comments and requests. I am supportive at this stage

Reviewer #2 (Remarks to the Author):

Gong L. provide a detailed point-by-point response providing some new experimental data or discussing the critical comments. However, some major criticism has not been sufficiently addressed.

1. Autologous versus allogeneic CD4+ T cells. A main criticism was that all experiments were done in allogeneic setting using PBMCs or CD4+ T cells co-cultivated with allogeneic tumor cells. In addition, these allogeneic T cells have been activated via anti-CD3 antibody. This situation is far away from physiological conditions since the frequency of allogeneic specific T cells recognizing a target cell is probably 100 fold higher than the frequency of tumor-antigen-specific autologous T cells. Moreover, the strengths of the T cell receptor signal by anti-CD3 antibody leads to a much stronger activation than the physiological interaction of TCR with MHC class I/peptide. The novel provided data with co-incubating PBMCs with primary NPC cells is for several reasons inconclusive. 1. Data from just one patient is provided with a statistical analysis of technical replicates. 2. PBMCs are co-cultivated with the ADCC optimized antibody cusatuzumab. It is impossible to distinguish in this experimental setting between ADCC-mediated effects versus blocking of the CD70/CD27 interaction (see point 2). The control conditions used are unclear. In the panels it is indicated (+) versus (-) Cusatuzumab. In the text it is stated that IgG has been used as a control. However, if cusatuzumab is used, a ADCC-optimized IGG antibody is the correct control. 2. CD70-blocking by cusatuzumab. The goal of the entire manuscript was to analyze the function of the CD70/CD27 interaction in the induction of Treg cells. There is absolutely no rationale to use an ADCC-optimized antibody to perform these experiments. The experiments should have been done by a blocking-only antibody (ADCC-death variant) or at least include the proper controls such as Potelligent ADCC optimized control antibody. The new data provided in Figure S3e and S3f only documents that after anti-CD3 activation, mainly CD3+ T cells (CD4+ and CD8+ T cells) expand. However, initial culture conditions contain a target to PBMCs ratio of 1:5. PBMCs contain 5-20% cells that can mediate ADCC (NK cells and monocytes). Thus, even though NK cells and monocytes do not expand after anti-CD3mAb T cell activation and the final percentage of CD64+ cells in the cell culture is approximately 2%, the actual effector target ratio of cells that can mediate ADCC is still 1:1, a condition where cusatuzumab was shown to mediate strong ADCC effects. Thus, the data provided by the authors do not exclude an ADCC effect of cusatuzumab.
3. Lipid signaling cause versus consequence. This point is adequately addressed.
4. Regulation of CD70 expression: this point is adequately addressed.

Reviewer #3 (Remarks to the Author):

The authors have made much effort improving the impact of the manuscript based on the reviewers comments. I appreciate all additional experiments in response to my concerns. However, several issues remain:

1. Flow cytometry data remain not convincing. Though it may certainly be true that CD70 expression by nasopharyngeal carcinoma cells impact on Treg polarization based on the expression of FOXP3, it

remains questionable whether this involves poolarization of naive T cells into induced Tregs, or whether there is an expansion of committed Tregs present in PBMC. Also, important controls are missing (for example: in co-cultures and transwell cultures the condition without any tumor cells is missing). Expression of CTLA is not convinving and no direct impact on true suppressive function of the Tregs is not provided.

2. The additional analyses on the organoids do not take away my concerns on the high cell death already in the vehicle treated group.

3. The additional metabolic studies in Tregs are extensive, but since naive CD4 T cells were cocultured with NPC, how sure are the authors that they are looking at the metabolism of Tregs as they are studying a mixture of cells that are FOXP3+ and FOXP3-. Furthermore, the culture set-up is different from the flow cytometric analyses done in PBMC/NPC cultures, making it impossible to evaluate whether the effect of CD70 expressed by NPC is mediated through metabolism.

4. No causal link between Treg metabolism and suppressive function of Tregs has been provided.

Reviewer #4 (Remarks to the Author):

All my comments have been addressed.

Reviewer #1 (Remarks to the Author):

The manuscript is substantially improved and the authors have been responsive to reviewer comments and requests. I am supportive at this stage

Dear reviewer,

Thank you very much for your support of our revision and manuscript. In the 1st round review, we followed your comments and suggestions very carefully and revised the manuscript substantially to improve scientific novelty and validity. Thank you very much again for your help.

Reviewer #2 (Remarks to the Author):

Gong L. provide a detailed point-by-point response providing some new experimental data or discussing the critical comments. However, some major criticism has not been sufficiently addressed.

Dear reviewer,

We highly appreciate your recognition of our first-round revision and additional suggestions on our revised manuscript. During the past three months, we have performed autologous co-culture experiments using CAR-T cells and additional NPC patient samples. Meanwhile, we have also performed tumor-killing assays using a CD70-blocking antibody, as you suggested. Below are the point-by-point revisions that we have made to address your concerns. Thank you very much again for your time, efforts and support.

- 1. Autologous versus allogeneic CD4⁺ T cells. A main criticism was that all experiments were done in allogeneic setting using PBMCs or CD4⁺ T cells co-cultivated with allogeneic tumor cells. In addition, these allogeneic T cells have been activated via anti-CD3 antibody. This situation is far away from physiological conditions since the frequency of allogeneic specific T cells recognizing a target cell is probably 100 fold higher than the frequency of tumor-antigen-specific autologous T cells. Moreover, the strength of the T cell receptor signal by anti-CD3 antibody leads to a much stronger activation than the physiological interaction of TCR with MHC class I/peptide.*

Reply: We agree with your comments that it is very important to validate our findings in an autologous system. Thus, following your advice, we established a CAR-targeting autologous co-culture system between CD19-expressing CD70-NC/CD70-KO NPC cells and anti-CD19 CAR-T cells (Figure S3O and S3P). We did not choose to use TCR-T cells to perform autologous co-culture experiments because the activation of CD4⁺ T cells and CD8⁺ T cells are mediated by MHC class II and MHC class I separately. Followed by CAR-mediated antigen-specific

activation, CD19+CD70-KO NPC cells decreased the fractions of total CD4+/FOXP3+ Tregs and CD4+/FOXP3+/CTLA4+ eTregs, as well as lowered the concentrations of immunosuppressive cytokines IL-10 and TGF- β , which resulted in elevated antigen-specific T-cell killing and release of cytotoxic cytokines, such as IFN- γ and TNF- α (Figure S3Q to S3S). Meanwhile, to enhance statistical reliability, we also performed co-culture experiments using primary tumor cells and paired PBMCs collected from additional 2 NPC patients (detailed results and discussion shown in Q2). Overall, the results from CAR-targeted autologous co-culture experiments are consistent with the ones from NPC patient-derived autologous co-culture and *in vitro* allogeneic co-culture.

The novel provided data with co-incubating PBMCs with primary NPC cells is for several reasons inconclusive.

1. Data from just one patient is provided with a statistical analysis of technical replicates.

Reply: Thank you very much for your feedback. As you suggested, we isolated primary tumor cells and paired PBMCs from additional 2 NPC patients and performed CD70 inhibition in the co-culture system. The new data for Treg immunophenotyping, the tumor-killing assay, and ELISAs of immunosuppressive and cytotoxic cytokines are consistent with previous data from one NPC patient shown in our first-round revision (Figures S4M to S4P).

2. The control conditions used are unclear. In the panels it is indicated PBMCs are co-cultivated with the ADCC optimized antibody cusatuzumab. It is impossible to distinguish in this experimental setting between ADCC-mediated effects versus blocking of the CD70/CD27 interaction (see point 2). Control (-) versus (+) Cusatuzumab. In the text it is stated that IgG has been used as a control. However, if cusatuzumab is used, a ADCC-optimized IGG antibody is the correct control.

Reply: Thank you very much for pointing out the question. As you have seen in the figure panels, we used “Cusatuzumab (+)” to show that cusatuzumab was added in the co-culture system, and used “Cusatuzumab (-)” to show that vehicle IgG control, instead of cusatuzumab, was added in the co-culture system. To avoid misunderstanding, we changed the text in figure panels from “Cusatuzumab (-)” to “Control”. Following your suggestion, we also tested a CD70-blocking-only antibody (ANC-222-050, AdipoGen) in the co-culture system, and the results are discussed in the question below.

3. CD70-blocking by cusatuzumab. The goal of the entire manuscript was to analyze the function of the CD70/CD27 interaction in the induction of Treg cells. There is absolutely no rationale to use an ADCC-optimized antibody to perform these experiments. The experiments should have been done by a blocking-only antibody (ADCC-death variant) or at least include the proper controls such as Potelligent ADCC optimized control antibody. The new data provided in Figure S3e and S3f only documents that after anti-CD3 activation, mainly CD3⁺ T cells (CD4⁺ and CD8⁺ T cells) expand. However, initial culture conditions contain a target to PBMCs ratio of 1:5. PBMCs contain 5-20% cells that can mediate ADCC (NK cells and monocytes). Thus, even though NK cells and monocytes do not expand after anti-CD3mAb T cell activation and the final percentage of CD64⁺ cells in the cell culture is approximately 2%, the actual effector target ratio of cells that can mediate ADCC is still 1:1, a condition where cusatuzumab was shown to mediate strong ADCC effects. Thus, the data provided by the authors do not exclude an ADCC

effect of cusatuzumab.

Reply: Thank you very much for your comments. We purchased and used a CD70-blocking-only antibody (ANC-222-050, AdipoGen) and repeated co-culture experiments between NPC cells and PBMCs. Consistent with the results using cusatuzumab, the blocking-only antibody also significantly enhanced anti-tumor immunity by increasing CD8+ T cell cytotoxicity (Figures S4I to S4L), confirming that CD70 inhibition is able to enhance ADCC-independent tumor-killing effect and T-cell cytotoxicity.

The initial percentages of NK cells and monocytes in human PBMCs are usually 5-20%. If we did not perform T-cell activation and expansion prior to co-culture with NPC cells with a 5:1 ratio, the E-to-T ratio for cells that could mediate ADCC would be maximally 1:1, resulting in detectable ADCC-mediated tumor death in our co-culture system. Since we performed 3-day T-cell activation, expansion, and 2-day co-culture, the final percentages of NK cells and monocytes dropped from 5-20% to less than 2%, as shown by flow cytometry analysis and scRNA-seq (Figures 5A and 5B), resulting in lower than a 0.1:1 E-to-T ratio for cells that could mediate ADCC. Thus, we initially considered that the ADCC effect would not be a significant factor in affecting the tumor-killing efficacy of cusatuzumab, compared to its inhibitory effect on Treg-mediated immunosuppression.

At the beginning of the project, we chose to use cusatuzumab because it is a well-established antibody that has been investigated in phase I and phase II clinical trials in hematopoietic malignancies. Thus, we planned to first demonstrate its efficacy in nasopharyngeal carcinoma because cusatuzumab has a high translational potential and safety profile to facilitate our preliminary clinical investigation on NPC patients in the future.

4. *Lipid signaling cause versus consequence. This point is adequately addressed.*

Reply: Thank you very much for your detailed comments and suggestions in our first-round revision, which significantly improved our molecular investigations in lipid metabolism in CD4⁺ T cells caused by CD70-CD27 interaction.

5. *Regulation of CD70 expression: this point is adequately addressed.*

Reply: Thank you very much for your detailed comments in our first-round revision which made us delineate the upstream mechanism of CD70 upregulation in NPC cells more comprehensively.

Reviewer #3 (Remarks to the Author):

The authors have made much effort improving the impact of the manuscript based on the reviewers comments. I appreciate all additional experiments in response to my concerns. However, several issues remain:

Dear reviewer,

Thank you very much for your support on our revised data and your valuable suggestions this time. According to your comments and suggestions, we performed additional Treg suppression assays, added the control data in figure 1, and repeated the organoid co-culture experiment to address your concerns. The detailed revisions are shown below. Thank you very much again for your time, efforts and support.

1. *Flow cytometry data remain not convincing. Though it may certainly be true that CD70 expression by nasopharyngeal carcinoma cells impact on Treg polarization based on the expression of FOXP3, it remains questionable whether this involves polarization of naive T cells into induced Tregs, or whether there is an expansion of committed Tregs present in PBMC. Also, important controls are missing (for example: in co-cultures and transwell cultures the condition without any tumor cells is missing). Expression of CTLA is not convincing and no direct impact on true suppressive function of the Tregs is not provided.*

Reply: Thank you very much for your detailed comments. To further validate that CD70-CD27 interaction increased Treg immunosuppression activity based on FOXP3 and CTLA4 staining in our previous manuscript, we performed additional Treg suppression assays between CD4⁺ T cells (after co-culture with NPC cells) and paired CD8⁺ T cells. CD4⁺ naïve T cells co-cultured with CD70-KO and cusatuzumab-treated NPC cells exhibited an impaired suppression effect on the proliferation of paired CD8⁺ T cells, compared to the ones co-cultured with CD70-NC and IgG-treated NPC cells (Figures S3D and S4F).

Moreover, we performed CellTrace™ staining on CD4⁺ T cells and showed there was no difference in CD4⁺/FOXP3⁺ Treg proliferation between CD70-NC and CD70-KO co-culture systems.

We consider that the 48-hour co-culture was too short to induce a higher proliferation of FOXP3⁺ Tregs just differentiated from CD4⁺ naïve T cells via CD70-CD27 interaction, and the higher Treg fraction was mainly caused by CD70-induced Treg differentiation. As suggested, we also added the control flow cytometry data for induced CD4⁺ naïve T-to-Treg differentiation without tumor cell co-culture (Figures S1M and S1N). Overall, The Treg suppression assays further corroborated the true suppressive activity of Tregs on the basis of FOXP3/CTLA4 staining, IL-10/TGF- β ELISAs as well as qRT-PCR in our previous manuscript.

- The additional analyses on the organoids do not take away my concerns on the high cell death already in the vehicle treated group.*

Reply: Thank you very much for your comment. To address your concerns, we repeated the organoid-PBMC co-culture experiment and dissociated the organoids for a longer time in order to fully isolate them into single cells. The new result showed that the vehicle-treated group had <40% dead tumor cells (Figures 4L and 4N), which was much lower than the original result with about 70% dead cells.

3. The additional metabolic studies in Tregs are extensive, but since naive CD4 T cells were cocultured with NPC, how sure are the authors that they are looking at the metabolism of Tregs as they are studying a mixture of cells that are FOXP3⁺ and FOXP3⁻. Furthermore, the culture set-up is different from the flow cytometric analyses done in PBMC/NPC cultures, making it impossible to evaluate whether the effect of CD70 expressed by NPC is mediated through metabolism.

Reply: Thank you very much for your questions. In most of our metabolism assays except fillipin III staining (where we co-stained fillipin III with PE-FOXP3 and APC-CTLA4), we could only quantify the changes in intracellular lipid/cholesterol/fatty acids, their metabolism processes and related genes in the entire co-culture system, without quantifying such subtle changes in both CD4⁺/FOXP3⁻ and CD4⁺/FOXP3⁺ cells separately. The first conclusion we can make so far is that CD70-CD27 interaction increases intracellular lipid contents and enhances multiple lipid metabolism processes in CD4⁺ naïve T cells and Tregs, because all of the co-culture experiments starts with 100% CD4⁺ naïve T cells in the system and CD70-CD27 interaction is shown to increase the Treg fraction in the co-culture system. The second conclusion we can make is that Tregs co-cultured with CD70-NC NPC cells exhibited higher lipid metabolism activities, shown by scRNA-seq and bulk RNA-seq analysis in our NPC clinical cohort, PBMC co-culture NPC cohort and multiple public cohorts (Figures 5E, 5F and 6J to 6O).

In addition, we conducted extensive literature research on the experimental association between lipid metabolism and Treg differentiation/activation. According to the recent advances, enhanced lipid metabolism, including but not limited to cholesterol metabolism, fatty acid metabolism and mitochondrial functions, is shown to facilitate Treg differentiation and activation, which echoes the major metabolism findings in our manuscript. The co-culture set-up for all of our lipid metabolism assays, including the T-cell activation time, the E-to-T ratio, co-culture time, IL-2/TGF- β concentrations, were exactly the same as the set-up we used in previous sections. For all assays performed in a lipid-depleted medium, we just changed the normal heat-inactivated FBS into lipid-depleted heat-inactivated FBS. In addition, CD70-CD27 interaction could no longer promote Treg-mediated immunosuppression in a lipid-depleted system, further corroborating that the effect of CD70 expressed by NPC is mediated through lipid metabolism (Figures 6P to 6X).

4. *No causal link between Treg metabolism and suppressive function of Tregs has been provided.*

Reply: Thank you very much for your comment. We performed the Treg suppression assay between CD8⁺ T cells and CD4⁺ naïve T cells after CD70-NC/CD70-KO co-culture in a lipid-depleted system. Our result revealed that lipid depletion significantly inhibited the effect of CD70-CD27 interaction on Treg suppressive activity (Figure S7G), compared to CD70-KO and inhibition in lipid-containing systems (Figures S3D and S4F). The data confirmed the link between lipid metabolism and the suppressive function of Tregs mediated by CD70⁺ NPC cells.

Reviewer #4 (Remarks to the Author):

All my comments have been addressed.

Dear reviewer,

It has been a pleasure to have your recognition of our revised manuscript. We have taken your comments and suggestions very seriously and tried to address your concerns as thoroughly as we could. Thank you very much again for helping us improve the quality of our manuscript.

REVIEWERS' COMMENTS

Reviewer #2 (Remarks to the Author):

All my comments have been addressed

Reviewer #3 (Remarks to the Author):

All my concerns have been addressed.

3rd Review

Reviewers' comments:

Reviewer #2 (Remarks to the Author):

All my comments have been addressed

Dear reviewer,

Thank you very much for your suggestions and support.

Reviewer #3 (Remarks to the Author):

All my concerns have been addressed.

Dear reviewer,

Thank you very much for your suggestions and support.